# Nano3P-seq: transcriptome-wide analysis of gene expression and tail dynamics using end-capture nanopore cDNA sequencing

Oguzhan Begik [1,2,3], Gregor Diensthuber[1], Huanle Liu [1],
Anna Delgado-Tejedor[1,4], Cassandra Kontur[5], Adnan Muhammad Niazi [6],
Eivind Valen[6,7], Antonio J. Giraldez [5], Jean-Denis Beaudoin [8],
John S. Mattick [2,9] & Eva Maria Novoa [1,4] ✉

RNA polyadenylation plays a central role in RNA maturation, fate, and stability. In response to developmental cues, polyA tail lengths can vary, affecting the translation efficiency and stability of mRNAs. Here we develop Nanopore 3′ end-capture sequencing (Nano3P-seq), a method that relies on nanopore cDNA sequencing to simultaneously quantify RNA abundance, tail composition, and tail length dynamics at per-read resolution. By employing a template-switching-based sequencing protocol, Nano3P-seq can sequence RNA molecule from its 3′ end, regardless of its polyadenylation status, without the need for PCR amplification or ligation of RNA adapters. We demonstrate that Nano3P-seq provides quantitative estimates of RNA abundance and tail lengths, and captures a wide diversity of RNA biotypes. We find that, in addition to mRNA and long non-coding RNA, polyA tails can be identified in 16S mitochondrial ribosomal RNA in both mouse and zebrafish models. Moreover, we show that mRNA tail lengths are dynamically regulated during vertebrate embryogenesis at an isoform-specific level, correlating with mRNA decay. Finally, we demonstrate the ability of Nano3P-seq in capturing non-A bases within polyA tails of various lengths, and reveal their distribution during vertebrate embryogenesis. Overall, Nano3P-seq is a simple and robust method for accurately estimating transcript levels, tail lengths, and tail composition heterogeneity in individual reads, with minimal library preparation biases, both in the coding and non-coding transcriptome.

RNA molecules are subject to multiple co- and post-transcriptional modifications, shaping them to their final mature form[1]. Polyadenylation of RNA is one such modification, which is known to affect the stability and translation efficiency of the RNA molecule[2–4] and to play an essential role in determining the fate of RNA molecules in a wide range of biological processes[5,6].

One context in which polyadenylation has been shown to play a major role in determining RNA fate and decay is vertebrate embryogenesis[6]. Indeed, in the first hours after fertilization, vertebrate embryos undergo major cellular reprogramming, a process known as the maternal-to-zygotic transition (MZT)[7]. During the MZT, maternally inherited RNA and proteins are responsible for activation of the zygotic

genome and are later replaced by the zygotic program[8,9]. Because the MZT begins in a transcriptionally silent embryo, this transition relies heavily on post-transcriptional regulatory mechanisms[7], including modulation of the polyadenylation status of the RNA molecules[6,10]. Therefore, characterizing the dynamics of RNA polyadenylation is key to understanding how these modifications regulate the fate and function of RNA molecules.

In the past few years, several transcriptome-wide methods have become available for studying the dynamics of polyadenylated tails (polyA tails) based on next-generation sequencing (NGS), such as PAL-seq or TAIL-seq[10,11]. While these methods have been successfully employed to characterize the dynamics of polyA tail lengths in various contexts, they have several caveats: (i) they provide a limited perspective on isoform–tail relationships owing to the short-read-length nature of NGS-based technologies; (ii) they do not provide single-molecule resolution; (iii) they are severely affected by PCR amplification biases; and (iv) they can only measure tail lengths that are shorter than the read length.

To overcome these limitations, the direct RNA sequencing (dRNA-seq) platform offered by Oxford Nanopore Technologies (ONT) has been proposed as a means to study both the transcriptome and polyA tail lengths simultaneously[12,13]. To sequence native RNAs using dRNA-seq, polyA-tailed RNA molecules are ligated to a 3′ adapter that contains an oligo(dT) overhang. Consequently, dRNA-seq libraries capture the full-length polyA tail; however, ligation occurs only on RNA molecules that anneal to the oligo(dT) overhang, thus exclusively capturing polyA transcripts with tail lengths greater than 10 nucleotides. A variation of this method consisting of in vitro poly(I)-tailing the transcriptome before library preparation has been proposed for studying nascent RNAs using dRNA-seq[14,15], thus capturing both polyadenylated and non-polyadenylated mRNAs. However, major limitations to this variation include low sequencing yields compared with standard dRNA-seq (10–30%)[14] and a lack of tools to distinguish poly(I) and poly(A) signals; therefore, polyA tail length information is lost in these datasets[14,15]. An alternative approach for studying the transcriptome using nanopore technologies is direct cDNA sequencing (dcDNA-seq), but this approach cannot sequence the polyA⁻ transcriptome, nor can it capture polyA tail length information. Overall, both dRNA-seq and dcDNA-seq nanopore library preparation protocols are limited to the sequencing of polyadenylated transcripts and thus cannot provide a comprehensive view of both polyadenylated and deadenylated RNA molecules, in addition to being unable to capture RNA molecules with other types of RNA tails (for example, polyuridine).

Here we present a novel method that employs nanopore sequencing to simultaneously obtain per-isoform transcriptome abundance and tail lengths in full-length individual reads, with minimal library preparation steps, which we term Nanopore 3′-end-capture sequencing (Nano3P-seq). Notably, Nano3P-seq uses template switching to initiate reverse transcription and, therefore, does not require 3′ end adapter ligation steps, PCR amplification, or second-strand cDNA synthesis. We demonstrate that Nano3P-seq can capture any type of RNA molecule regardless of its 3′ sequence, including polyA-tailed and non-tailed RNA. Moreover, we show that Nano3P-seq can accurately quantify RNA abundances in both the coding and non-coding transcriptome, as well as quantify the polyA tail lengths and tail composition of individual RNA molecules in a highly reproducible manner.

## Results
### Nano3P-seq robustly captures both polyadenylated and non-polyadenylated RNA
Because nanopore sequencing is typically limited to the analysis of polyA⁺ RNA molecules (Fig. 1a), previous efforts have opted to perform in vitro polyadenylation reactions of the total RNA to capture non-polyadenylated RNA in the sequencing run[16]. While this option can capture any given transcript present in the sample, it also leads to

a loss of polyA tail length information. Therefore, we reasoned that by coupling template switching to cDNA nanopore sequencing, we would simultaneously capture the polyA⁺ and polyA⁻ transcriptome, while retaining polyA tail length information from each individual RNA molecule (Fig. 1a).

To assess the ability of Nano3P-seq to sequence both polyA⁺ and polyA⁻ RNA, we first sequenced two synthetic RNAs, one lacking a polyA tail and one that had been in vitro polyadenylated (Methods) (Extended Data Fig. 1a–c). Our results show that Nano3P-seq efficiently captures both polyadenylated and non-polyadenylated RNA molecules, as well as the diversity of polyA tail lengths in individual RNAs (Extended Data Fig. 1c). We then examined the performance of Nano3P-seq on in vivo samples, and sequenced total RNA samples from mouse brain, previously enriched in nuclear and mitochondrial content via subcellular fractionation to increase the content of non-coding RNAs[17] (Methods). We confirmed that Nano3P-seq captured RNA biotypes that are typically polyadenylated (that is, mRNA, long intervening non-coding RNAs (lincRNA), and processed transcript) as well as non-polyadenylated (that is rRNA, miscellaneous RNA (miscRNA), small nuclear RNA (snRNA), small nucleolar RNA (snoRNA)), the majority of them being rRNA, mRNA, and snRNA (Fig. 1b and Extended Data Fig. 1d). In addition, our results confirmed that polyA tail length information was retained in individual reads. Specifically, the majority of reads corresponding to mRNAs had polyA tails (Fig. 1c and Extended Data Fig. 1e,f), whereas non-coding RNAs such as snoRNAs (Extended Data Fig. 1f) and snRNAs did not have polyA tails (Extended Data Fig. 1g), as expected.

To assess the accuracy and reproducibility of Nano3P-seq in quantifying RNA abundances, we examined the performance of Nano3P-seq in synthetic RNA mixes (sequins)[18] that had been spiked into samples in independent flow cells. Our results showed that Nano3P-seq provided accurate estimates of RNA abundances both at the per-gene (Pearson's $R = 0.93$, slope = 0.93) (Fig. 1d) and per-isoform (Pearson's $R = 0.89$, slope = 0.92) (Extended Data Fig. 1h) level. These quantifications were highly reproducible across biological replicates, both at the per-gene (Pearson's $R = 0.98$) and per-isoform (Pearson's $R = 0.97$) (Extended Data Fig. 1i) level. We should note that previous works using Illumina RNA-seq reported a global correlation of 0.89 and 0.86 between observed and expected sequin counts at per-gene and per-transcript level, respectively[18].

### Nano3P-seq recapitulates transcriptomic switch that occurs during MZT
Next, we employed Nano3P-seq to examine the RNA dynamics that occur during the maternal-to-zygotic transition (MZT) at single-molecule resolution (Fig. 2a). To this end, we isolated total RNA from zebrafish embryos at 2, 4, and 6 hours post-fertilization (h.p.f.) in biological duplicates, ribodepleted the samples, and sequenced them using the Nano3P-seq protocol (Extended Data Fig. 2a,b; Methods). Quantification of the mRNA abundances in three independent biological replicates showed that per-gene abundances (reads per million (RPM)) obtained using Nano3P-seq were highly reproducible (Fig. 2b and Extended Data Fig. 2c,d).

A comparative analysis of mRNA population dynamics across time points showed that Nano3P-seq recapitulated the transcriptomic switch that occurs during the MZT[7,8], with a decay of mRNA genes previously reported to have a 'maternal decay mode' (Fig. 2c), in agreement with previous studies[19,20]. Notably, in addition to polyadenylated RNAs, Nano3P-seq captured a wide variety of RNA biotypes without polyA tails that are also present in early embryo stages. We observed a significant increase in the abundance of lincRNAs and snoRNAs during the MZT, as well as a sharp increase in miRNA expression at 4 h.p.f., followed by a decrease at 6 h.p.f. (Fig. 2d). Analysis of individual miRNA gene expression patterns during the MZT revealed that the sharp increase observed at 4 h.p.f. was primarily caused by increased expression of genes belonging to the miR-430 family, which is known to be essential

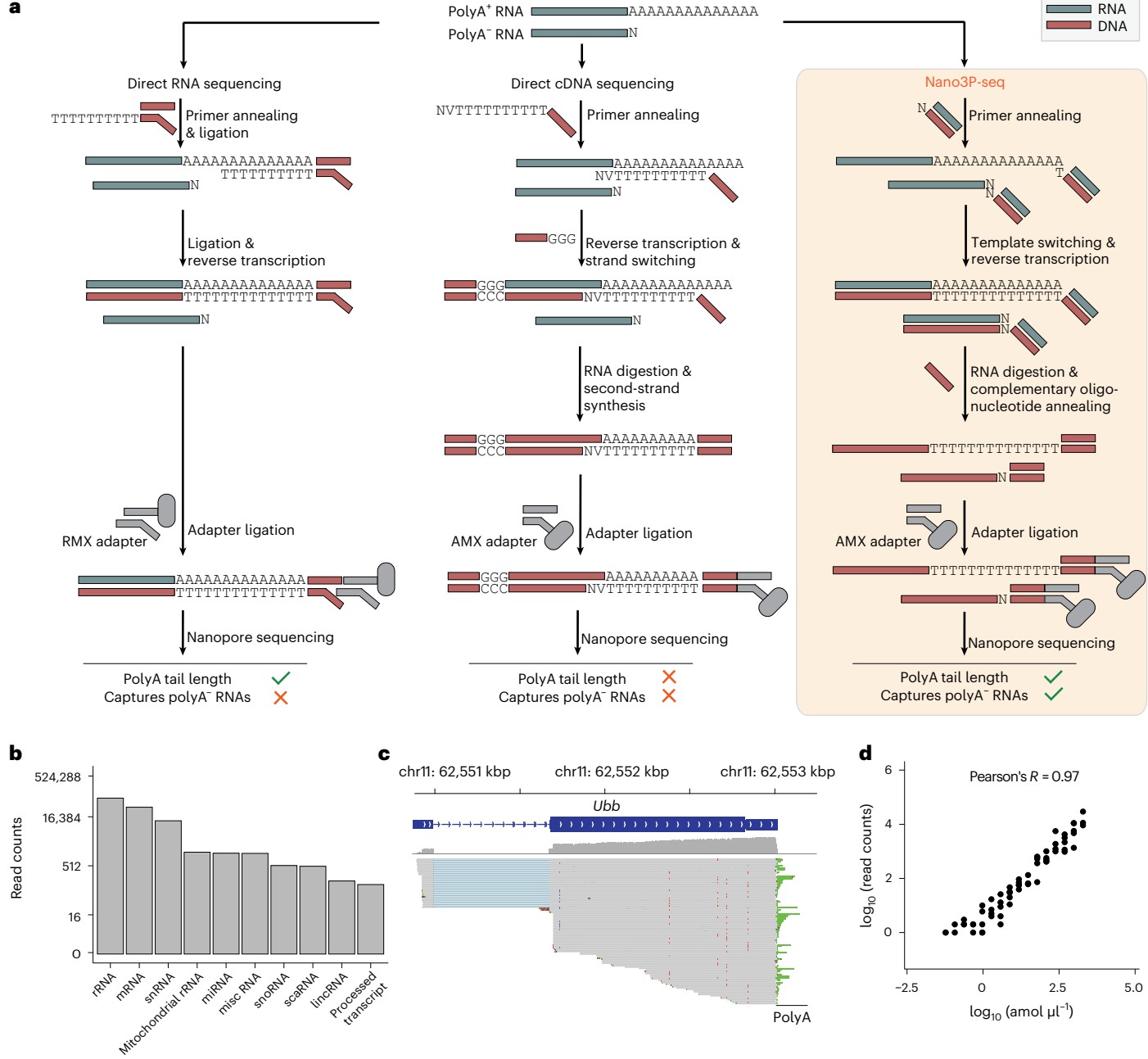

**Fig. 1 | Nano3P-seq captures polyadenylated and non-polyadenylated RNAs, while retaining polyA tail length information. a**, Schematic overview comparing three different library preparation methods for studying the transcriptome using nanopore sequencing. RMX, RNA adapter mix (provided with the SQK-RNA002 dRNA-seq library preparation kit); AMX, adapter mix (provided with the SQK-DCS109 dcDNA-seq library preparation kit). **b**, Nano3P-seq captures a wide range of RNA biotypes in a mouse brain nuclear/ mitochondrial RNA sample. **c**, Integrative Genome Viewer (IGV) snapshot of reads generated with Nano3P-seq, mapped to the *Ubb* gene, illustrating the diversity of polyA tail lengths captured across different reads. The polyA tail region is shown in green. kbp, kilobase pairs; scaRNA, small Cajal body-specific RNA. **d**, Scatter plot of log transformed concentrations (amol µl$^{-1}$) and read counts of sequin genes (Pearson's $R = 0.93$, slope = 0.93). Each dot represents a sequin. See also Extended Data Fig. 1h,i.

in mediating the decay of a group of maternal mRNAs during the MZT (mirR-430-dependent decay)[21] (Extended Data Fig. 2e). By contrast, much fewer non-coding RNA populations were globally captured (relative to coding RNA populations) when dRNA-seq was applied to the same samples (Fig. 2e).

We noted, however, that mitochondrial rRNAs were not enriched in Nano3P-seq datasets relative to dRNA-seq datasets (Fig. 2f). Indeed, analysis of zebrafish mitochondrial rRNA reads revealed that many 16S mitochondrial rRNA reads contained a polyA tail, which explains the lack of enrichment of mitochondrial rRNAs in Nano3P-seq datasets

relative to dRNA-seq datasets (Extended Data Fig. 2f). In agreement with this observation, we found that polyA-tailed 16S mitochondrial rRNAs were present not only in zebrafish (Extended Data Fig. 2g), but also in mouse (Extended Data Fig. 2h,i), suggesting that this feature is conserved across species and not a sequencing artifact, in agreement with previous reports[22,23]. The presence of polyA tails in mouse 16S mitochondrial rRNAs was further validated using an orthogonal method (polyA tail length assay coupled to Sanger sequencing; Methods), which confirmed that 16S mitochondrial rRNA is polyadenylated (Extended Data Fig. 2j).

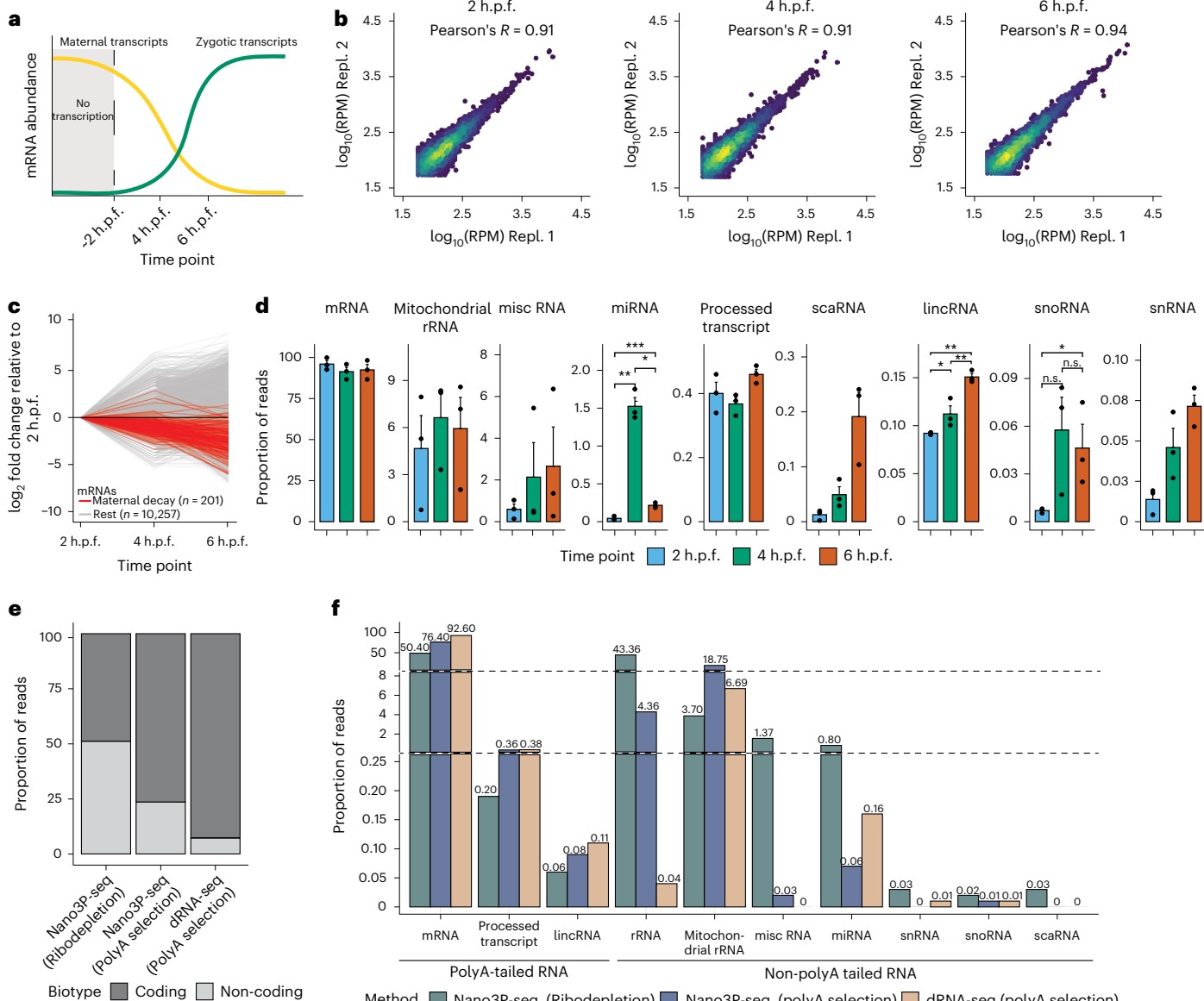

**Fig. 2 | Nano3P-seq captures a wide diversity of coding and non-coding RNAs and their expression dynamics during the maternal-to-zygotic transition (MZT). a**, Schematic overview of the transcriptional change that occurs during the MZT in zebrafish. **b**, Scatter plots depicting the correlation of mRNA log transformed RPM between biological replicates at three different time points during the MZT. **c**, Changes in mRNA abundance during the MZT ($t = 2$, 4, and 6 h.p.f.), relative to 2 h.p.f. Genes previously reported to have a 'maternal decay mode' are depicted in red. **d**, Bar plots depicting the abundance of different RNA biotypes captured by Nano3P-seq during the MZT (2, 4, and 6 h.p.f., shown in blue, green, and red, respectively). Statistical analyses were performed using the Kruskal–Wallis test. $n = 3$ biological replicates, data are presented as mean ± s.e.m. *$P \le 0.05$, **$P \le 0.01$, $P \le 0.001$, ****$P \le 0.0001$. n.s., not significant ($P > 0.05$). **e**, Relative proportion of coding and non-coding RNAs captured using dRNA-seq (on polyA-selected samples), Nano3P-seq (on polyA-selected samples), and Nano3P-seq (on ribodepleted samples). **f**, Percentage of reads mapping to distinct biotypes captured using Nano3P-seq (on ribodepleted samples) (green), Nano3P-seq (on polyA-selected samples) (blue), and dRNA-seq (on polyA-selected samples) (light brown).

## PolyA tail lengths can be accurately estimated using Nano3P-seq

We then examined whether Nano3P-seq accurately estimated polyA tail lengths. We note that algorithms for detecting polyA tails in native RNA nanopore sequencing reads are well established and benchmarked[12,24–26], but their applicability to cDNA reads, such as those from Nano3P-seq, remains unclear. To this end, we first examined whether the tailfindR polyA tail prediction software[25] would capture the presence or absence of polyA tails on synthetic RNAs that were either polyadenylated or non-polyadenylated and had been sequenced using Nano3P-seq. We found that tailfindR can identify both polyadenylated and non-polyadenylated Nano3P-seq reads (Fig. 3a and Extended Figure 3a). Then, we assessed the accuracy of polyA tail length

predictions of tailfindR in Nano3P-seq datasets that included a battery of synthetic RNAs (sequins)[18] or synthetic cDNA sequences with known polyA tail lengths (Fig. 3b). Our results showed that polyA tail length estimations of sequins in Nano3P-seq data were highly reproducible across replicates ($R = 0.993$; Extended Data Fig. 3b), with an accuracy similar to that observed when performing polyA tail length estimations in sequins that had been sequenced using dRNA-seq (Fig. 3c and Extended Data Fig. 3c). Moreover, the variance of tail length estimates across reads belonging to the same transcript was smaller in Nano3P-seq datasets than in dRNA-seq datasets (Extended Data Fig. 3d,e). Similar results were obtained when we assessed the performance of the algorithm in a set of synthetic cDNA sequences that

spanned a broader range of polyA tail lengths (0, 15, 30, 60, 90, and 120 nucleotides) (Fig. 3b,c and Extended Data Fig. 3f).

Finally, we performed a comparative analysis of mRNA polyA tail lengths from human, mouse, yeast, and zebrafish. We observed that mouse brain mRNAs, had the longest mRNA tails among the four species, with a median polyA tail length of 90 nucleotides, whereas the shortest polyA tail lengths were observed in yeast, with a median polyA tail length of 23.5 nucleotides (Fig. 3d), in agreement with previous studies[10].

### Charting polyA tail length dynamics in vivo using Nano3P-seq

We explored whether Nano3P-seq could be used to investigate polyA tail length dynamics in vivo. To this end, we first examined the ability of Nano3P-seq to identify which RNA biotypes were polyadenylated in mouse brain total RNA samples that had been previously enriched in nuclear/mitochondrial content to increase the proportion of non-coding RNAs. We found that polyA tails were mainly predicted on mRNAs, but also in lincRNAs and processed transcripts, which are also known to be polyadenylated[27,28] (Fig. 3e).

We next analyzed the polyA tail length dynamics across developmental stages of zebrafish mRNAs during the MZT ($t = 2$, 4, and 6 h.p.f.). PolyA tail length estimates were highly reproducible across independent biological replicates sequenced in independent flow cells for all three time points studied ($R = 0.85$–$0.95$) (Fig. 3f). We observed an overall increase in the mean mRNA polyA tail length during the MZT (Fig. 3g and Extended Data Fig. 3g), in agreement with previous reports[10]. All mRNAs examined were found to be polyadenylated, with the exception of histone mRNAs, which had a median polyA tail length of zero (Extended Data Fig. 3h), in agreement with previous works reporting their non-polyadenylated status[29]. These findings show that Nano3P-seq can capture RNA molecules with structured 3′ ends, such as those found in histones[30]. Finally, we note that per-gene polyA tail length estimates obtained by Nano3P-seq during the MZT showed a good correlation with those obtained using orthogonal methods such as PAL-seq[10] (Pearson's $R = 0.71$–$0.85$) (Extended Data Fig. 3i,j).

Next, we examined the correlation between polyA tail length dynamics and mRNA decay. To this end, mRNA transcripts were binned depending on their decay mode (maternal decay, zygotic activation-dependent decay, miR-430-dependent decay, and no decay), as previously described[21]. We observed that the three groups of mRNAs that are known to decay (maternal, zygotic, and miR-430) showed a significant decrease in mRNA abundance (Fig. 3h), as expected. However, the patterns of polyA tail length dynamics strongly varied depending on the decay mode of the transcript (Fig. 3i). Specifically, we observed that transcripts that decayed in an miR-430-dependent manner showed a significant decrease in polyA tail length during the MZT, in agreement with previous studies[19,21]. By contrast, for mRNAs with the zygotic genome activation-dependent decay mode, this shortening only occurred after 4 h.p.f., and maternal mRNAs did not present a decrease in polyA tail length, but instead showed a consistent increase in tail length throughout the MZT. These observations are consistent with the reanalysis of PAL-seq data (Extended Data Fig. 3k,l). Overall, these results show that not all decay modes are associated with a reduction in transcript polyA tail lengths and demonstrate the applicability of Nano3P-seq to identify polyadenylated RNA populations, study their RNA abundance, and estimate their polyA tail length dynamics, at both the global level and the level of individual transcripts, and thus provide mechanistic insights into different gene regulatory programs.

### Nano3P-seq captures isoform-specific polyA tail length changes during the MZT

A major feature that distinguishes nanopore sequencing from NGS is its ability to produce long reads, allowing RNA polyadenylation dynamics to be studied at the isoform level. Therefore, we wondered whether Nano3P-seq could identify differentially polyadenylated transcript isoforms during the MZT.

To this end, we first examined whether polyA tail lengths significantly diverged across time points at the per-isoform level. We note that only reads mapping to genes encoding for at least two annotated isoforms, and with mapping coverage greater than ten reads per isoform, were maintained for further analyses. Our analyses revealed that 55.3% (±8.62%) of analyzed transcripts varied significantly in polyA tail length ($P < 0.05$) during the MZT. Notably, we observed that analyses at the per-gene level often revealed a different picture compared with analyses at per-isoform level. For example, in *khdrbs1a* and *syncrip*, per-isoform analysis revealed opposite tail length dynamics among isoforms during the MZT, with one isoform decreasing and another isoform increasing in polyA tail length as the MZT progressed (Fig. 4a,b and Extended Data Fig. 4a,b).

Next, we compared isoform-specific polyA tail lengths across isoforms encoded by the same gene and found that 17.3% (±6.7%) of analyzed genes presented significant differences in their polyA tail lengths across isoforms (Fig. 4c and Extended Data Fig. 4c). Altogether, these results show that polyA tail length dynamics are not only dependent on the gene and embryogenesis stage, but are also specific to individual transcript isoforms. Moreover, these findings demonstrate that Nano3P-seq can provide transcriptome-wide measurements of the polyadenylation status of diverse biological samples with both single-read and single-isoform resolution.

### Detection of isoform-specific RNA modifications using Nano3P-seq

RNA molecules are decorated with chemical modifications, which are essential for the stability, maturation, fate, and function of the RNA[31–34]. Some modifications occur in base positions that are involved in Watson–Crick (WC) base pairing, causing a disruption during reverse transcription. Consequently, these modifications can be seen as increased 'errors' and drop-off rates in RNA-seq datasets[35–38]. One example is the hypermodified base m¹acp³Ψ, which is present in the eukaryotic small subunit (SSU) rRNA[39] crucial for the final processing steps of precursor rRNA (pre-rRNA) into mature SSU rRNA[40,41].

---

**Fig. 3 | Nano3P-seq can be used to accurately estimate polyA tail lengths in individual molecules. a**, PolyA tail length estimates of non-polyadenylated (curlcake 1) and polyadenylated (curlcake 2) synthetic RNAs sequenced with Nano3P-seq. See also Extended Data Fig. 1a–c. nt, nucleotides. **b**, Schematic overview of the standards used to assess the tail length estimation accuracy of Nano3P-seq. **c**, Box plots depicting tail length estimations of RNA and cDNA standards sequenced with Nano3P-seq. Values on box plots indicate the median polyA tail length estimation for each standard. **d**, PolyA tail length distribution of yeast, zebrafish, and mouse mRNAs represented as single-transcript values (left) and per-gene medians (right). **e**, PolyA tail length estimates across different RNA biotypes from mouse brain total RNA enriched in nuclear/mitochondrial RNA. Each dot represents a read. **f**, Replicability of median per-gene polyA tail length estimations of zebrafish embryonic mRNAs between two biological replicates for three different time points (2, 4, and 6 h.p.f.). **g**, Median per-gene polyA tail length distribution of zebrafish embryonic mRNAs across zebrafish developmental stages (2, 4, and 6 hpf, shown in blue, green, and red, respectively) in three biological replicates (shown as full lines, dashed lines, and dotted/dashed lines, respectively). **h**, Comparative analysis of mRNA abundances (shown as log₁₀(RPM) counts) of zebrafish mRNAs binned according to their annotated decay mode (maternal decay, zygotic activation-dependent decay, miR-430-dependent decay, and no decay) during early embryogenesis ($t = 2$, 4, and 6 h.p.f.). **i**, Median per-gene polyA tail length estimations of zebrafish mRNAs binned according to their decay mode (maternal, miR-430, zygotic, and no decay) at 2, 4, and 6 h.p.f. For Fig. 3h,i; statistical analyses were performed using the Kruskal–Wallis test. **c,e,h,i**, The number of observations included in the analysis is shown below each box and violin plot. Box plot limits are defined by lower (bottom) and upper (top) quartiles. The bar indicates the median, and whiskers indicate ±1.5× interquartile range.

Therefore, we examined whether Nano3P-seq could capture RNA modification differences across precursor and mature rRNA molecules from distinct maturation stages. To this end, reads mapping to SSU rRNAs were assigned to either 'precursor' or 'processed' isoforms (on the basis of the location of the 3′ end of the read) and analyzed the sequencing error patterns in the two populations (Fig. 4d). We observed that the mismatch frequency (misincorporations from the reverse transcriptase) at the $m^1acp^3\Psi$-modified site was very high in mature rRNAs but not present in pre-rRNAs, suggesting that this

modification is only present in mature rRNA populations. The presence of the hypermodification was also accompanied by a drop-off in sequencing coverage at the $m^1acp^3\Psi$-modified site, in agreement with its ability to disrupt the Watson–Crick base pairing. Analysis of the 'error' signatures along the SSU transcripts showed that this position was the only one position to change between precursor and processed rRNA molecules (Fig. 4d). These results were also orthogonally confirmed in dRNA-seq datasets, although the difference between pre-rRNA and mature rRNA error patterns were less evident than in

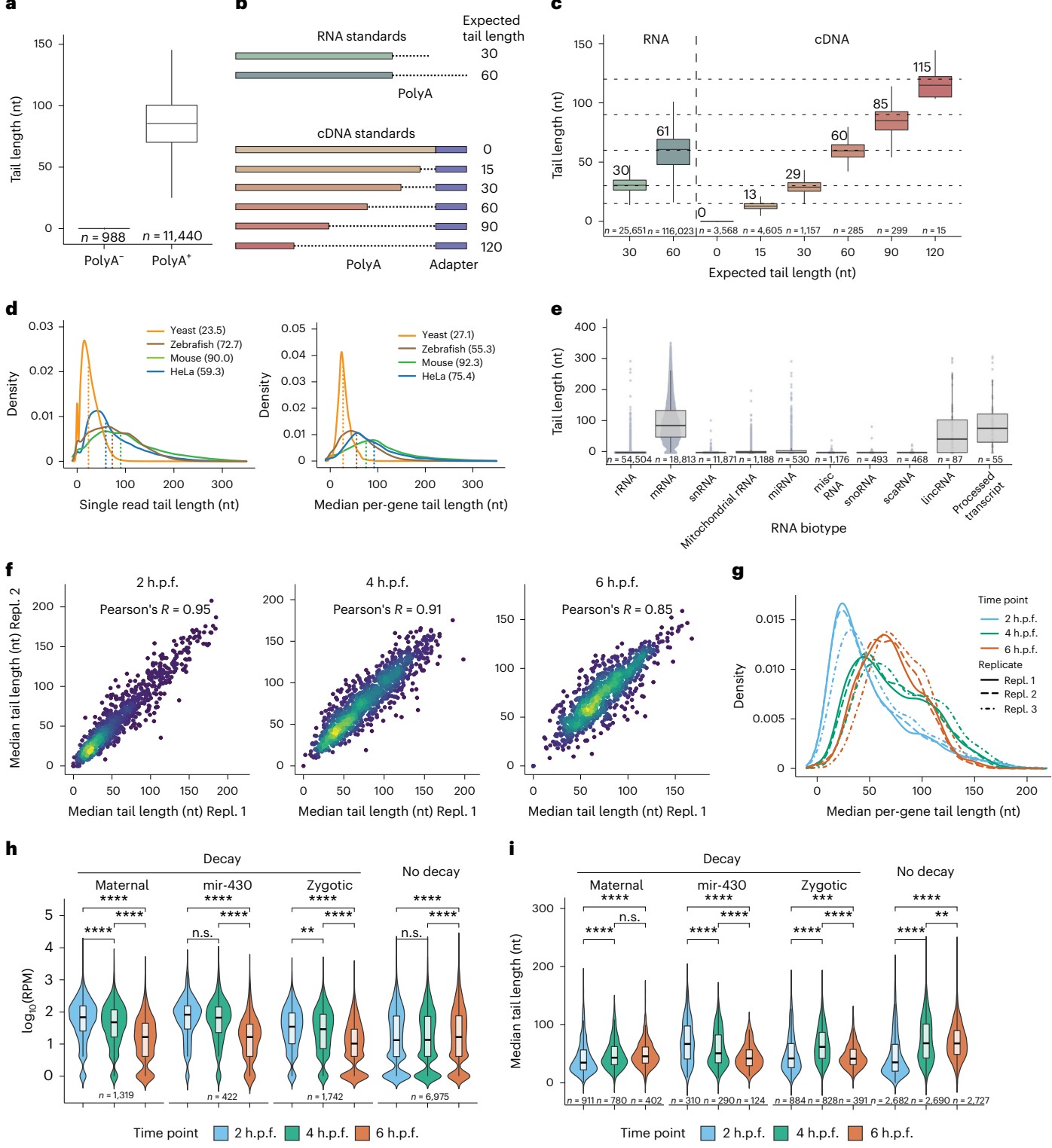

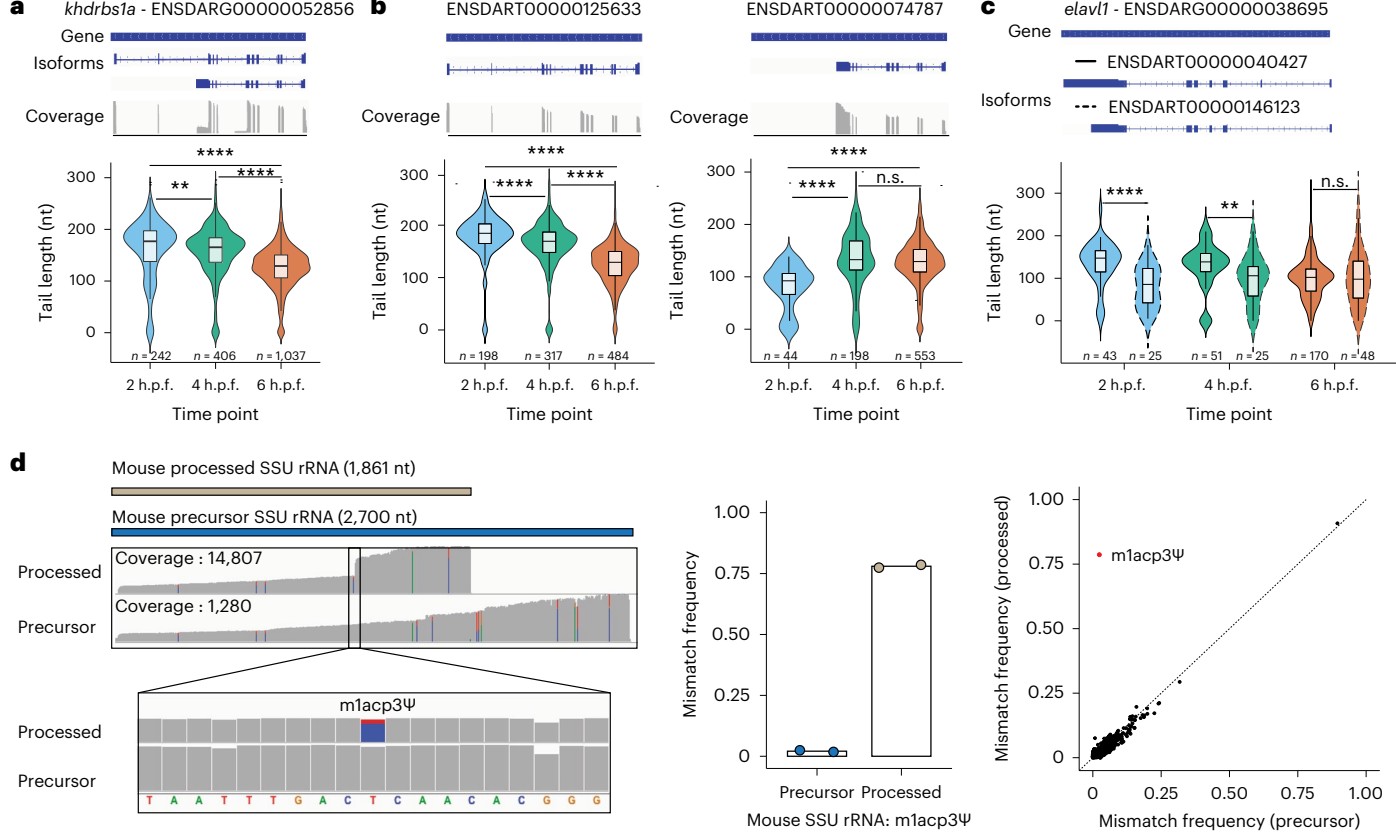

**Fig. 4 | Isoform-specific polyA tail and modification dynamics can be captured using Nano3P-seq. a**, Comparison of polyA tail length distributions of reads mapping to *khdrbs1a*, illustrated at the per-gene level, measured at three time points during the zebrafish MZT. Annotations of the gene and two main isoforms are shown at the top of the panels, along with an IGV coverage track of the reads mapping to the gene. **b,c**, Comparison of polyA tail length distributions of reads mapping to two distinct isoforms (full and dashed outline) of *elavl1* measured at three time points during the zebrafish MZT. Annotations of the gene and two main isoforms are shown at the top of the panels. **a–c**, Only isoforms with more than ten reads are shown. The number of reads included in the analysis is shown below each violin plot. *P* values have been computed using the Kruskal–Wallis test and corrected for multiple testing using the Benjamini–Hochberg method. Box plot limits are defined by lower (bottom) and upper (top) quartiles.

The bar indicates the median, and whiskers indicate ±1.5× interquartile range. **d**, IGV coverage tracks of reads mapping to mouse processed small subunit rRNA (top track) and precursor SSU rRNA (bottom track), including a magnified image at the position known to be modified with $m^1acp^3\Psi$ (left). Reads mapping to SSU rRNAs were assigned to either 'precursor' or 'processed' isoforms on the basis of the overlap between 3′ end of the read and annotated end of the isoforms. Only reads with 3′ ends within ±10 nucleotides of the annotated end of an isoform were kept. Positions with a mismatch frequency lower than 0.1 are shown in gray. Middle, the mismatch frequency values in mouse precursor and processed SSU rRNA at the position known to be modified with $m^1acp^3\Psi$ (*n* = 2 biological replicates) are shown. Right, the per-site mismatch frequencies observed in reads mapping to mouse precursor SSU rRNA and mouse processed SSU rRNA are compared, showing that the only outlier is $m^1acp^3\Psi$.

Nano3P-seq datasets, likely owing to the presence of $m^1\Psi$ modification (precursor of $m^1acp^3\Psi$) at this site in the pre-rRNAs, which also causes increased 'errors' in dRNA-seq datasets, but not in Nano3P-seq datasets (Extended Data Fig. 4d). Finally, we noted that the same pattern of $m^1acp^3\Psi$ modification was observed in yeast SSU rRNA (Extended Data Fig. 4e), in agreement with previous studies[41]. This analysis could not be performed in the zebrafish Nano3P-seq datasets owing to the absence of de novo transcription of zygotic rRNAs in early embryo stages[42]. Altogether, our results demonstrate that Nano3P-seq can identify isoform-specific and/or maturation-dependent RNA modification in the form of altered mismatch frequencies and/or reverse transcription drop-offs.

## Nano3P-seq identifies variations in tail composition during the MZT

Recent works using TAIL-seq have reported that a number of terminal modifications in polyA tails, such as polyuridine stretches, play a role in mRNA decay[11,20,43,44]. However, these methods cannot be used to detect tail modifications among the vast majority of tail nucleotides, as Illumina sequencing quality strongly deteriorates in homopolymeric stretches and with increased read length. By contrast,

Pacific Biosciences (PacBio)-based approaches such as FLAM-seq can sequence through the entire tail, however, they cannot unambiguously identify 3′ terminal modifications[45].

Therefore, we explored whether Nano3P-seq could accurately identify nucleotide composition variations (either internal or terminal) within polyA tails. To this end, we designed synthetic cDNA molecules with polyA tails ending with A (30A), U (1U), UUU (3U), UUUUU (5U), CCCCC (5C), and GGGGG (5G), as well as a polyA tail that contained several internal G nucleotides at fixed positions (IntG) (Fig. 5a; Methods). Synthetic molecules were sequenced using Nano3P-seq, and for each read, the 'tail' was defined as the set of nucleotides found between the nanopore adapter and the last nucleotide mapped to the cDNA standard reference (Fig. 5b and Extended Data Fig. 5a). Nucleotide composition analysis of the last 20 nucleotides of each synthetic tail revealed that Nano3P-seq accurately estimated the non-A base content in the tails (Extended Data Fig. 5b) and accurately identified the position in which these non-A bases were found (Fig. 5c).

Next, we analyzed the mRNA tail composition of zebrafish embryos from different developmental stages (2, 4, and 6 h.p.f.). Analysis of tail base abundance revealed that G was the most common non-A base in zebrafish mRNAs and that there was a significant decrease in non-A

bases with progression of the MZT (Extended Data Figure 6a,b). As a control, we examined the composition of bases in synthetic molecules ('sequins'), which are expected to contain an homogeneous polyA tail, and found that the frequency of non-A bases in biological sequences was significantly higher (6–15-fold) than in synthetic 'sequins' molecules (Extended Data Figure 6a), suggesting that the majority of non-A bases observed in Nano3P-seq biological datasets are not base-calling artifacts.

We then binned the tails on the basis of the composition of the last 30 nucleotides (All-A, contains only A; Int-G, contains internal Gs; Int-U, contains internal Us; Int-C, contains internal Cs; Term-G, contains terminal Gs; Term-U, contains terminal Us; Term-C, contains terminal Cs) (Fig. 5d and Extended Data Figure 6c). Our analysis showed that non-A internal bases were more abundant than non-A terminal bases, with internal Gs (Int-G) being the most abundant type of non-A bases in polyA tails (Fig. 5d). Moreover, we found that the abundance of transcripts with non-A base tails (internal or terminal) typically decreased with progression of the MZT, except for transcripts with terminal Us, for which the observed decrease was not statistically significant.

Finally, we examined the relationship between tail length and the presence of non-A bases in the tails. Although the median tail length increases during the MZT (Figs 5e and 3g), we found that this increase did not occur for all transcripts, as it depended on tail composition. Specifically, terminally uridylated tails (Term-U) were short (median lengths of 43, 50, and 32 nucleotides at 2, 4, 6 h.p.f., respectively) regardless of the time point examined (Fig. 4e,f, see also Table S1). An equivalent analysis in mouse Nano3P-seq datasets revealed that Term-U tails were also significantly shorter than the other tail types examined (Extended Data Figure 6d and Supplementary Table 1); by contrast, this trend was not observed in yeast Nano3P-seq datasets (Extended Data Figure 6e, and Supplementary Table 1). Our results are in agreement with previous studies in mouse and human cell lines showing that G and U bases are common in polyA tails and that tails ending with U bases are more frequent in shorter polyA tails[11,46].

## Assessment of tail length bias caused by polyA selection

Because the vast majority of cellular RNA is composed of rRNA, transcriptomic studies typically remove a significant portion of rRNA molecules to sequence a wider diversity of RNA biotypes. This removal can be achieved by (i) ribodepletion of the sample using biotinylated oligonucleotides that are complementary to rRNAs or (ii) selective enrichment of polyA$^+$ transcripts using oligo(dT) beads. Although these two approaches are often used interchangeably, their effects on the transcriptome composition are not equal. Nano3P-seq allows us to compare the effects of these two approaches on both the transcriptome composition and polyA tail length distribution. In terms of its effects on transcriptome composition, we found that ribodepletion captures a larger variety of RNA biotypes compared with polyA selection, including several non-polyA-tailed RNA biotypes, as expected (Fig. 2f). However, we did not observe a significant difference in the distribution of mRNA polyA tail lengths between the two methods (Extended Data Figure 7a–c), suggesting that, at least in zebrafish MZT transcriptomes, oligo(dT) enrichments do not significantly bias the polyA$^+$ mRNA populations by preferentially enriching for those with longer polyA tails.

## Comparison of Nano3P-seq to orthogonal methods

We then performed a comparative analysis of zebrafish polyA tail lengths, read lengths, and per-read quality in libraries sequenced using either Nano3P-seq or dRNA-seq. We found that Nano3P-seq captured RNA molecules regardless of their tail ends, resulting in the capture of diverse RNA biotypes (Fig. 1b and Extended Data Fig. 1c–g) including deadenylated mRNA molecules (Extended Data Figs 3h and 7) and molecules terminating with non-A bases in their 3′ ends (Fig. 5 and Extended Data Figs 5 and 6). By contrast, dRNA-seq only captured longer polyadenylated transcripts, as this method relies on the presence of polyA tail lengths greater than 10 nucleotides. Indeed, when comparing the distribution of per-read polyA tail length estimations of mRNAs, we observed that Nano3P-seq captured some mRNAs with predicted tail lengths of zero, whereas dRNA-seq only captured reads with longer tails (Extended Data Figure 7d,e). Moreover, we found that Nano3P-seq produced reads with significantly longer lengths than dRNA-seq (Extended Data Figure 7f), whereas the per-read qualities and 3′ end biases were relatively similar between the two methods (Extended Data Figure 7g–i), in agreement with previous works[12]. In terms of sequencing output, the yields of Nano3P-seq runs were similar—or slightly better—to those observed in dRNA-seq runs, producing ~100,000–200,000 reads in Flongle devices and ~500,000–2,000,000 in MinION devices, depending on the RNA input type and quality of the flowcell (Table S2).

In addition to comparing Nano3P-seq to dRNA-seq in matched zebrafish samples, we also compared Nano3P-seq polyA tail length estimates to publicly available datasets generated using orthogonal methods, including PAL-seq[10], TAIL-seq[11] PAT-seq[47], and FLAM-seq[45] (Extended Data Figs 3i and 8). The highest correlation was observed between Nano3P-seq and PAL-seq on zebrafish samples (Pearson's $R = 0.71$–$0.85$; Extended Data Fig. 3i and Supplementary Table 3). Correlations were more modest when comparing Nano3P-seq to FLAM-seq (Pearson's $R = 0.47$) and TAIL-seq (Pearson's $R = 0.19$) on HeLa cell lines (Extended Data Fig. 8a and Supplementary Table 3), as well as when comparing Nano3P-seq to PAT-seq and PAL-seq on *Saccharomyces cerevisiae* samples (Pearson's $R = 0.43$; Extended Data Fig. 8b and Supplementary Table 3) most likely due to the intrinsic differences between the samples, given that the correlations were also modest when comparing the estimates of polyA tail lengths across orthogonal methods ($R = 0.1$–$0.42$; Supplementary Table 3).

## Discussion

In the past few years, a variety of NGS-based high-throughput methods have been developed to characterize the 3′ ends of RNA molecules at a transcriptome-wide scale, including methods to reveal polyA tail sites (for example, 3P-seq[6], PAS-seq[48], and PAT-seq[47]) and to estimate polyA tail lengths (for example, PAL-seq[10], TAIL-seq[11], and mTAIL-seq[49]). However, a major limitation of NGS-based methods is their inability to assign a given polyA tail length to a specific transcript isoform, which causes a loss of isoform-specific tail length information. In addition, NGS-based methods cannot measure tail lengths greater than the read length, thus biasing our view of polyA tail dynamics to those transcripts that display shorter tail lengths.

---

**Fig. 5 | Analysis of tail composition using Nano3P-seq. a**, Schematic overview of the standards used to assess the ability of Nano3P-seq to accurately quantify the base content of polyA tails. **b**, IGV snapshots of nucleotide composition in cDNA standard tails sequenced using Nano3P-seq. Gray regions indicate the mapped part of the reads, whereas colored letters indicate soft-clipped bases (unmapped), which are the base-called tails, after trimming the adapter. **c**, Probability of base composition (A, green; G, orange; C, blue; U, red) per position in the last 20 nucleotides of the cDNA standard tails. See also Supplementary Note 1. **d**, Percentage of reads belonging to groups classified on the basis of their polyA tail base composition. Some sequence examples belonging to different groups are illustrated below the bar plots. Samples in this analysis are embryonic

mRNAs across zebrafish developmental stages (2, 4, and 6 h.p.f., shown in blue, green, and red, respectively) in three biological replicates and a control that includes sequin R1 and R2 groups of RNAs (gray). Statistical comparison of means was performed using the Kruskal–Wallis test. $n = 3$ biological replicates, data are presented as mean ± s.e.m. **e**, PolyA tail length estimation distributions of mRNA reads belonging to groups classified on the basis of their polyA tail base composition across zebrafish development stages (2, 4, and 6 h.p.f.). **f**, Left, IGV snapshots of reads mapping to zebrafish *actb* mRNA. Right, zoomed images of individual reads with different terminal bases (top, all-A reads; middle, Term-G reads; bottom, Term-U reads) are shown.

More recently, methods for estimating polyA tail lengths using PacBio long-read sequencing technologies have been developed, such as FLAM-seq[45] and PAIso-seq[46]. In contrast to NGS, these methods can capture isoform–tail relationships; however, they are still affected by PCR amplification and ligation biases, in addition to producing relatively modest outputs in terms of the number of reads[50–52]. In this

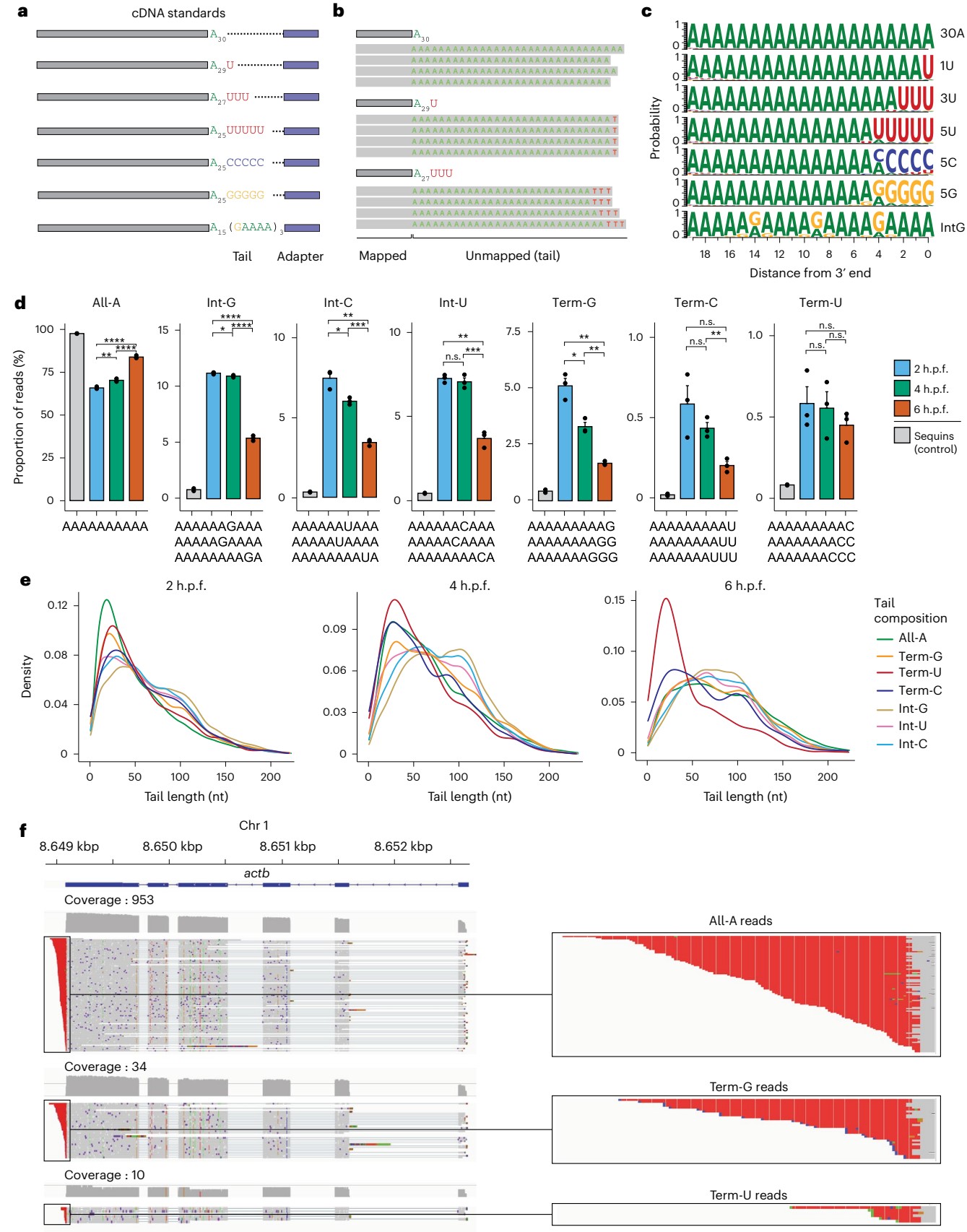

regard, several works have concluded that nanopore sequencers do not suffer from GC-content biases, as these are mainly introduced during the PCR amplification step[53–55].

Nanopore dRNA-seq has been proposed as an alternative long-read sequencing technology for studying polyA tail lengths[12,13]; however, the standard dRNA-seq approach cannot capture deadenylated RNAs, molecules with non-canonical tailings (for example, polyuridine), or molecules with polyA tails shorter than 10 nucleotides, thus biasing the view of the transcriptome toward polyadenylated molecules. Customized dRNA-seq methods involving in vitro poly(G/I)-tailing have been developed to overcome some of these limitations, but a lack of bioinformatic tools to distinguish polyI and polyA signals limits their applicability to study polyA tail length differences across transcripts in these datasets[14,15]. In addition, dRNA-seq requires 500 ng RNA as input, whereas Nano3P-seq requires as little as 50 ng, thus decreasing the required input material by tenfold. Nano3P-seq addresses the current limitations by offering a simple and robust solution for studying the coding and non-coding transcriptome simultaneously regardless of the presence or absence of polyA tails or 3′ tail composition, without PCR or ligation biases, and with single-read and single-isoform resolution. Moreover, the use of thermostable group II intron reverse transcriptase (TGIRT) in the Nano3P-seq protocol not only maximizes the production of full-length cDNAs, but also ensures the inclusion of RNA molecules that are highly structured and/or modified, which would often not be captured (or their representation would be significantly biased) using standard viral reverse transcriptases[56,57].

Nano3P-seq also provides quantitative measurements of RNA abundances (Fig. 1d) and captures diverse RNA biotypes regardless of their tail end composition (Fig. 2d). We have shown that Nano3P-seq can be applied to diverse species with a wide range of polyA tail lengths (Fig. 3d,e) and can be used to study the dynamics of polyadenylation (Fig. 3f,g,i). Specifically, we have demonstrated that Nano3P-seq provides per-read-resolution transcriptome-wide maps of RNA abundance and polyadenylation dynamics during the zebrafish MZT.

Overall, our work demonstrates that Nano3P-seq can simultaneously capture both non-polyA-tailed and polyA-tailed transcriptomes, making it possible to accurately quantify the RNA abundances and polyA tail lengths at per-read and per-isoform levels, while minimizing the amount of biases introduced during library preparation. These features set Nano3P-seq as a simple, potent, and robust method that can provide mechanistic insights into the regulation of RNA molecules and improve our understanding of mRNA tailing processes and post-transcriptional control.

## Online content

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

[1]Centre for Genomic Regulation (CRG), The Barcelona Institute of Science and Technology, Barcelona, Spain. [2]Garvan Institute of Medical Research, Sydney, New South Wales, Australia. [3]Faculty of Medicine, University of New South Wales, Sydney, New South Wales, Australia. [4]Universitat Pompeu Fabra, Barcelona, Spain. [5]Department of Genetics, Yale University, New Haven, CT, USA. [6]Computational Biology Unit, Department of Informatics, University of Bergen, Bergen, Norway. [7]Sars International Centre for Marine Molecular Biology, University of Bergen, Bergen, Norway. [8]Department of Genetics and Genome Sciences, Institute for Systems Genomics, University of Connecticut Health Center, Farmington, CT, USA. [9]School of Biotechnology and Biomolecular Sciences, University of New South Wales, Sydney, New South Wales, Australia. ✉e-mail: eva.novoa@crg.eu

## Methods

### In vitro transcription of RNAs

The synthetic 'curlcake' sequences[58] (Curlcake 1, 2,244 bp and Curlcake 2, 2,459 bp) were in vitro transcribed using Ampliscribe T7-Flash Transcription Kit (Lucigen-ASF3507). Curlcake 2 was polyadenylated using *Escherichia coli* polyA Polymerase (NEB-M0276S). polyA-tailed RNAs were purified using RNAClean XP beads. The quality of the in vitro transcribed (IVT) products as well as the addition of polyA tail to the synthetic constructs was assessed using Agilent 4200 Tapestation (Extended Data Fig. 1a). Concentration was determined using Qubit Fluorometric Quantitation. The purity of the IVT products was measured with NanoDrop 2000 Spectrophotometer.

### Poly(A) tail length assay and Sanger sequencing

Poly(A) Tail-Length Assay Kit (Thermo Fisher, 764551KT) was used according to the manufacturer's instructions. PCR products corresponding to the tail- and gene-specific primer combinations of mouse 16S rRNA (universal forward primer: 5′-GGTCGGTTTCTATCTATTTACGATTTCTC-3′, gene-specific reverse primer: 5′-TTCTCTAGGTTAGAGGGTGTACGTA TAT-3′) and human *ACTB* (assay control, primer composition not disclosed by manufacturer) were loaded on a 2.5% agarose gel (Lonza, 50010) and stained with GelRed (Merck, SCT123). Product sizes determined using the GeneRuler 50-base-pair DNA ladder (Thermo Scientific, SM0371). Subsequently, tail- and gene-specific PCR products of mouse 16S rRNA were purified by gel-elution (Cytiva Life Sciences, 28903470) and sent for Sanger sequencing with the shared forward primer (5′-GGTCGGTTTCTATCTATTTACGATTTCTC-3′). Resulting chromatograms were analyzed using SnapGene (v.6.0.2). After confirming alignment to the reference sequence, the unclipped chromatograms were used to visualize 3′ ends.

### Yeast culturing and total RNA extraction

*Saccharomyces cerevisiae* (strain BY4741) was grown at 30 °C in standard YPD medium (1% yeast extract, 2% Bacto Peptone and 2% dextrose). Cells were then quickly transferred into 50-ml pre-chilled falcon tubes, and centrifuged for 5 min at 3,000*g* in a 4 °C pre-chilled centrifuge. Supernatant was discarded, and cells were flash frozen. Flash frozen pellets were resuspended in 700 µl TRIzol (Life Technologies) with 350 µl acid washed and autoclaved glass beads (425–600 µm, Sigma G8772). The cells were disrupted using a vortex on top speed for 7 cycles of 15 s (the samples were chilled on ice for 30 s between cycles). Afterwards, the samples were incubated at room temperature for 5 min and 200 µl chloroform was added. After briefly vortexing the suspension, the samples were incubated for 5 min at room temperature. Then they were centrifuged at 14,000*g* for 15 min at 4 °C and the upper aqueous phase was transferred to a new tube. RNA was precipitated with 2× volume molecular grade absolute ethanol and 0.1× volume sodium acetate (3 M, pH 5.2). The samples were then incubated for 1 h at −20 °C and centrifuged at 14,000*g* for 15 min at 4 °C. The pellet was then washed with 70% ethanol and resuspended with nuclease-free water after air drying for 5 min on the benchtop. Purity of the total RNA was measured with the NanoDrop 2000 Spectrophotometer. Total RNA was then treated with Turbo DNase (Thermo, #AM2238) (2 µl enzyme for 50 µl reaction of 200 ng µl⁻¹ RNA) at 37 °C for 15 min, with a subsequent RNAClean XP bead cleanup.

### Mice breeding

Experiments were performed with male mice aged between 8 and 10 weeks. All mice were euthanized using $CO_2$ and tissues were snap frozen in liquid nitrogen. Animals were kept on a 12:12 h light:dark cycle and provided with water and food ad libitum.

### RNA isolation from mouse brain

To isolate nuclear/mitochondrial-enriched RNA from the mouse (*Mus musculus*) brain, we followed previously published protocols (https://doi.org/10.17504/protocols.io.3fkgjkw) with minor changes. A quarter of a C57BL6/J mouse brain was used for this protocol, and all samples and reagents were kept on ice during the protocol. Brain tissue was mined with a razor blade into smaller pieces. Cold Nuclei EZ Lysis Buffer (0.01 M Tris-Cl, pH 7.5, 0.06 M KCl, 0.001 M EDTA, 0.5% NP40, 1× Protease Inhibitor (complete Protease Inhibitor Cocktail Tablets, #11697498001, Roche)) was added to the tissue in 1.5-ml eppendorf tube. The sample was homogenized using a 1-ml dounce (stroking ~10–20 times), and the homogenate was transferred into a 2-ml eppendorf tube. One milliliter of cold Nuclei EZ Lysis Buffer was added and mixed, followed by 4 min incubation on ice. During the incubation, the sample was gently mixed a couple of times using a pipette. Homogenate was filtered using a 70-µm strainer mesh, and the flowthrough was collected in a polystyrene round-bottom FACS tube and subsequently transferred into a new 2-ml tube. The sample was centrifuged at 500*g* for 5 min at 4 °C and the supernatant was removed. The nuclei/mitochondria enriched sample was resuspended in another 1.5 ml EZ Lysis buffer and incubated for 5 min on ice. The sample was centrifuged at 500*g* for 5 min 4 °C and the supernatant was discarded (cytoplasm). Five-hundred microliters of nuclei wash and resuspension buffer (1 M phosphate-buffered saline, 1% bovine serum albumin, 5 µl SUPERase In RNase Inhibitor (Thermo Fisher, AM2694)) was added to the sample and incubated for 5 min without resuspending to allow buffer interchange. After incubation, 1 ml nuclei wash and resuspension buffer was added and the sample was resuspended. The sample was centrifuged at 500*g* for 5 min at 4 °C. The supernatant was removed and only ~50 µl was left. Using 1.4 ml nuclei wash and resuspension buffer, the sample was resuspended and transferred to a 1.5-ml eppendorf tube. The last washing step was repeated and the pellet was resuspended in 300 µL nuclei wash and resuspension buffer. RNA was extracted using TRIzol (Life Technologies) following the manufacturer's protocol.

### Zebrafish breeding

Wild-type zebrafish (*Danio rerio*) embryos were obtained through natural mating of the TU-AB strain of mixed ages (5–18 months). Mating pairs were randomly chosen from a pool of 60 males and 60 females allocated for each day of the month. Embryos and adult fish were maintained at 28 °C.

### Zebrafish total RNA extraction and polyA selection

For RNA samples, 25 embryos per developmental stage and per replicate were collected and flash frozen in liquid nitrogen. Frozen embryos were thawed and lysed in 1 ml TRIzol (Life Technologies) and total RNA was extracted using the manufacturer's protocol. Total RNA concentration was calculated by nanodrop.

For polyA-selected RNA samples, polyadenylated RNAs were isolated with oligo(dT) magnetic beads (New England BioLabs, S1419S) according to the manufacturer's protocol and eluted in 30 µl before nanodrop quantification.

### Zebrafish total RNA ribodepletion

Ribodepletion was performed on zebrafish total RNA using riboPOOL oligos (siTOOLs, 055) following the manufacturer's protocol. In brief, 5 µg total RNA in 14 µl was mixed with 1 µl resuspended riboPOOL oligos, 5 µl hybridization buffer and 0.5 µl SUPERase•In RNase Inhibitor (Thermo Fisher, AM2694). The mix was incubated for 10 min at 68 °C, followed by a slow cool down (3 °C min⁻¹) to 37 °C for hybridization. In the meantime, Dynabeads MyOne Streptavidin C1 (Thermo Fisher, 65001) beads were resuspended by carefully vortexing at medium speed. Eighty microliters of Dynabeads bead resuspension (10 mg ml⁻¹) was transferred into a tube, which then was placed on a magnetic rack. After aspirating the supernatant, 100 µl of bead resuspension buffer (0.1 M NaOH, 0.05 M NaCl) was added to the sample and beads were resuspended in this buffer by agitating the tube. Sample was placed on a magnet and the supernatant was aspirated. This step was performed twice. Beads were then resuspended in 100 µl bead wash buffer (0.1 M NaCl)

and placed on magnet to aspirate the supernatant. Beads were then resuspended in a 160 µl depletion buffer (5 mM Tris-HCl pH 7.5, 0.5 mM EDTA, 1 M NaCl). This suspension was then divided into two tubes of 80 µl, which will be used consecutively. Twenty microliters of hybridized riboPOOL and total RNA was briefly centrifuged to spin down droplets and it was pipetted into the tube containing 80 µl of beads in depletion buffer (10 mM Tris-HCl pH 7.5, 1 mM EDTA, 2 M NaCl). The tube containing the mix was agitated to resuspend the solution well. Then the mix was incubated at 37 °C for 15 min, followed by a 50 °C incubation for 5 min. Immediately before use, the second tube containing 80 µl of beads was placed on a magnetic rack and the supernatant was aspirated. After the incubation at 50 °C, the first depletion reaction was placed on a magnet and the supernatant was transferred into the tube containing the other set of beads. The mix was incubated again at 37 °C for 15 min, followed by a 50 °C incubation for 5 min. After briefly spinning down the droplets, the mix was placed on a magnet for 2 min. The supernatant was transferred into a different tube and cleaned up using RNA Clean & Concentrator-5 (Zymo, R1013).

### HeLa cell line culture and polyA RNA selection

HeLa cell lines (obtained from ATCC, *Mycoplasma*-tested in-house) were cultured in DMEM high glucose (Thermo Fisher, 41965) with 10% fetal bovine serum supplement (Thermo Fisher, #10270106). Pellets were obtained from 6 million cells and 1 ml TRIzol (Life Technologies) was added to each pellet, Sample was incubated at room temperature for a few minutes after adding 200 µl chloroform and vortexing briefly. After the incubation, the solution was centrifuged at 4 °C with 14,000*g*. Aqueous phase from the previous step was transferred to another eppendorf tube and equal amount of absolute ethanol (Merck, #1009835000) was added to the solution. This suspension was then transferred to an RNeasy Mini Spin Column from RNeasy Mini Kit (Qiagen, #74104), and total RNA was isolated following the manufacturer's protocol. Then, polyadenylated RNAs were isolated with Dynabeads Oligo(dT) 25 (Thermo Fisher, #61005) according to the manufacturer's protocol and eluted in 20 µL. Concentration and quality of RNA was evaluated using a Nanodrop 2000 Spectrophotometer and an Agilent 4200 TapeStation, respectively.

### Nano3P-Seq library preparation

The protocol is based on the direct cDNA Sequencing ONT protocol (DCB_9091_v109_revC_04Feb2019), with several modifications to be able to perform TGIRT template switching. Before starting the library preparation, 1 µl 10µM R_RNA (oligo: 5′ rGrArArGrArUrArGrArGrCr-GrArCrArGrGrCrArArGrUrGrArUrCrGrGrArArG/3SpC3/3′) and 1 µl 10 µM D_DNA (5′/5Phos/CTTCCGATCACTTGCCTGTCGCTCTATCTTCN 3′) were mixed with 1 µl 0.1 M Tris pH 7.5, 1 µl 0.5 M NaCl, 0.5 µl RNAse Inhibitor Murine (NEB, M0314S) and 5.5 µl RNase-free water. The mix was incubated at 94 °C for 1 min and the temperature was ramped down to 25 °C (−0.1 °C s⁻¹) in order to pre-anneal the oligos. Then, 50 ng RNA was mixed with 2 µl pre-annealed R_RNA + D_DNA oligo, 1 µl 100 mM dithiothreitol, 4 µl 5× TGIRT Buffer (2.25 M NaCl, 25 mM MgCl$_2$, 100 mM Tris-HCl, pH 7.5), 1 µl RNAse Inhibitor Murine (NEB, M0314S), 1 µl TGIRT (InGex) and nuclease-free water up to 19 µl. We should note that if 50 ng are used as input, only 1 µl TGIRT is needed, whereas if 100 ng is used as input, 2 µl TGIRT enzyme is needed. The reverse transcription mix was initially incubated at room temperature for 30 min before adding 1 µl 10 mM dNTP mix. Then the mix was incubated at 60 °C for 60 min and inactivated by heating at 75 °C for 15 min before moving to ice. RNAse Cocktail (Thermo Scientific, AM2286) was added to the mix to digest the RNA, and the mix was incubated at 37 °C for 10 min. The reaction was then cleaned up using 0.8× AMPure XP Beads (Agencourt, A63881). To be able to ligate the sequencing adapters to the first cDNA strand, 1 µl 10 µM CompA_DNA (5′ GAAGATAGAGCGACAGGCAAGTGATCG-GAAGA 3′) was annealed to the 15 µl cDNA in a tube with 2.25 µl 0.1 M Tris pH 7.5, 2.25 µl 0.5 M NaCl and 2 µl nuclease-free water. The mix was

incubated at 94 °C for 1 min and the temperature was ramped down to 25 °C (−0.1 °C s⁻¹) to anneal the complementary to the first-strand cDNA. Then, 22.5 µl first-strand cDNA was mixed with 5 µl Adapter Mix (AMX), 22.5 µl Rnase-free water and 50 µl Blunt/TA Ligase Mix (NEB, M0367S) and incubated in room temperature for 10 min. The reaction was cleaned up using 0.8× AMPure XP beads, using WSB (Washing buffer) buffer for washing. The sample was then eluted in elution buffer and mixed with sequencing buffer and loading beads before loading onto a primed R9.4.1 flowcell. Libraries were run on either Flongle or MinION flow cells with MinKNOW acquisition software version v.3.5.5. A detailed step-by-step Nano3P-seq protocol is provided as a Supplementary Protocol.

### Annealing-based dcDNA-seq library preparation with TGIRT

Some adjustments were made to the original Direct cDNA-Sequencing ONT protocol (SQK-DCS109), to be able to use TGIRT (InGex) as reverse transcription enzyme for nanopore sequencing, as this enzyme does not produce CCC overhang, which is typically exploited by the dcDNA-seq library preparation protocol (Fig. 1a). In brief, 1 µl 100 µM reverse transcription primer VNP (5′ /5Phos/ACTTGCCTGTCGCTC-TATCTTCTTTTTTTTTTTTTTTTTTTTTTVN 3′) and 1 µl 100 µM of in-house designed complementary oligo (CompA: 5′ GAAGATAGAGCGACAG-GCAAGTA 3′) were mixed with 1 µl 0.2 M Tris pH 7.5, 1 µl 1 M NaCl and 16 µl RNase-free water. The mix was incubated at 94 °C for 1 min and the temperature was ramped down to 25 °C (−0.1 °C s⁻¹) to pre-anneal the oligonucleotides. Then, 50 ng polyA-tailed RNA was mixed with 1 µl pre-annealed VNP + CompA, 1 µl 100 mM dithiothreitol, 4 µl 5× TGIRT Buffer (2.25 M NaCl, 25 mM MgCl2, 100 mM Tris-HCl, pH 7.5), 1 µl RNAse Inhibitor Murine (NEB, M0314S), 1 µl TGIRT and nuclease-free water up to 19 µl. The reverse transcription mix was initially incubated at room temperature for 30 min before adding 1 µl 10 mM dNTP mix. Then the mix was incubated at 60 °C for 60 min and inactivated by heating at 75 °C for 15 min before moving onto ice. Furthermore, RNAse Cocktail (Thermo Scientific, AM2286) was added to the mix to digest the RNA and the mix was incubated at 37 °C for 10 min. Then the reaction was cleaned up using 0.8× AMPure XP Beads (Agencourt, A63881). To be able to ligate the sequencing adapters the the first strand, 1 µl 10 µM CompA was again annealed to the 15 µl cDNA in a tube with 2.25 µl 0.1 M Tris pH 7.5, 2.25 µl 0.5 M NaCl and 2 µl nuclease-free water. The mix was incubated at 94 °C for 1 min and the temperature was ramped down to 25 °C (−0.1 °C s⁻¹) to anneal the complementary to the first-strand cDNA. Furthermore, 22.5 µl first-strand cDNA was mixed with 5 µl Adapter Mix (AMX), 22.5 µl Rnase-free water and 50 µl Blunt/TA Ligase Mix (NEB, M0367S) and incubated at room temperature for 10 min. The reaction was cleaned up using 0.8× AMPure XP beads, using WSB Buffer for washing. The sample was then eluted in elution buffer and mixed with sequencing buffer and loading beads before loading onto a primed R9.4.1 flowcell and run on a MinION sequencer with MinKNOW acquisition software version v.3.5.5.

### Synthetic cDNA standards

A total of 12 synthetic cDNA standards were synthesized as ultramers by IDT (Integrated DNA Technologies) to assess the tail length estimation and tail composition quantification accuracy of Nano3P-seq.

Synthetic cDNA standards designed to assess accuracy in tail length estimation:

cDNA_pA_standard_0: /5Phos/CTTCCGATCACTTGCCTGTCGCTC-TATCTTCGTAAATAGAAATAGACTAGCTCCACTTTTAAGAATTATTTATG-CAATTAAATACATGGGTGACCAAAAGAGCGGGCGGATACACGCGTCAC-CACAAGCAGAATAAAAGGTAAACCTGAAATTGTTTTAACATAAAAT-GAAAAATGCTTGTTTGCAACCCTATATAGAA

cDNA_pA_standard_15: /5Phos/CTTCCGATCACTTGCCTGTCGCTC-TATCTTCTTTTTTTTTTTTTTTTTGTAAATAGAAATAGACTAGCTCCACTTT-TAAGAATTATTTATGCAATTAAATACATGGGTGACCAAAAGAGCGGGCG-GATACACGCGTCACCACAAGCAGAATAAAAGGTAAACCTGAAATT-GTTTTAACATAAAATGAAAAATGCTTGTTTG

cDNA_pA_standard_30: /5Phos/CTTCCGATCACTTGC-CTGTCGCTCTATCTTCTTTTTTTTTTTTTTTTTTTTTTTTTTTTTTGTAAATAGAAATAGACTAGCTCCACTTTTAAGAATTATTTATGCAATTAAATACATGGGTGACCAAAAGAGCGGGCGGATACACGCGTCAC-CACAAGCAGAATAAAAGGTAAACCTGAAATTGTTTTAACATAAAATG

cDNA_pA_standard_60: /5Phos/CTTCCGATCACTTGCCT-GTCGCTCTATCTTCTTTTTTTTTTTTTTTTTTTTTTTTTTTTTTTTTTTTTTTTTTTTTTTTTTTTGTAAATAGAAATAGACTAGCTC-CACTTTTAAGAATTATTTATGCAATTAAATACATGGGTGACCAAAA-GAGCGGGCGGATACACGCGTCACCACAAGCAGAATAAAAG

cDNA_pA_standard_90: /5Phos/CTTCCGATCACTTGCCT-GTCGCTCTATCTTCTTTTTTTTTTTTTTTTTTTTTTTTTTTTTTTTTTTTTTTTTTTTTTTTTTTTTTTTTTTTTTTTTTTTTTTTTTTTGTAAATAGAAATAGACTAGCTCCACTTTTAAGAATTATT-TATGCAATTAAATACATGGGTGACCAAAAGAGCGGGCGG

cDNA_pA_standard_120: /5Phos/CTTCCGATCACTTGCCT-GTCGCTCTATCTTCTTTTTTTTTTTTTTTTTTTTTTTTTTTTTTTTTTTTTTTTTTTTTTTTTTTTTTTTTTTTTTTTTTTTTTTTTTTTTTTTTTTTTTTTTTTTTTTTTTTTTTTTGTAAATAGAAATA-GACTAGCTCCACTTTTAAGAATTATTTATGCAATT

Synthetic cDNA standards designed to assess accuracy in tail composition analyses:

cDNA_p29A_pU1_standard_30: /5Phos/CTTCCGATCACTT-GCCTGTCGCTCTATCTTCATTTTTTTTTTTTTTTTTTTTTTTTTTGTAAATAGAAATAGACTAGCTCCACTTTTAAGAATTATTTATGCAATTAAATACATGGGTGACCAAAAGAGCGGGCGGATACACGCGTCAC-CACAAGCAGAATAAAAGGTAAACCTGAAATTGTTTTAACATAAAATG

cDNA_p27A_pU3_standard_30: /5Phos/CTTCCGATCACTT-GCCTGTCGCTCTATCTTCAAATTTTTTTTTTTTTTTTTTTTTTTTTGTAAATAGAAATAGACTAGCTCCACTTTTAAGAATTATTTATGCAATTAAATACATGGGTGACCAAAAGAGCGGGCGGATACACGCGTCAC-CACAAGCAGAATAAAAGGTAAACCTGAAATTGTTTTAACATAAAATG

cDNA_p25A_pU5_standard_30: /5Phos/CTTCCGATCACTT-GCCTGTCGCTCTATCTTCAAAAATTTTTTTTTTTTTTTTTTTTTTTGTAAATAGAAATAGACTAGCTCCACTTTTAAGAATTATTTATGCAATTAAATACATGGGTGACCAAAAGAGCGGGCGGATACACGCGTCAC-CACAAGCAGAATAAAAGGTAAACCTGAAATTGTTTTAACATAAAATG

cDNA_p25A_pC5_standard_30: /5Phos/CTTCCGATCACTT-GCCTGTCGCTCTATCTTCGGGGGTTTTTTTTTTTTTTTTTTTTTTTGTAAATAGAAATAGACTAGCTCCACTTTTAAGAATTATTTATGCAATTAAATACATGGGTGACCAAAAGAGCGGGCGGATACACGCGTCAC-CACAAGCAGAATAAAAGGTAAACCTGAAATTGTTTTAACATAAAATG

cDNA_p25A_pG5_standard_30: /5Phos/CTTCCGATCACTTGC-CTGTCGCTCTATCTTCCCCCCTTTTTTTTTTTTTTTTTTTTTTTTTGTAAATAGAAATAGACTAGCTCCACTTTTAAGAATTATTTATGCAAT-TAAATACATGGGTGACCAAAAGAGCGGGCGGATACACGCGTCAC-CACAAGCAGAATAAAAGGTAAACCTGAAATTGTTTTAACATAAAATG

cDNA_pA_internalG_standard_30: /5Phos/CTTCCGATCACTT-GCCTGTCGCTCTATCTTCTTTTCTTTTCTTTTCTTTTTTTTTTTTTTGTAAATAGAAATAGACTAGCTCCACTTTTAAGAATTATTTATGCAATTAAATACATGGGTGACCAAAAGAGCGGGCGGATACACGCGTCAC-CACAAGCAGAATAAAAGGTAAACCTGAAATTGTTTTAACATAAAATG

## Sequencing and analysis of dRNA-seq datasets

dRNA-seq library preparations were prepared following manufacturer's recommendations, using 450 ng (in the case of polyA-enriched RNA zebrafish run) or 500 ng (in the case of in vitro transcribed 'sequins') as input material. Samples were sequenced in an R 9.4.1 MinION flowcell using a GridION sequencing device in the case of 'sequins', and in a R 9.4.1 PromethION flowcell using a PromethION sequencing device in the case of zebrafish RNA. For sequins, reads were base-called using stand-alone Guppy v.3.0.3 with default parameters and then the base-called reads were mapped to sequin sequences[18] with minimap2 with '-ax splice -k14 -uf --MD' parameters[59]. For zebrafish dRNA-seq samples, reads were base-called with Guppy v.4.0. Base-called reads

were first mapped to maternal and somatic zebrafish ribosomal RNA sequences taken from[42] and then to the genome (GRCz11) with minimap2[59] with '-ax splice -k14 -uf --MD' parameters. Mapped reads were intersected with ENSEMBL version 103 annotation (Danio_rerio.GRCz11.103.2.gtf) using bedtools v.2.29.1 intersect option[60].

## Analysis of Nano3P-seq datasets

All the Nano3P-seq runs were base-called and demultiplexed using stand-alone Guppy v.6.0.1 with default parameters. All runs were mapped using minimap2[59] with the following '-ax splice -k14 -uf --MD' parameters when mapping to genome and '-ax map-ont -k14 --MD' when mapping to transcriptome, unless stated otherwise.

For the analysis of synthetic RNA constructs (curlcakes), base-called reads were mapped to Curlcake 1 and 2 sequences[58], and mapped reads were then intersected with annotations of Curlcake 1 and 2 sequences to filter out the incomplete reads using bedtools v.2.29.1 intersect option. For yeast total RNA, we first mapped the base-called reads to S. cerevisiae ribosomal RNAs (25S, 18S, 5S and 5.8S) and then mapped the rest of the reads to the S. cerevisiae genome (SacCer3). Mapped reads were then intersected with SacCer64 annotation exon ends, to filter out incomplete reads. For the analysis of nuclear/mitochondrial-enriched mouse brain RNA spiked in with sequins[18], we first mapped the base-called reads to Mus musculus ribosomal RNAs (28S, 18S, 5S and 5.8S) and then mapped the rest of the rest of the reads to the M. musculus genome (GRCm38), supplemented with sequin chromosome (chrIS). Mapped reads were then intersected with ENSEMBL version 102 annotation (Mus_musculus.GRCm38.102.gtf) and sequin annotation (RNAsequins.v2.2.gtf) exon ends, to filter the incomplete reads. For zebrafish RNA (polyA-selected and ribodepleted), we first mapped the base-called reads to ribosomal RNAs and then mapped the rest of the reads to the genome (GRCz11). Mapped read starts were then intersected with ENSEMBL version 103 annotation (Danio_rerio.GRCz11.103.2.gtf) exon ends, to filter the incomplete reads. For HeLa mRNA, we first mapped the base-called reads to human ribosomal RNAs (28S, 18S, 5S and 5.8S); then, the rest of the reads (unmapped) were mapped to the Homo sapiens genome (GRCh38). Mapped read starts were then intersected with ENSEMBL version 107 annotation (Homo_sapiens.GRCh38.107.gtf) exon ends, to filter out incomplete reads (please see Nano3p-seq GitHub repository for detailed steps and script used: https://github.com/novoalab/Nano3P_Seq). For cDNA standards, base-called reads were mapped to cDNA sequence reference using minimap2, except for the cDNA_pA_standard_120, which was mapped using bwa short-read aligner[61] with the following parameters 'mem -xont2d'. A different aligner was used for cDNA_pA_standard_120 because minimap2, which is a long-read mapping algorithm, did not yield any mapped reads for this standard owing to its short length[62] (30 nucleotides once the pA tail length has been soft-clipped). We should note that the use of a different aligner should not affect the polyA tail length estimations of the reads, as these are done at the level of current intensity (that is, before mapping and base-calling). We should note that fewer reads were base-called and mapped to cDNA_pA_standard_120, cDNA_pA_standard_90 and cDNA_pA_standard_60, in comparison to the other cDNA standards (Supplementary Table 2). This phenomenon is probably caused by the inefficiency of nanopore sequencing to capture shorter cDNA sequences (cDNA_pA_standard_120 will have a mappable region of 30 nucleotides, and cDNA_pA_standard_90 will have a mappable region of 60 nucleotides, once the polyA tail region has been soft-clipped).

For the assignment of reads to isoforms, IsoQuant package (https://github.com/ablab/IsoQuant) was used with Danio_rerio.GRCz11.103.2.gtf annotation using the following parameters '--genedb gtf_file --complete_genedb --bam bam_file --data_type nanopore -o output'. A complete step-by-step command line of the bioinformatic analysis done on Nano3P-seq datasets can be found in the GitHub repository (https://github.com/novoalab/Nano3P_Seq). All reference

sequences used to map the runs mentioned above are included in the Nano3P-seq GitHub repository.

## Estimation of polyA tail lengths

For dRNA-seq reads, polyA tail length estimation was performed using NanoTail, a module from Master of Pores[63], a nextflow workflow for the analysis of direct RNA datasets, which uses internally Nanopolish v0.11.1[26]. In NanoTail, all reads stored in the fastq files are first indexed with 'nanopolish index' using default parameters, and the function 'nanopolish polya' is used to perform polyA tail length estimations. We should note that neither Nanopolish nor TailfindR estimate polyA tail lengths from the base-called sequence (owing to increased deletion frequencies observed in homopolymeric regions, which is an inherent problem of nanopore sequencing[64]); rather, they estimate polyA tail lengths by comparing the relative duration of the current signal corresponding to the polyA tail region to the total duration of the sequenced read.

For Nano3P-seq reads, polyA tail length estimation was performed using the Nano3P-seq version of tailfindr (https://github.com/adnaniazi/tailfindr/tree/nano3p-seq). All code used to estimate polyA tail lengths and post-process Nano3P-seq data can be found at https://github.com/novoalab/Nano3P_Seq.

## Analysis of tail composition

Base-called reads mapped to the zebrafish genome and cDNA standards for tail composition quantification were first trimmed using the Porechop tool (https://github.com/rrwick/Porechop) with the following parameters '--extra_end_trim 0 --end_threshold 50', to remove the adapter sequences. Because Porechop sometimes failed at removal of the adapter sequences, only reads containing more than 80% A bases in their tail composition were kept for downstream analyses, thus ensuring that untrimmed reads are not included in downstream analyses (Extended Data Figure 9). Finally, an in-house Python script was used to extract the tail FASTA sequences of the tail regions from trimmed reads, which has been made available in GitHub (https://github.com/novoalab/Nano3P_Seq/blob/master/python_scripts/soft_clipped_content.py).

## Animal ethics

Fish lines were maintained according to the International Association for Assessment and Accreditation of Laboratory Animal Care research guidelines, and protocols were approved by the Yale University Institutional Animal Care and Use Committee (IACUC). Mice maintenance was approved by the Garvan/St Vincent's Hospital Animal Ethics Committee, in accordance with the guidelines of the Australian Code of Practice for the Care and Use of Animals for Scientific Purposes (Project No. 16/14 and 16/26). All animals were entered into the study in a randomized order and operators were blinded to genotype and treatments.

## Reporting summary

Further information on research design is available in the Nature Portfolio Reporting Summary linked to this article.

## Data availability

Base-called FAST5 reads from Nano3P-seq and dRNA-seq libraries are publicly available in the European Nucleotide Archive, under accession code PRJEB53494. PAL-Seq polyA tail length estimates used in this work were obtained from the Gene Expression Omnibus (GEO) with the accession code GSE52809 (ref. [10]). PAT-seq polyA tail length estimates used in this work were obtained from GEO with the accession code GSE53461 (ref. [47]). FLAM-seq polyA tail length estimates used in this work were obtained from GEO with the accession code GSE126465 (ref. [45]). TAIL-seq polyA tail length estimates used in this work were obtained from GEO with the accession code GSE51299 (ref. [45]). All sequencing runs included in this work are listed in Supplementary Table 2. Source data are provided with this paper.

## Code availability

All scripts and code used in this work have been made available on GitHub (https://github.com/novoalab/Nano3P_Seq). The code for analyzing Nano3P-seq polyA tail lengths using tailfindR is available on GitHub (https://github.com/adnaniazi/tailfindr/tree/nano3p-seq) and is included as Supplementary Software.

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

## Acknowledgements

We thank all the members of the Novoa lab for their valuable insights and discussion. O.B. is supported by a UNSW International PhD fellowship and Australian Government Research Training Program Scholarship. G.D. is part of the ROPES ITN, which received funding from the European Union's Horizon 2020 research and innovation program under the Marie Sklodowska-Curie grant agreement number 956810. A.D.-T. is supported by an FPI-SO fellowship (PRE2019-088498). This work was supported by the Australian Research Council (DP180103571 to E.M.N.), the Spanish Ministry of Economy, Industry and Competitiveness (MEIC) (PGC2018-098152-A-100 to E.M.N.), the European Research Council (ERC-StG-2021 No 101042103 to E.M.N.) and the NIH (R35GM122580 and RO1HD00035 to A.J.G). We acknowledge the support of the MEIC to the EMBL partnership, Centro de Excelencia Severo Ochoa and CERCA Programme/Generalitat de Catalunya. We thank T. Mercer (Garvan Institute, Australia) for providing us with the sequins used as spike-ins in Nano3P-seq zebrafish runs. We thank the Sdelci lab (CRG, Spain) for providing us with HeLa cell lines pellets used in this work.

## Author contributions

O.B. performed the majority of wet laboratory experiments. O.B. analyzed the data, together with E.M.N. G.D. performed quantitative PCR experiments and G/I-based polyA tail length experiments to orthogonally validate Nano3P-seq predictions. H.L. contributed code for the analysis of polyA tail lengths and nucleotide content. A.D.-T. processed and analyzed the direct RNA sequencing zebrafish data. A.M.N. and E.V. adapted tailfindR code to accommodate the adapters that are used in Nano3P-seq libraries. C.K., A.J.G. and J.-D.B. provided the zebrafish total and polyA-selected RNA samples used in this study. E.M.N. conceived the project. E.M.N. supervised the work, with the assistance of J.S.M. O.B. and E.M.N. wrote the paper, with contributions from all authors.

## Competing interests

E.M.N. has received travel and accommodation expenses to speak at Oxford Nanopore Technologies conferences. E.M.N. is a member of the Scientific Advisory Board of IMMAGINA Biotech. All authors declare that the research was conducted in the absence of any

commercial or financial relationships that could be construed as a potential conflict of interest.

## Additional information

**Extended data** is available for this paper at https://doi.org/10.1038/s41592-022-01714-w.

**Correspondence and requests for materials** should be addressed to Eva Maria Novoa.

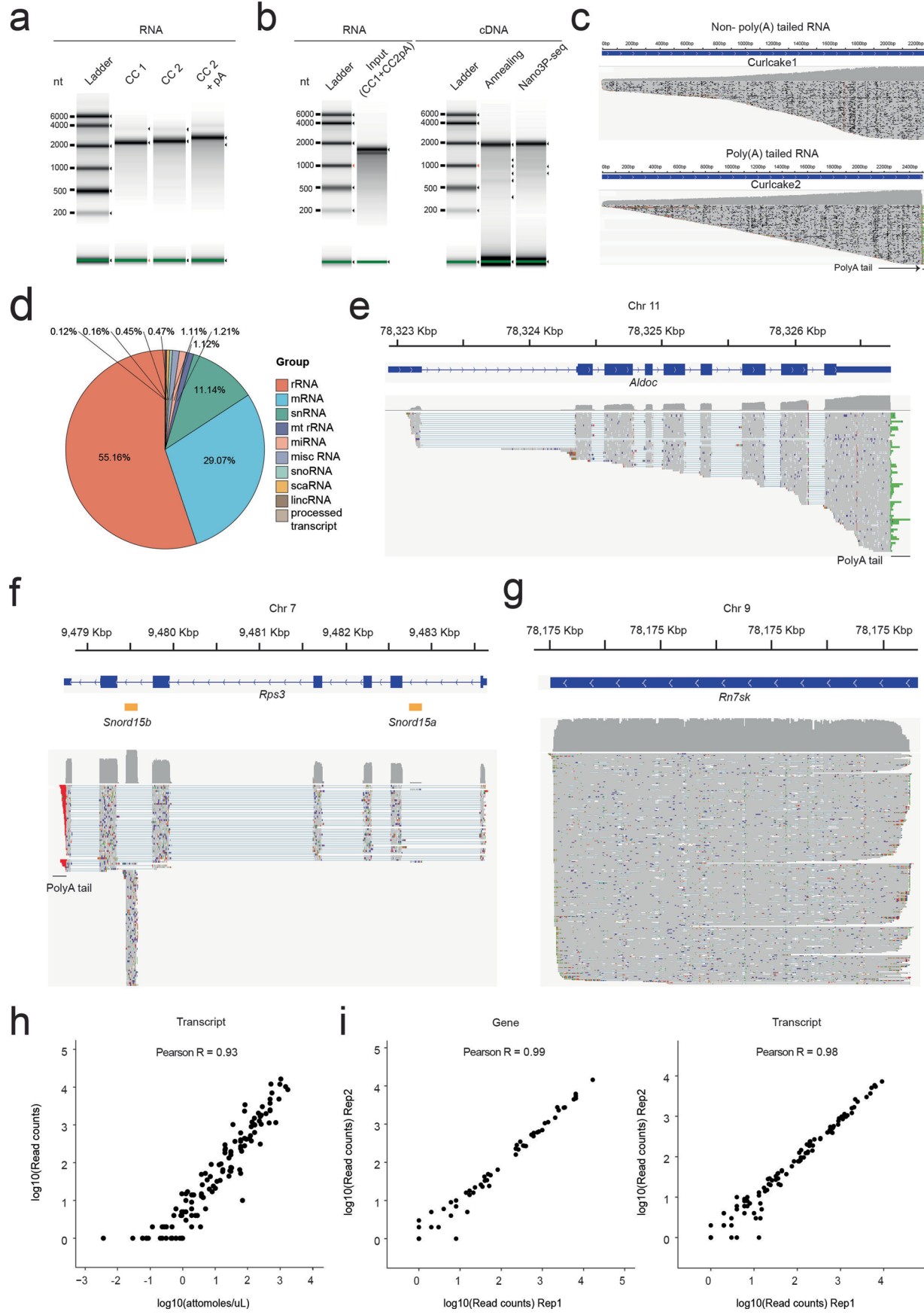

**Extended Data Fig. 1 | See next page for caption.**

**Extended Data Fig. 1 | Nano3P-seq captures non-polyA-tailed and polyA-tailed RNAs. (a)** Tapestation profiles of synthetic RNAs ('curlcakes') after being *in-vitro* transcribed and polyA tailed (pA). Similar profiles were consistently obtained in independent experiments. **(b)** Tapestation profiles of the input RNA (curlcake mix) for reverse-transcription and cDNA produced after annealing-based or template-switching based (Nano3P-seq) reverse-transcription. **(c)** IGV snapshots of synthetic RNAs (Curlcake1 and Curlcake2, see *Methods*) illustrating that Nano3P-seq captures both non-polyadenylated (above) and polyadenylated (below) RNAs. The PolyA tail region is shown in green. **(d)** Pie chart showing the abundance of different RNA types in Nano3P-seq of mouse nuclear/mitochondria enriched RNA. **(e)** IGV snapshot of reads mapping to *Aldoc* gene with polyA tail shown in green. **(f)** IGV snapshot of reads mapping to *Rps3* and *Snord15b* genes. PolyA tail can be seen in green on the reads mapping to *Rps3* mRNA, while it can't be seen in *Snord15b* snoRNA. **(g)** IGV snapshot of reads mapping to *Rn7sk* miscRNA, which are not expected to contain polyA tails. **(h)** Scatter plot of the log transformed concentrations (Attomoles/uL) and read counts of sequin transcripts (Pearson R: 0.89, Slope: 0.92). Each dot represents a sequin transcript. **(i)** Scatter plot of the replicability of the log (read counts) of synthetic sequins using Nano3P-seq, both at per-gene level (left panel, Pearson's R: 0.99) as well as per-transcript level (right panel, Pearson's R: 0.98).

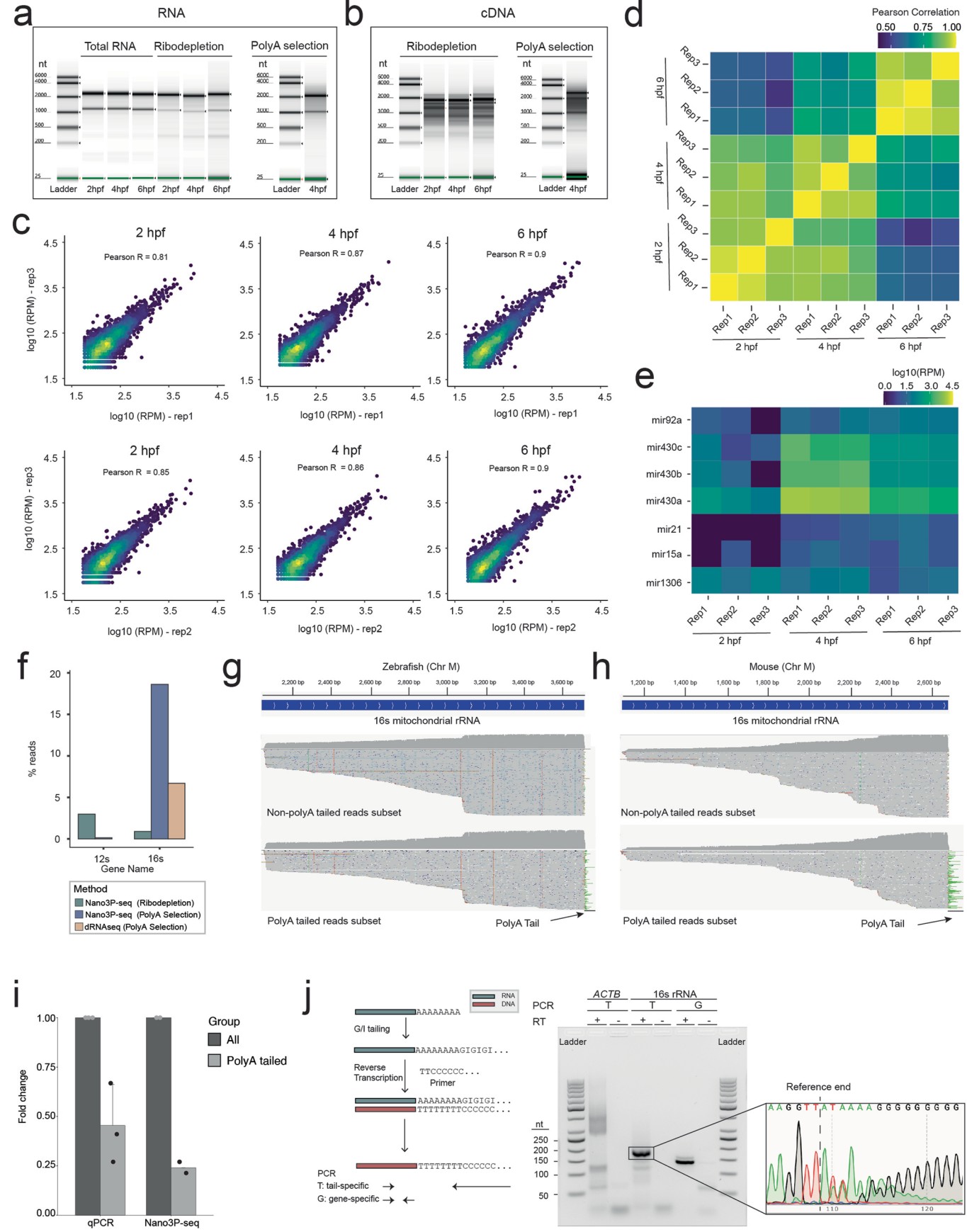

**Extended Data Fig. 2 | See next page for caption.**

**Extended Data Fig. 2 | Analysis of abundances and polyA tails in mitochondrial rRNAs. (a)** Tapestation profiles of total, ribodepleted and polyA-selected RNAs from zebrafish embryos collected in three different time points during the MZT. Similar TapeStation profiles were obtained across other biological replicates (n = 3). **(b)** Tapestation profiles of the reverse-transcription products of ribodepleted (left) and polyA-selected (right) samples from zebrafish embryos collected in three different time points during the MZT. Similar TapeStation profiles were obtained across other biological replicates (n = 3). **(c)** Scatterplots depicting the correlation of mRNA RPM (Read per million) levels biological replicates in three different time points during the MZT. **(d)** Pearson correlation matrix illustrating the similarity between biological replicates and different time points during the MZT. **(e)** Heatmap of log10(RPM) values of micro-RNAs in three different time points during the MZT in three biological replicates**. (f)** Percentage of reads mapping to 12 s and 16 s mitochondrial rRNAs in two different methods: Nano3P-seq of ribodepleted and polyA-selected samples and dRNA-seq of polyA-selected samples from zebrafish embryos at 4 hours post-fertilization. **(g)** IGV snapshot of reads mapping to zebrafish 16 s mitochondrial rRNA, where reads have been grouped as non-polyA tailed and polyA tailed based on their predicted polyA tail length. PolyA tail region is shown with an arrow and colored green. **(h)** IGV snapshot of reads mapping to mouse 16 s mitochondrial rRNA, where reads have been grouped as non-polyA tailed and polyA tailed based on their predicted polyA tail length. PolyA tail region is shown with an arrow and colored green. **(i)** Fold change of the polyA tailed 16 s mitochondrial rRNA amount to the total 16 s mitochondrial rRNA amount measured by qPCR and Nano3P-seq ($n$ = 3 technical replicates for qPCR and n = 2 biological replicates for Nano3P-seq. Data are presented as mean values+/− standard deviation). **(j)** Outline of PolyA Tail-Length Assay Kit (Thermo, #764551KT) (left panel). Agarose gel electrophoresis image of the PCR products of the PolyA Tail-Length Assay Kit illustrating bands in the tail-specific PCR of *ACTB* control and both tail-specific and gene-specific PCR of mouse 16 s mitochondrial rRNA (middle panel). Sanger sequencing result of the PCR product extracted from the agarose gel showing the presence of polyA tail after the reference end indicated by a dashed line (right panel). Also refer to Supplementary Note 2. PolyA Tail Length Assay experiments were performed in 2 independent replicates.

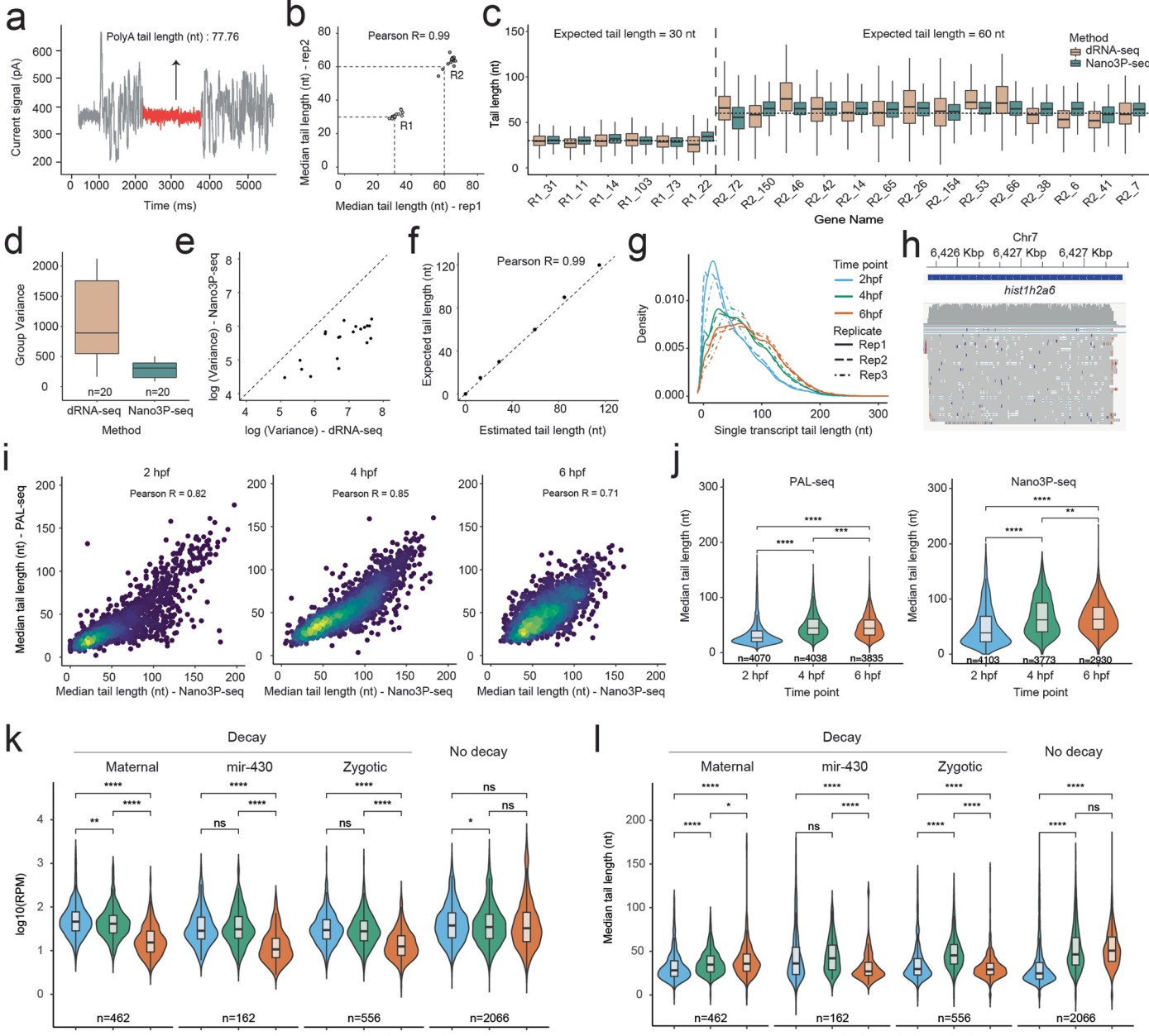

**Extended Data Fig. 3 | Analysis of polyA tail lengths using Nano3P-seq. (a)** Current intensity (pA) plot of a synthetic polyA+ read, obtained using Nano3P-seq. The homopolymeric poly(T) region is highlighted in orange. **(b)** Replicability of median per-gene polyA tail length estimation in sequins captured with Nano3P-seq. The polyA tail length of synthetic sequins is 30 nt (R1 sequins) or 60 nt (R2 sequins). **(c)** Overall comparison of polyA tail length estimation of R1 and R2 sequins which contain 30 nt and 60 nt polyA tail lengths, respectively, was obtained using dRNA-seq (orange) and Nano3P-seq (green). Genes with coverage greater than 30 reads are included. Data are presented as mean values+/− standard error (Please see Table S4 for exact median tail length estimations and read counts for each sequin gene). **(d)** Per-gene variance of polyA tail length estimations of sequins obtained using dRNA-seq (orange) and Nano3P-seq (green). The number of reads included in the analysis is shown below each boxplot. **(e)** Scatter plot showing the comparison between per-gene variance of polyA tail length estimations obtained using dRNA-seq and Nano3P-seq. **(f)** Correlation between expected tail length (nt) and estimated tail length (nt) of cDNA standards. **(g)** Distribution of polyA tail lengths in mRNAs across zebrafish developmental stages (2, 4 and 6 hpf, shown in blue, green and red respectively)

in three biological replicates (shown as full lines, dashed lines, and dotted/dashed lines respectively). **(h)** IGV snapshot of reads mapping to *hist1h2a6* mRNA, which lacks polyA tails. **(i)** Scatterplots of median per-gene polyA tail length estimations using Nano3P-seq and PAL-seq from zebrafish mRNAs at 2 hpf (left), 4 hpf (middle) and 6 hpf (right). Each dot represents the median polyA tail length of a given gene. **(j)** Violin plots depicting the distribution of median per-gene polyA tail length estimations during the zebrafish MZT, estimated using PAL-Seq (left) or Nano3P-seq (right). **(k)** Comparative analysis of the abundance (shown as $\log_2$ RPM) of zebrafish mRNAs that have been binned according to their previously annotated decay mode (maternal decay, zygotic activation-dependent decay, miR-430-dependent decay and no decay) during MZT using PAL-seq data. **(l)** Median per-gene polyA tail length estimations of the 4 groups of zebrafish mRNAs during MZT using PAL-seq data. For figures S3 j–l; the number of genes included in the analysis is shown below each violinplot. Boxplot limits are defined by lower (bottom) and upper (top) quartile, while the bar indicates the median and whiskers indicate+/− 1.5X inter-quartile range. Statistical analyses were performed using Kruskal-Wallis test. (p > 0.05:ns, p ≤ 0.05:*, p ≤ 0.01:**, p ≤ 0.001:***, p ≤ 0.0001:****).

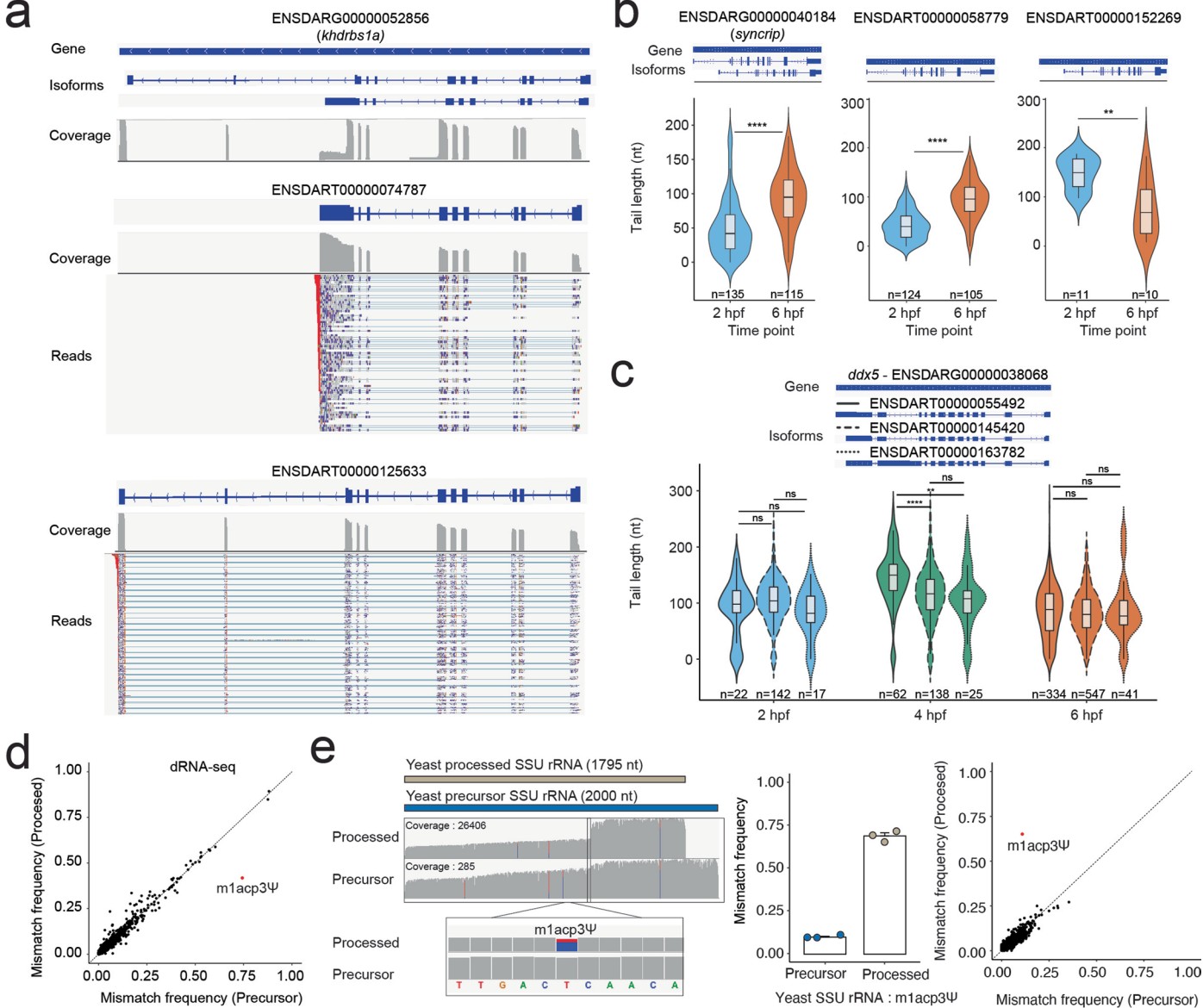

**Extended Data Fig. 4 | Analysis of isoform-specific polyA tail and modification dynamics using Nano3P-seq. (a)** IGV coverage tracks of reads mapping to *khdrbs1a* gene from zebrafish embryos at 2 hpf obtained by Nano3P-seq. Annotations of the gene and two main isoforms are shown at the top of the panels. Individual reads mapping to each isoform is illustrated below each coverage track. PolyA tails of individual reads are shown in red. **(b)** Comparison of polyA tail length distributions of reads mapping to *syncrip*, illustrated at the per-gene (left panel) and per-isoform (middle and right panel) level measured at two time points during the zebrafish MZT. Annotations of the gene and two main isoforms are shown at the top of the panels. **(c)** Comparison of polyA tail length distributions of reads mapping to three distinct isoforms (full, dashed and dotted outline) of *ddx5* measured at the three time points during the zebrafish MZT. Annotations of the gene and three main isoforms are shown at the top of the panels. Only isoforms with more than 10 reads are shown. The number of reads included in the analysis is shown below each violinplot. P-values have been computed using the Kruskal–Wallis test and corrected for multiple testing using the Benjamini–Hochberg procedure (p > 0.05:ns, p ≤ 0.05:*, p ≤ 0.01:**, p ≤ 0.001:***). **(d)** Comparison of the per-site mismatch frequencies observed in reads mapping to yeast precursor SSU rRNA and to yeast processed SSU rRNA sequenced by dRNA-seq, showing that the unique outlier is m$^1$acp$^3$Ψ. **(e)** IGV coverage tracks of reads mapping to yeast processed small subunit (SSU) rRNA (upper track) and precursor SSU rRNA (lower track), including a magnified image at the position known to be modified with m$^1$acp$^3$Ψ (left panel). Positions with a mismatch frequency lower than 0.1 are shown in gray. Mismatch frequency values in yeast precursor and processed SSU rRNA at the position known to be modified with m$^1$acp$^3$Ψ (middle panel) (*n* = 3 biological replicates, data are presented as mean values+/− standard error). Comparison of the per-site mismatch frequencies observed in reads mapping to yeast precursor SSU rRNA and to yeast processed SSU rRNA, showing that the unique outlier is m$^1$acp$^3$Ψ (right panel). Boxplot limits are defined by the lower (bottom) and upper (top) quartile. The bar indicates the median, and whiskers indicate+/− 1.5X interquartile range.

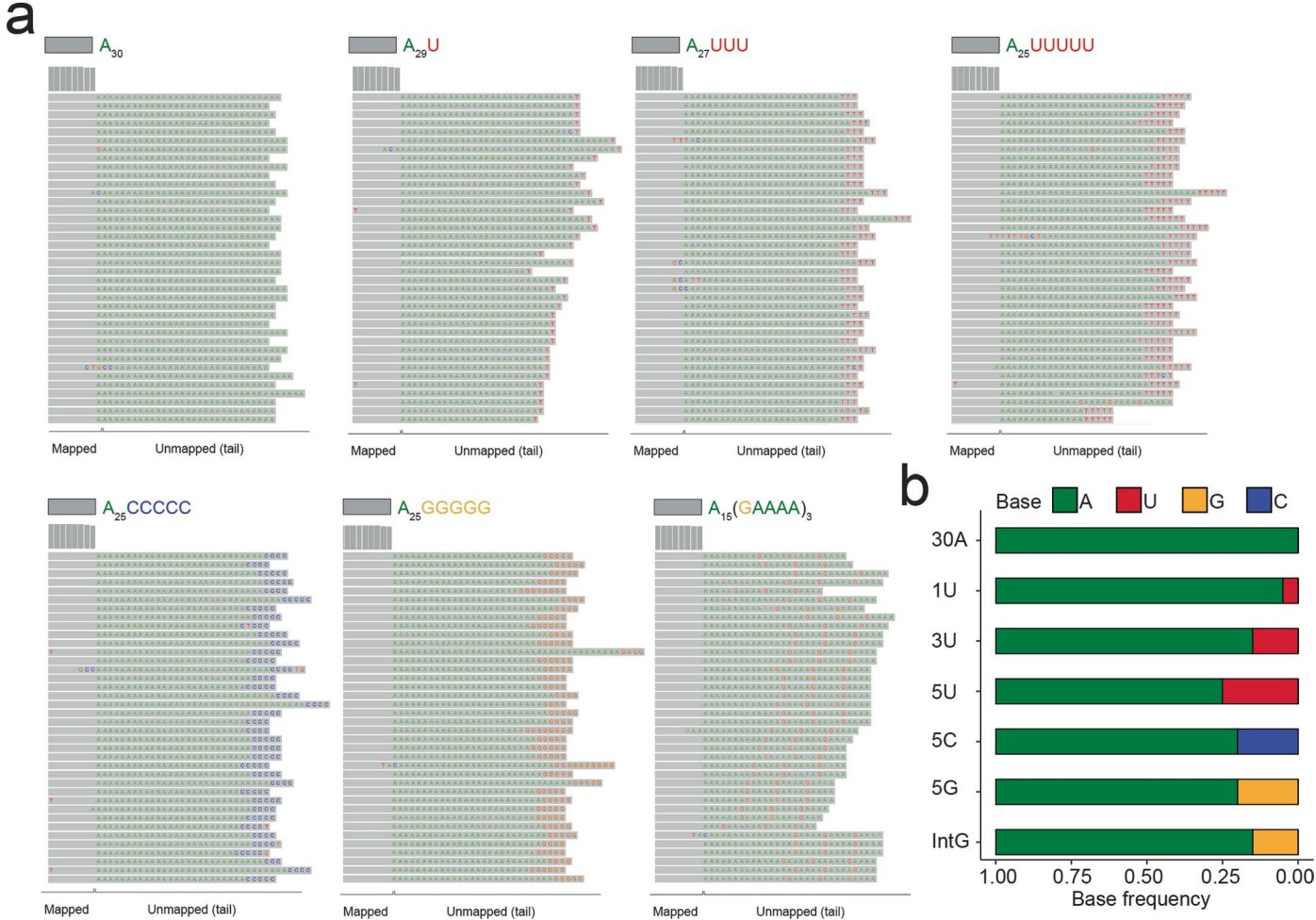

**Extended Data Fig. 5 | Analysis of tail composition using Nano3P-seq in in vitro samples. (a)** IGV snapshots of nucleotide composition in cDNA standard tails sequenced using Nano3P-seq. Gray regions indicate the mapped part of the reads, whereas colored letters indicate soft-clipped bases (unmapped) which are the base-called tails, after trimming the adapter. **(b)** PolyA tail base frequency distribution (A: green, G: orange, C: blue, U: red) of cDNA standards sequenced with Nano3P-seq.

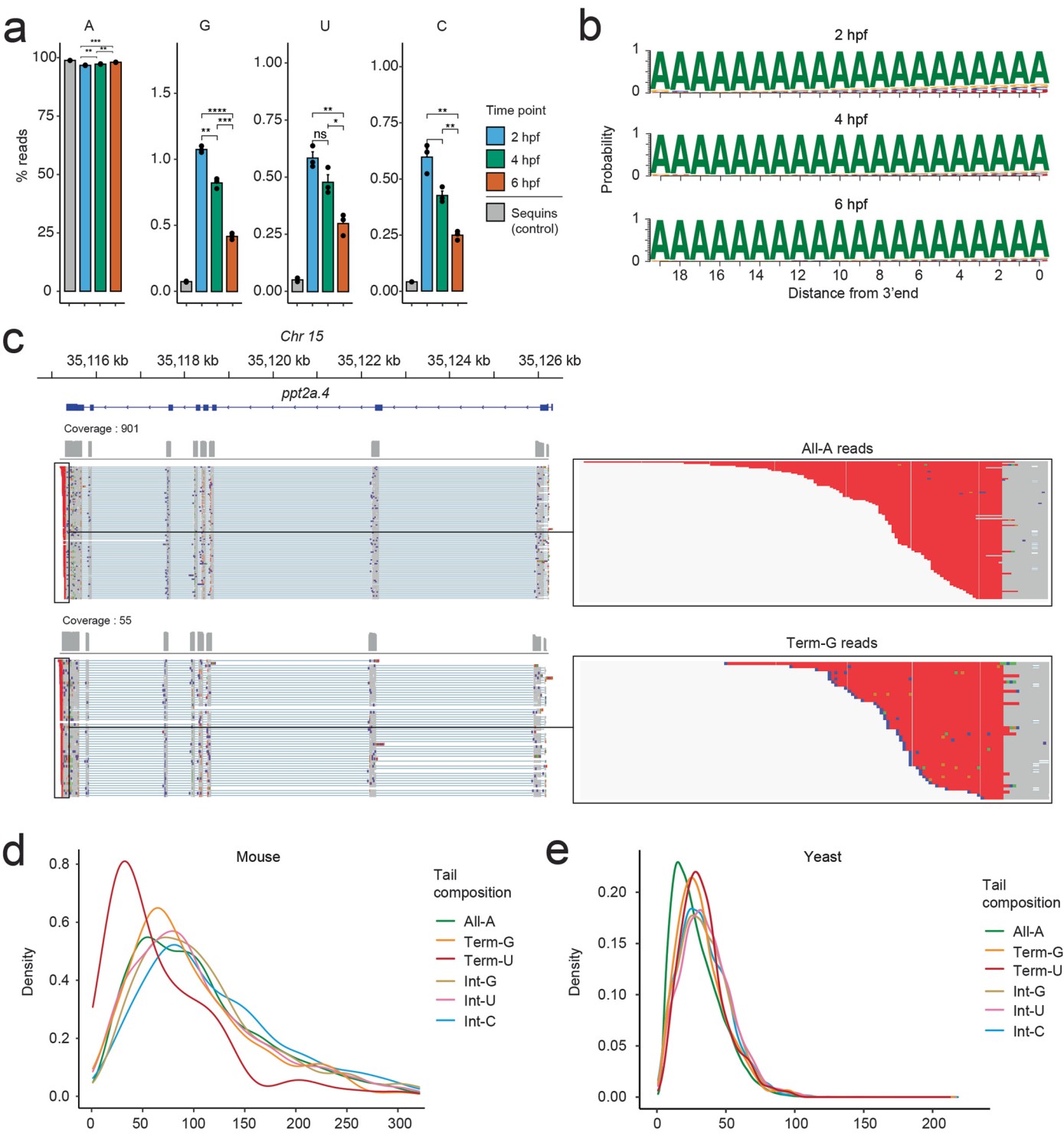

**Extended Data Fig. 6 | Analysis of tail composition using Nano3P-seq in in vivo samples. (a)** Overall nucleotide composition in mRNA tails at 3 time points during the zebrafish MZT (2, 4 and 6 hpf, shown in blue, green and red respectively) and control that includes sequin R1 and R2 groups of RNAs (gray). P-values have been computed using Kruskal-Wallis test. ($n$ = 3 biological replicates, data are presented as mean values+/− standard error). **(b)** Probability of base composition (A: green, G: orange, C: blue and U: red) per position in the last 20 nucleotide of the mRNA tails at 3 time points during the zebrafish MZT (2, 4 and 6 hpf). **(c)** IGV snapshots of reads mapping to zebrafish *ppt2a.4* mRNA (left panel). In the right panel, zoomed images of 3' ends of individual reads with different terminal bases (all reads: top, Term-G reads: bottom) are shown. **(d)** PolyA tail length estimation distributions of mRNA reads belonging to groups classified based on their polyA tail base composition in mouse. **(e)** PolyA tail length estimation distributions of mRNA reads belonging to groups classified based on their polyA tail base composition in yeast. (p > 0.05:ns, p ≤ 0.05:*, p ≤ 0.01:**, p ≤ 0.001:***, p ≤ 0.0001:****).

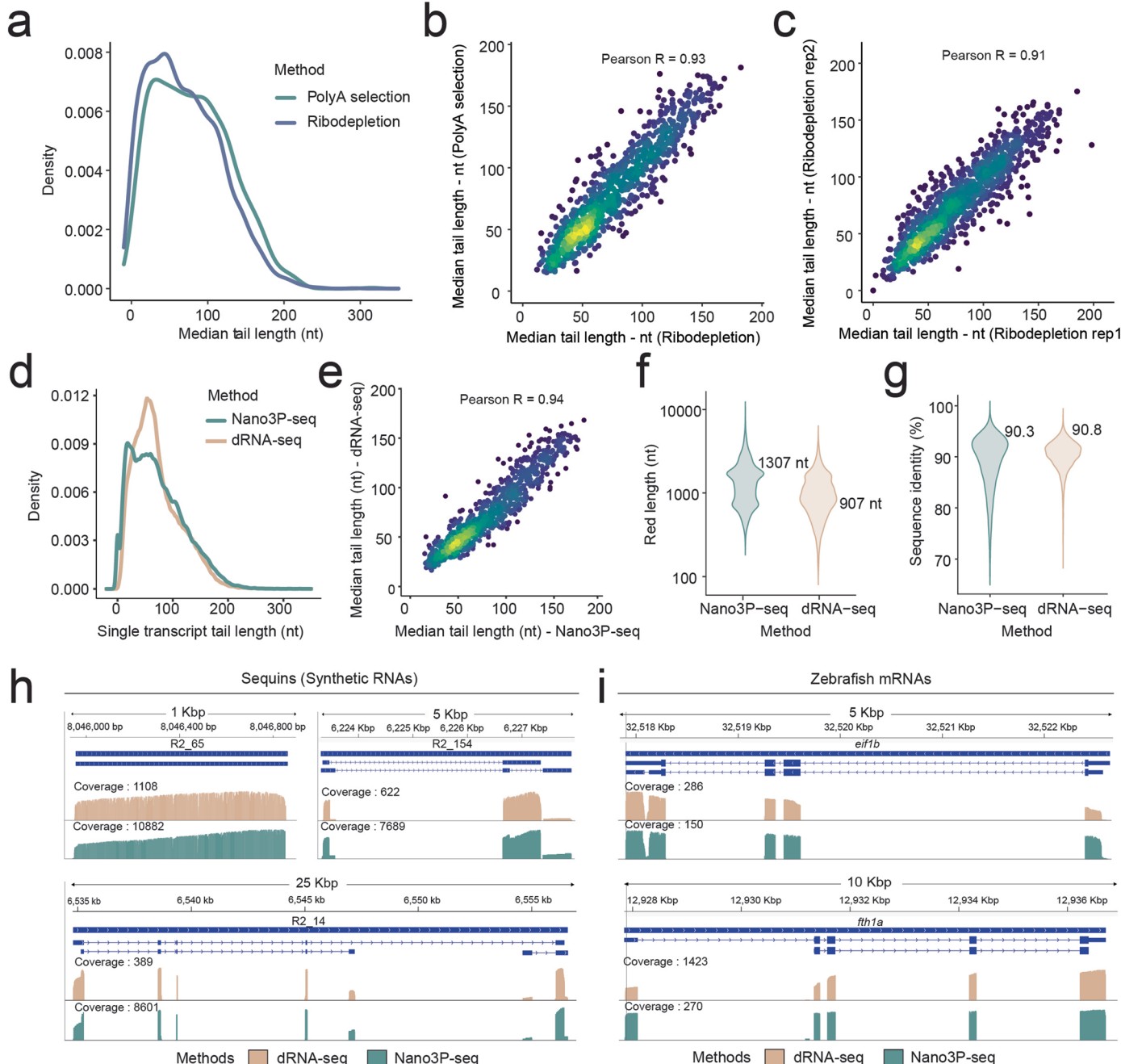

**Extended Data Fig. 7 | Comparison of polyA tail length estimations and run stats between dRNA-seq and Nano3P-seq. (a)** Distribution of median per-gene polyA tail length estimations from 4 hpf zebrafish embryos, isolated using either polyA selection (green) or ribodepletion (blue). **(b)** Comparison of median per-gene polyA tail length estimations between polyA-selected and ribodepleted zebrafish mRNAs isolated at 4 hpf. Each dot represents a gene. **(c)** Comparison of median per-gene polyA tail length estimations of mRNAs in zebrafish ribodepleted samples (replicate 1 and 2) isolated at 4 hpf. Each dot represents a gene. **(d)** Distribution of polyA tail length estimations in mRNAs from 4 hpf zebrafish embryos isolated using polyA selection and sequenced with dRNA-seq (orange) or Nano3P-seq (green). **(e)** Comparison of median per-gene polyA tail length estimations of polyA-selected mRNAs isolated at 4 hpf with dRNA-seq or Nano3P-seq. Each dot represents a gene. **(f)** Read length distribution of mapped reads from 4 hpf zebrafish embryos isolated using polyA selection and sequenced with Nano3P-seq (green) and dRNA-seq (orange), with median lengths of 1307 nt and 907 nt, respectively. **(g)** Sequence identity (%) of the reads from 4 hpf zebrafish embryos isolated using polyA selection and sequenced with either Nano3P-seq (green) or dRNA-seq (orange) with median values of 90.3 and 90.8, respectively. The reads were mapped in both cases to *D. rerio* GRCz11 reference. **(h)** IGV coverage tracks of reads mapping to R2_65, R2_154, and R2_14 from synthetic 'sequin' RNAs, obtained using either dRNA-seq (orange) and Nano3P-seq (green). Annotations of the gene and isoforms are shown at the top of each panel. **(i)** IGV coverage tracks of reads mapping to genes *eif3b* and *fth1a* from zebrafish embryos at 4 hpf, obtained using either dRNA-seq (orange) or Nano3P-seq (green). Annotations of the gene and isoforms are shown at the top of each panel.

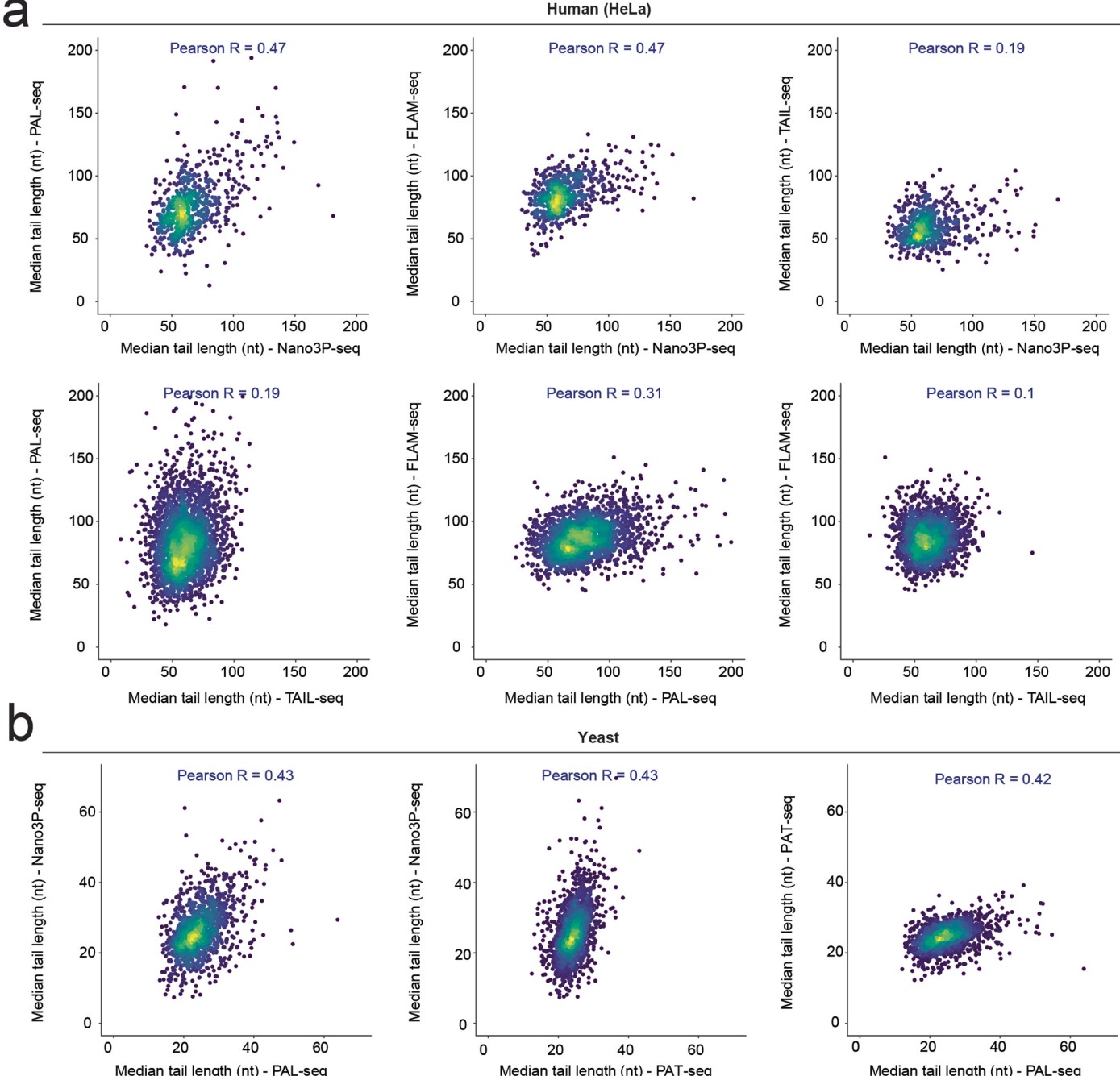

**Extended Data Fig. 8 | PolyA tail length estimation comparisons between different methods. (a)** Scatterplots of median per-gene polyA tail length estimations from HeLa mRNAs using four different methods: Nano3P-seq, PAL-seq (data from Subtelny et al.,[10]), FLAM-seq (data from Legnini et al.,[45]), and TAIL-seq (data from Lim et al.,[43]). **(b)** Scatterplots of median per-gene polyA tail length estimations from *S. cerevisiae* mRNAs using three different methods: Nano3P-seq, PAL-seq (data from Subtelny et al.,[10]), and PAT-seq (data from Harrison et al.,[47]). Each dot represents the median polyA tail length of a given gene.

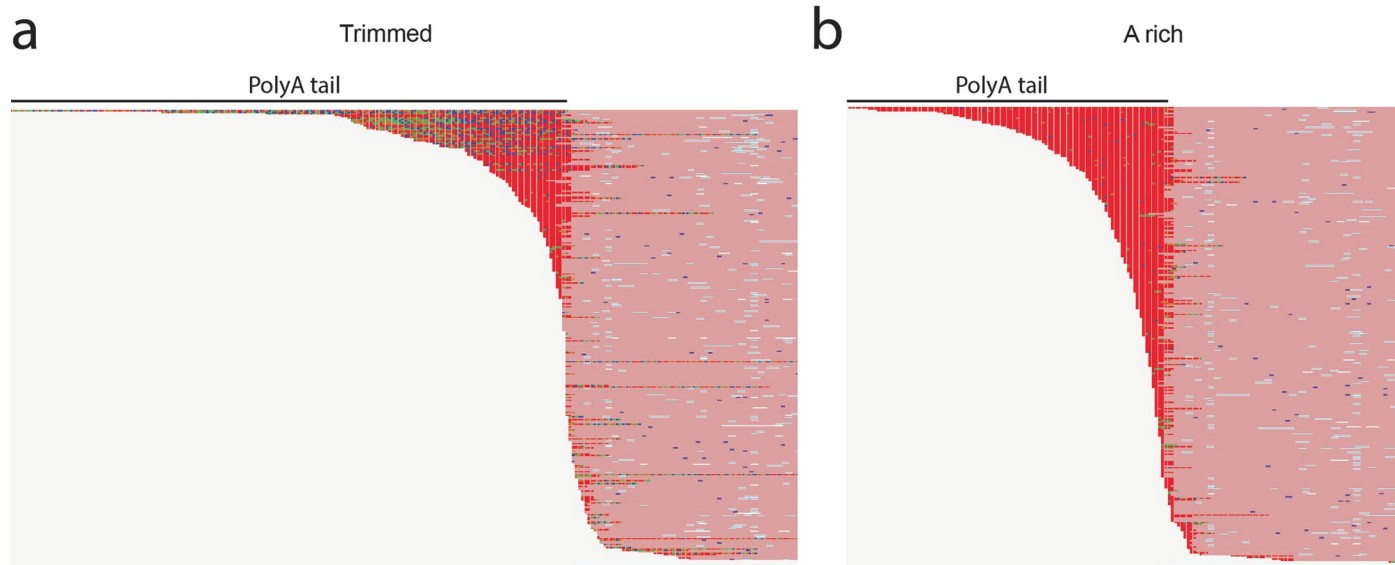

**Extended Data Fig. 9 | Comparison of read ends mapping to *actb1* gene before and after filtering by adenine base (A) enrichment. (a)** Reads that are trimmed with porechop. **(b)** Same reads after removing incorrectly trimmed ones (filtered based on their A content). The labeled part indicates the polyA tail, which is dominantly colored in red.

# Reporting Summary

## Statistics

For all statistical analyses, confirm that the following items are present in the figure legend, table legend, main text, or Methods section.

| n/a | Confirmed | |
|---|---|---|
| ☐ | ☒ | The exact sample size (*n*) for each experimental group/condition, given as a discrete number and unit of measurement |
| ☐ | ☒ | A statement on whether measurements were taken from distinct samples or whether the same sample was measured repeatedly |
| ☐ | ☒ | The statistical test(s) used AND whether they are one- or two-sided *Only common tests should be described solely by name; describe more complex techniques in the Methods section.* |
| ☐ | ☒ | A description of all covariates tested |
| ☐ | ☒ | A description of any assumptions or corrections, such as tests of normality and adjustment for multiple comparisons |
| ☐ | ☒ | A full description of the statistical parameters including central tendency (e.g. means) or other basic estimates (e.g. regression coefficient) AND variation (e.g. standard deviation) or associated estimates of uncertainty (e.g. confidence intervals) |
| ☐ | ☒ | For null hypothesis testing, the test statistic (e.g. *F*, *t*, *r*) with confidence intervals, effect sizes, degrees of freedom and *P* value noted *Give P values as exact values whenever suitable.* |
| ☒ | ☐ | For Bayesian analysis, information on the choice of priors and Markov chain Monte Carlo settings |
| ☒ | ☐ | For hierarchical and complex designs, identification of the appropriate level for tests and full reporting of outcomes |
| ☐ | ☒ | Estimates of effect sizes (e.g. Cohen's *d*, Pearson's *r*), indicating how they were calculated |

*Our web collection on statistics for biologists contains articles on many of the points above.*

## Software and code

Policy information about availability of computer code

| Data collection | Nanopore data was acquired using Minknow software v.3.5.5 . |
|---|---|
| Data analysis | Nanopore data was basecalled using Guppy version 4.0. TailfindR tool (Nano3P-seq mode) was used for tail length estimation. Minimap2 and BWA tools were used for mapping the basecalled nanopore reads. Code used in this study is made available in the following link : https://github.com/novoalab/Nano3P_Seq. TailfindR nano3pseq mode can be accessed using this link : https://github.com/adnaniazi/tailfindr/tree/nano3p-seq. |

For manuscripts utilizing custom algorithms or software that are central to the research but not yet described in published literature, software must be made available to editors and reviewers. We strongly encourage code deposition in a community repository (e.g. GitHub). See the Nature Portfolio guidelines for submitting code & software for further information.

## Data

Policy information about availability of data

All manuscripts must include a data availability statement. This statement should provide the following information, where applicable:
- Accession codes, unique identifiers, or web links for publicly available datasets
- A description of any restrictions on data availability
- For clinical datasets or third party data, please ensure that the statement adheres to our policy

Base-called FAST5 reads from Nano3P-seq and dRNA-seq libraries are publicly available in ENA, under accession code PRJEB53494. PAL-Seq polyA tail length

# Human research participants

Policy information about studies involving human research participants and Sex and Gender in Research.

| | |
|---|---|
| Reporting on sex and gender | *Use the terms sex (biological attribute) and gender (shaped by social and cultural circumstances) carefully in order to avoid confusing both terms. Indicate if findings apply to only one sex or gender; describe whether sex and gender were considered in study design whether sex and/or gender was determined based on self-reporting or assigned and methods used. Provide in the source data disaggregated sex and gender data where this information has been collected, and consent has been obtained for sharing of individual-level data; provide overall numbers in this Reporting Summary. Please state if this information has not been collected. Report sex- and gender-based analyses where performed, justify reasons for lack of sex- and gender-based analysis.* |
| Population characteristics | *Describe the covariate-relevant population characteristics of the human research participants (e.g. age, genotypic information, past and current diagnosis and treatment categories). If you filled out the behavioural & social sciences study design questions and have nothing to add here, write "See above."* |
| Recruitment | *Describe how participants were recruited. Outline any potential self-selection bias or other biases that may be present and how these are likely to impact results.* |
| Ethics oversight | *Identify the organization(s) that approved the study protocol.* |

Note that full information on the approval of the study protocol must also be provided in the manuscript.

# Field-specific reporting

Please select the one below that is the best fit for your research. If you are not sure, read the appropriate sections before making your selection.

☒ Life sciences ☐ Behavioural & social sciences ☐ Ecological, evolutionary & environmental sciences

For a reference copy of the document with all sections, see nature.com/documents/nr-reporting-summary-flat.pdf

# Life sciences study design

All studies must disclose on these points even when the disclosure is negative.

| | |
|---|---|
| Sample size | Sample size was chosen as n=3 for most experiments, allowing to perform statistical tests to assess significance in the observed differences between groups. Sample size in this case corresponds to biological replicates |
| Data exclusions | No data was excluded from our analyses. |
| Replication | In-vitro constructs (RNA and cDNA) were sequenced once. S. cerevisiae data has three biological replicates. Mus musculus data has two biological replicates. Danio rerio has three replicates for three time-points. HeLa (Human cell line) has two biological replicates. |
| Randomization | Randomization is not relevant to our study. |
| Blinding | Blinding was not relevant to our study. |

# Reporting for specific materials, systems and methods

We require information from authors about some types of materials, experimental systems and methods used in many studies. Here, indicate whether each material, system or method listed is relevant to your study. If you are not sure if a list item applies to your research, read the appropriate section before selecting a response.

## Materials & experimental systems

| n/a | Involved in the study |
|-----|----------------------|
| ☒ ☐ | Antibodies |
| ☐ ☒ | Eukaryotic cell lines |
| ☒ ☐ | Palaeontology and archaeology |
| ☐ ☒ | Animals and other organisms |
| ☒ ☐ | Clinical data |
| ☒ ☐ | Dual use research of concern |

## Methods

| n/a | Involved in the study |
|-----|----------------------|
| ☒ ☐ | ChIP-seq |
| ☒ ☐ | Flow cytometry |
| ☒ ☐ | MRI-based neuroimaging |

## Eukaryotic cell lines

Policy information about cell lines and Sex and Gender in Research

| | |
|---|---|
| Cell line source(s) | HeLa-CCL-2 was purchased from ATCC. |
| Authentication | None of the cell lines were authenticated. |
| Mycoplasma contamination | Cell lines were tested for mycoplasma contamination. |
| Commonly misidentified lines (See ICLAC register) | No commonly misidentified samples were used in this study. |

## Animals and other research organisms

Policy information about studies involving animals; ARRIVE guidelines recommended for reporting animal research, and Sex and Gender in Research

| | |
|---|---|
| Laboratory animals | Mus musculus and Danio rerio were used in this study. Experiments were performed with male mice aged between 8 and 10 weeks. All mice were euthanized using CO2 and tissues were snap frozen in liquid nitrogen. Animals were kept on a 12:12h light:dark cycle and provided with water and food ad libitum. Wild-type zebrafish (Danio rerio) embryos were obtained through natural mating of the TU-AB strain of mixed ages (5–18 months). Mating pairs were randomly chosen from a pool of 60 males and 60 females allocated for each day of the month. Embryos and adult fish were maintained at 28°C. |
| Wild animals | No wild animals were used in this study. |
| Reporting on sex | Male mouse were used in this study. |
| Field-collected samples | No field collected samples were used in this study. |
| Ethics oversight | Fish lines were maintained according to the International Association for Assessment and Accreditation of Laboratory Animal Care research guidelines, and protocols were approved by the Yale University Institutional Animal Care and Use Committee (IACUC). Mice maintenance was approved by the Garvan/St Vincent's Hospital Animal Ethics Committee, in accordance with the guidelines of the Australian Code of Practice for the Care and Use of Animals for Scientific Purposes (Project No. 16/14 and 16/26). |

Note that full information on the approval of the study protocol must also be provided in the manuscript.

