## [Peer Review File · Nature Methods]

Peer Review Information

Manuscript Title: Nano3P-seq: transcriptome-wide analysis of gene expression and tail dynamics using end-capture nanopore cDNA sequencing

Corresponding author name: Eva Maria Novoa

Editorial Notes: n/a

Reviewer Comments & Decisions:

Decision Letter, initial version:

Dear Eva,

Your Article entitled "Nano3P-seq: transcriptome-wide analysis of gene expression and tail dynamics using end-capture nanopore sequencing" has now been seen by 3 reviewers, whose comments are attached. While they find your work of potential interest, they have raised serious concerns which in our view are sufficiently important that they preclude publication of the work in Nature Methods, at least in its present form.

As you will see, the reviewers raise serious concerns about the performance evaluation of Nano3P-seq itself as well as its advance over existing tools.

Should further experimental data allow you to fully address these criticisms we would be willing to look at a revised manuscript (unless, of course, something similar has by then been accepted at Nature Methods or appeared elsewhere). This includes submission or publication of a portion of this work somewhere else. We hope you understand that until we have read the revised paper in its entirety we cannot promise that it will be sent back for peer-review.

If you are interested in revising this manuscript for submission to Nature Methods in the future, please contact me to discuss your appeal before making any revisions. Otherwise, we hope that you find the reviewers' comments helpful when preparing your paper for submission elsewhere.

Best regards,
Lei

Lei Tang, Ph.D.
Senior Editor
Nature Methods

Reviewers' Comments:

Reviewer #1:

Remarks to the Author:

Begik et al present the Nano3P-Seq method. Nano3P-Seq uses adapters and a template-switch reaction. It makes cDNAs with gene body and entire poly(A) tail. Importantly, it can equally analyze PolyA(-) RNAs. The authors apply Nano3P-Seq to the maternal-to-zygotic transition (MZT) and compare mouse, zebrafish and yeast transcriptomes. They find increased non-coding RNAs as the MZT progresses. They also see longer poly(A) tails during the MZT and show differences between different decay modes. Furthermore, poly(A) lengths also distinguish isoforms in one gene. Mismatch frequencies correlate very highly between pre-rRNA and mature rRNA in yeast and mouse. M1acp3², however is more frequent in mature rRNA. Finally, authors show that RNAs with polyuridylated tail ends have different tail lengths.

Nano3p-Seq has conceptual advances over published methods (e.g., FLAM-Seq). However, extensive experimental work is needed to prove that all findings are correct. Moreover, parts of the work lack clarity.

Major comments:

1. "Indeed, per-read analysis of mitochondrial rRNA reads revealed that a significant proportion of 16S mitochondrial rRNA contained a polyA tail": Is Nano3P-Seq accurately quantifying both versions of 16S mitochondrial rRNA? Experimental validation is needed here.
2. Figure 1B is central but hard to interpret. Y-axis ("Counts") and legend ("Nano3P-seq captures a wide range of RNA biotypes in the mouse brain") lack clarity. Are these gene counts or read counts? What is exact definition? They seem low for mRNAs (<2000?), which raises doubts. Similar issues affect Fig 2D.
3. In Figure 2, the authors use ribo depletion ["we isolated total RNAs from zebrafish embryos at 2, 4 and 6 hours post-fertilization (hpf) in biological duplicates, ribo-depleted the samples, and sequenced them using the Nano3P-seq protocol"]. Then they find similar rRNA and mRNA abundance (Figure 2D). Did ribo depletion not work? Is a different sample used here? Please clarify.

4. There is interesting biology in Figure 2D: “the abundance of non-coding RNA populations, including miscRNAs, scaRNAs and snoRNAs, increased as the MZT progressed”, but p values are not indicated. Please give p-values and statistical details.

5. Figure 3B, for the 30nt-tail sequences, dRNAseq seems to have medians closer to the expected line than Nano3P-Seq (e.g., for R1_14, R1_103 & R1_73). For some others Nano3P-Seq seems to be better. Please present details in text and improve figure for easy visual assessment.

6. The manuscript focuses on zebrafish, but analysis of pre- and mature rRNA only uses mouse and yeast. What is the reason? Please, compare across >2 species with identical experiments.

7. The authors determine m1acp3² levels in rRNA and pre-rRNA, which is “in agreement with previous observations”.

a) Please define variability of mismatch frequency across ≥ 3 replicates.

b) Please validate the m1acp3² estimate with a different method.

8. U, UU, UUU and UUUU 3'ends are extremely rare. There is little proof that these are not sequencing errors or other artifacts.

a) Please use spike-in sequences with A, U, UU, UUU and UUUU ends. Process these in separate runs and define recall and precision.

b) Please compare frequency of U runs (length=1,2,3,4) to G and C runs.

c) Please prove that imperfect adapter sequences do not contribute nonA nucleotides

Minor Comments:

- Depiction of poly(A) tails in Figure 1c, S1c is suboptimal. In S1c tail lengths and frequency are difficult to see.

- For figure 1C the authors say “In addition, our results confirmed that polyA tail length information was retained in individual reads. Specifically, the majority of reads corresponding to mRNAs had polyA tails”. 1C suggests that it is barely a majority: many reads have no green PolyA. This example plants more doubts than it helps the reader.

- Figure 1D lacks detail. Is correlation given before or after log-transformation?

- Fig 2B: What are the correlations between the time points

- Figure 3a: The material used here is not obvious. Is it the same as in Figures S1a-c?

- References 27 and 28 seem identical

- Figure 4a only shows one example of differences conserved across time points, but the main text says “often conserved across the different time points analyzed (Fig 4a)”

- Gene and isoform percentages (5.2% and 11.7%) in the text for Figure 4 require confidence intervals and exact numbers.
- Figure 5D is barely mentioned in the text

Reviewer #2:

Remarks to the Author:

SUMMARY

This paper presents a strategy for analyzing RNA expression using nanopore sequencing of first-strand cDNA copies primed by strand switching. A strength is detection and length estimates of 3' homopolymers, and a relatively straightforward library preparation scheme. The paper is casually written and not ready for careful consideration in its current form. Some essential data are missing, and some of the data are not presented in a transparent manner. This is surprising given the substantial experience of some of the co-authors. In the text that follows we highlight key technical and logical issues, and then we note specific issues line-by-line.

GENERAL COMMENTS

Title and Abstract

*It would be informative to include 'cDNA' in the title and abstract. It was unclear to us until well into the introduction that the new technique was based on first strand cDNA rather than native RNA.

*Here and elsewhere the authors use the term 'translatability'. Possibly this is now used in the molecular biology community, but a more traditional term is 'translational efficiency'. In either case, for a general audience it would be useful to precisely define what you mean.

*In the last sentence of the abstract, the authors assert that Nano3P-seq can accurately estimate transcript levels. It is unclear if this is at the gene level or at the isoform level.

In the last sentence of the abstract, the authors assert that Nano3P-seq can accurately estimate tail lengths of full-length transcripts. As far as we can tell, Nano3P-seq does not establish unequivocally that reads are full-length.

Results

*The current draft does not adequately address performance of the Nano3P-seq strategy. Most notably, in the main text there are no data for throughput, and as best we can tell, there are no data for base call accuracy or read length distribution anywhere in the manuscript. These data are important for a fair comparison between direct RNA sequencing and Nano3P-seq (see Discussion section comments below).

It is known that cDNA basecall accuracy is typically better than direct RNA basecall accuracy. So why not point this out in the text along with some data?

*The read count log plots are not up to NMeth standards and will be difficult for readers to assess. These include panels Fig 1.D, Fig 2.B, and Fig S1.H. Let's examine Figure 1.D as an example:

i) The X axis is labelled 'Log (Expected Counts)'. What is meant by expected counts in this context and how were these values derived? This is not defined in the panel, text, or Methods as far as we could see. Why not use the X-axis units as in the original Sequins manuscript, i.e. Log(10) Attomoles/uL? It is noteworthy that several of the authors on this paper were co-authors on the carefully-executed Sequin (2016) paper.

ii) It is unclear to us if the data in this panel are for genes or isoforms. In the Sequins(2016) paper, the authors presented Counts vs Attomoles/uL for genes, isoforms, and exons. This trio of plots would be a useful addition to the Nano3P-seq paper.

iii) To facilitate the reader's examination of these data, please plot untransformed data on Log(10)-Log(10) plots with tick marks. Once again, the original Sequins manuscript can serve as a guide.

iv) Please include the slope value for the fitted line. A naive examination of the original Fig 1D might suggest a 1-to-1 correspondence between expected and observed values which appears not to be the case.

v) In the Sequins (2016) paper, the relationship between counts and concentration was flat below 1 attomole/uL. The paper under review does not use concentration on the X axis so it is difficult to make a comparison. We are curious if any of the Sequin RNA isoforms were not observed in these control experiments.

vi) It is customary to use axis values beyond the upper and lower limits of plotted data. Please fix this.

vii) In Figure 2B the scales on the plots do not bound the data. Fix this. Also, what was the logic for using natural log? Was this the best way to convey the information to a reader – is that the explanation? Also, the highest density in the distribution is not bound by the axes. Please fix this.

*Line 209: section title. The authors claim that tail length can be accurately estimated using Nano3P-seq. Please explain to the reader what sort of precision they should anticipate and whether or not that precision is tail length dependent. Also, in Figure S3B, there appears to be a reasonable correlation between the two replicates in the range of 30-60 nt, but that they both overestimate tail lengths at 30 and 60 nucleotides. This is not a big issue, but please be forthright about what the technique delivers.

*Figure 3G: Are the values for transformed or untransformed data?

*The paper claims isoform documentation. However, Nano3P-seq does not unambiguously establish that the biological 5 prime end of transcripts have been sequenced. Please defend this position. Related (as pointed out above) there is no documentation of the read length distribution for this method.

DETAILED COMMENTS

Line 73. Try making the sentence more clear by bringing the subject and verb together.

Line 96. Others have extended the 3 prime ends of RNA strands using poly(I) followed by dRNA sequencing. Please see Vo et al RNA 2021. 27: 1497-1511) and Drexler et al Molecular Cell Volume 77: 985-998 (2020).

Lines 101-104. Please reconcile these assertions with the preceding comment.

Line 121. Please revisit this statement in light of Vo et al. above.

Line 153. Interpreting data in figure legends (as was done here) is usually discouraged. We are not sure of NMeth's policy on this.

Line 163. Please see discussion of Figure 1D in the general comments section.

Line 177. 'Drastic' is an odd word choice here. 'Drastic' measures are not just extreme, they are likely to have harmful side-effects. Better to use a neutral word.

Line 191. Polyadenylation of Mt 16s rRNA is also observed in human samples (see citation 12 in this draft paper).

Line 220. Here the fit is r^2 , but in Figure S3B it is r . Which is it? This lack of care in data presentation is concerning.

Line 258 (E). Awkward sentence.

Line 289. Here the advantage of nanopore read lengths is highlighted, but nowhere do the authors document the read length distribution of their technique. This needs to be included.

Line 310. The authors make a general statement ‘.....it demonstrates that Nano3P-seq can provide transcriptome-wide measurements of the polyadenylation status of diverse biological samples’. This is based on a few examples. While this may one day prove to be the case, claiming transcriptome-wide measurements is premature based on this paper.

Line 401. Not an accurate statement. See Vo paper and Drexler paper cited above.

Reviewer #3:

Remarks to the Author:

This manuscript reports the development of a new method, Nano3P-seq, which uses Oxford Nanopore sequencing to provide information on RNA expression level, poly(A) tail composition, and poly(A) tail length. This does not rely on PCR and can also capture non-poly(A) transcripts. The authors apply their new method to some relevant samples and confirm findings generally known about poly(A) tails. This application part is well done and a strength of the manuscript. More generally, there are many aspects of biology where this method could be used. Unfortunately, this manuscript falls short in two related areas – (1) how accurate is the method for poly(A) composition and length and (2) how does it compare directly to other methods for measuring poly(A) tails.

Major concerns.

1) The manuscript needs to be more rigorous in showing that Nano3P-seq is accurate in measuring poly(A) tail composition and length.

Figure S1 shows that poly(A) tail length can be measured but given that the initial tail length was not carefully characterized, it is not possible to assess accuracy in the sequence data. In Figures 3B, S3C, and S3D, the accuracy of the tail length is measured for the Sequins with known lengths of 30 and 60 bases. That is a good start, but it’s important to test a wider range of lengths. Moreover, the authors mention that the measured length with tailfindR was off by 15 bases and they adjusted all measurements accordingly (lines 646-652). It was not adequately explained how they know that this is due to the expectation of a double-stranded cDNA by tailfindR. Why would this be true for both 30 and 60 base tails?

For poly(A) tail composition, the manuscript does not present any data on the error rate for these sequencing runs, the base quality in the poly(A) tail, nor the accuracy of the bases with spike-ins of known composition. Even without PCR, it’s known that reverse transcription and nanopore sequencing produce errors. Error rates should be shown both for base substitutions and insertion/deletions. The authors focus on terminal U residues (Figures 5C and 5D), but also show other bases are present in the last 10 residues (Figure S7A). They should also explore bases along the entire length of the poly(A) tail with known controls to see whether non-A bases in other positions can be accurately measured and studied.

2) One of the most important aspects of a manuscript introducing a new method is to demonstrate how it compares to existing technologies. In the Introduction, the authors mention that PAL-seq and TAIL-seq are limited because they use Illumina short reads and rely on PCR. There, they also mention the dRNA-seq nanopore method is not able to sequence non-poly(A) RNA and their tails. Note that dRNA-seq does not rely on PCR. In the Discussion, the authors mention PacBio-based methods, FLAM-seq and PAIso-seq, which provide all the information that Nano3P provides for poly(A)+ transcripts, but with a PCR amplification step. The reads are more expensive, though this may not be a major factor in such experiments and details are not provided. While the PacBio sequencers are expensive, the authors should mention that there are many facilities that will sequence samples submitted by any scientist. If Nano3P-seq provides a meaningful advance for the field, there should be a direct comparison with one of the best Illumina and one of the best PacBio methods with aliquots from the exact same sample. The authors do compare to dRNA-seq in Figures 2F, 3B, S3C, S3D. Additionally, they compare PAL-seq to Nano3P-seq in Figure S4A and S4B, but these are published data derived from a different sample for PAL-seq. There are clearly differences too, so that it is not clear whether they are meaningful, nor which method is more accurate. If there are PCR biases that are not present in this new method, the authors need to show that. And the best way to do this is with a set of spike-in controls, including the Sequins, but also ones with different known poly(A) tail lengths and compositions (as in the PAIso-seq paper, ref. 44, but not necessarily limited to what is done in that paper).

Minor concerns.

- 1) The authors should mention how the input RNA amount for this protocol compares to other protocols because this can be a key experimental design feature. In the Methods section, they start with 100 ng of total RNA, but what is the lower limit for this method? This could be an issue given that this method does not use PCR.
- 2) While the authors show nicely that the RNA expression of the Sequins is accurate, what about looking at biases such as 5' vs. 3' ends of the transcript, GC content, and transcript length?
- 3) Line 180-181: What are miscRNAs and scaRNAs?
- 4) Lines 185-191: Although the authors found poly(A) tails on 16S mitochondrial rRNA in both zebrafish and mouse, the conservation of this finding does not fully validate it as correct. The text could be modified to clarify this point.
- 5) Box plot features should be explained in Figures 3B,D,G,H and elsewhere.
- 6) Lines 296-297: What is the justification for a threshold of 10 reads per isoform?
- 7) Line 343: Perhaps change to "modification" (singular) as there is only one type assessed here.
- 8) Line 407: TGIRT should be spelled out and the explanation for why it is better should be clarified.
- 9) Lines 426-439: Perhaps this should be in the Result section?
- 10) Line 454: If the authors state their method is "low cost," they should present information on the cost of reagents for all steps in this method.
- 11) Experimental details are missing in the Materials and Methods section.
 - a. Line 481: Sodium Acetate concentration?

- b. Line 485: Turbo DNase details (concentration, time, temperature) missing.
- c. Line 489: How were the mice cared for (feeding, day/night schedule, etc.)? Were the mice male or female?
- d. Line 493: Details about “Protease Inhibitor” missing.
- e. Line 494: Details about “dounce” (# of times, size) missing.
- f. Line 503: Details about “RNase Inhibitor” missing.
- g. Line 521: Details about which “oligo (dT) magnetic beads” missing.
- h. Line 532: What is “bead resuspension”? Is this from the Ribodepletion kit or the Dynabeads kit?
- i. Line 533: What is “bead resuspension buffer”? Same as above.
- j. Line 536: What is “bead wash buffer”? Same as above.
- k. Line 537: What is “depletion buffer”? Same as above.
- l. Lines 570, 572, 573: What is the source of the buffers, Adapter Mix, ABB Buffer, Elution Buffer, Sequencing Buffer, and Loading Buffer? All from Oxford Nanopore?
- m. Line 574: Please provide details on the length of the runs and whether any library reloading was done.
- n. Line 663: What about the care of the mice?
- o. Figure S1: Perhaps include an explanation for the term “curlcake” and refer the reader to the Materials and Methods for details on CC1 and CC2.

Author Rebuttal to Initial comments

Reviewers' Comments:

Reviewer #1:

Remarks to the Author:

Begik et al present the Nano3P-Seq method. Nano3P-Seq uses adapters and a template-switch reaction. It makes cDNAs with gene body and entire poly(A) tail. Importantly, it can equally analyze PolyA(-) RNAs. The authors apply Nano3P-Seq to the maternal-to-zygotic transition (MZT) and compare mouse, zebrafish and yeast transcriptomes. They find increased non-coding RNAs as the MZT progresses. They also see longer poly(A) tails during the MZT and show differences between different decay modes. Furthermore, poly(A) lengths also distinguish isoforms in one gene. Mismatch frequencies correlate very highly between pre-rRNA and mature rRNA in yeast and mouse. M1acp3, however is more frequent in mature rRNA. Finally, authors show that RNAs with polyuridylated tail ends have different tail lengths.

Nano3p-Seq has conceptual advances over published methods (e.g., FLAM-Seq). However, extensive experimental work is needed to prove that all findings are correct. Moreover, parts of the work lack clarity.

We thank the reviewer for his/her feedback and time in reviewing our work. We agree with the reviewer that Nano3P-seq has conceptual advances over published methods, and that additional experimental work would be beneficial to further validate and benchmark the Nano3P-seq method. Specifically, in our resubmission, we plan to:

1. Include a third replicate for zebrafish time-course experiments sequenced using Nano3P-seq, which will allow us to provide p-values to assess the significance in the increased representation of certain RNA biotypes during the embryo development
2. Include a third replicate for yeast total RNA samples sequenced using Nano3P-seq, to further confirm the absence of m1acp3Y in pre-rRNA, relative to mature rRNA, and include p-values.
3. In addition to using the sequins to assess the quantitative ability of Nano3P-seq to predict polyA tail lengths (which contain either 30nt or 60nt long tails), we will also sequence a set of synthetic sequences using Nano3P-seq, which will contain a wider range of polyA tails of diverse known lengths (0, 25, 50, 100, 150 and 200 nt) as well as with polyU tails of diverse lengths.
4. In addition to sequencing these new synthetic sequences (point #3) with Nano3P-seq, they will also be sequenced with direct RNA seq

In addition, please see below point-by-point responses to the individual comments.

Major comments:

1. “Indeed, per-read analysis of mitochondrial rRNA reads revealed that a significant proportion of 16S mitochondrial rRNA contained a polyA tail”: Is Nano3P-Seq accurately quantifying both versions of 16S mitochondrial rRNA? Experimental validation is needed here.

We will perform qPCR experiments to assess the relative abundance of 16S mitochondrial rRNA polyadenylated and non-polyadenylated populations. qPCR will also be performed on 12S mitochondrial rRNA, which will act as a control.

2. Figure 1B is central but hard to interpret. Y-axis (“Counts”) and legend (“Nano3P-seq captures a wide range of RNA biotypes in the mouse brain”) lack clarity. Are these gene counts or read counts? What is exact definition? They seem low for mRNAs (<2000?), which raises doubts. Similar issues affect Fig 2D.

These plots were log (normalized counts), not absolute counts. We see that this could be confusing for the reviewers and future readers, so we will now change all plots that were using log(normalized counts) for log(absolute read counts).

3. In Figure 2, the authors use ribo depletion [“we isolated total RNAs from zebrafish embryos at 2, 4 and 6 hours post-fertilization (hpf) in biological duplicates, ribo-depleted the samples, and sequenced them using the Nano3P-seq protocol”]. Then they find similar rRNA and mRNA abundance (Figure 2D). Did ribo depletion not work? Is a different sample used here? Please clarify.

From our experiments, we found that ribodepletion oligonucleotides were not as effective in ribodepleting maternally rRNAs compared to zygotic rRNAs. Total RNA profiles before and after ribodepletion can be found in Figure S2A in our original submission, where it can be seen that 18s maternal rRNA ribodepletion was more effective than 28s maternal rRNA ribodepletion.

We should note, however, that incomplete ribodepletion will not affect Nano3Pseq performance, but it will lead to a higher proportion of rRNAs in the sequenced zebrafish embryo samples.

4. There is interesting biology in Figure 2D: “the abundance of non-coding RNA populations, including miscRNAs, scaRNAs and snoRNAs, increased as the MZT progressed”, but p values are not indicated. Please give p-values and statistical details.

We had not provided p-values to assess the significance of this increase as we had only included biological duplicates (n=2) for each developmental time point in our previous analyses. To address the reviewer’s question, we will now include a third replicate for zebrafish time-course experiments sequenced using Nano3P-seq, which will allow us to provide p-values to assess the significance in the increased representation of certain RNA biotypes during the embryo development.

5. Figure 3B, for the 30nt-tail sequences, dRNAseq seems to have medians closer to the expected line than Nano3P-Seq (e.g., for R1_14, R1_103 & R1_73). For some others Nano3P-Seq seems to be better. Please present details in text and improve figure for easy visual assessment.

We will clarify in the text what proportion of sequins have polyA tail lengths that are better predicted by Nano3Pseq, relative to dRNAseq, as per the reviewer’s suggestion.

6. The manuscript focuses on zebrafish, but analysis of pre- and mature rRNA only uses mouse and yeast. What is the reason? Please, compare across >2 species with identical experiments.

We will clarify in the text that zebrafish embryos do not have de novo transcription of zygotic rRNAs during early embryo stages (Locati et al., 2017: PMID:28500251), explaining why we only included yeast

and mouse samples in the pre-rRNA/mature rRNA comparative analyses, but not zebrafish samples (i.e. we did not find any read mapping to pre-rRNA at t=2,4,6hpf).

In our resubmitted version, we will now include additional replicates of yeast samples, showing that the results are consistent across 3 biological replicates, both in terms of polyadenylation of 16s mitochondrial rRNA (but not 12s) as well as in terms of presence of m1acp3Y in mature rRNA (but not pre-rRNA).

7. The authors determine m1acp3 levels in rRNA and pre-rRNA, which is “in agreement with previous observations”.

a) Please define variability of mismatch frequency across ≥ 3 replicates.

We will now include a third replicate for yeast total RNA samples sequenced using Nano3P-seq, to further confirm the absence of m1acp3Y in pre-rRNA, relative to mature rRNA, and include p-values.

b) Please validate the m1acp3 estimate with a different method.

To provide an orthogonal validation of this result, we will now include a direct RNA sequencing run of in vitro polyadenylated total yeast RNA, to show that m1acp3Y can also be detected in mature rRNA but not pre-rRNA.

8. U, UU, UUU and UUUU 3'ends are extremely rare. There is little proof that these are not sequencing errors or other artifacts.

a) Please use spike-in sequences with A, U, UU, UUU and UUUU ends. Process these in separate runs and define recall and precision.

b) Please compare frequency of U runs (length=1,2,3,4) to G and C runs.

c) Please prove that imperfect adapter sequences do not contribute nonA nucleotides

In addition to using the sequins to assess the quantitative ability of Nano3Pseq to predict polyA tail lengths (which contain either 30nt or 60nt long tails), we will now also sequence using Nano3P-seq a new set of synthetic sequences (e.g. yeast enolase) that would contain polyA tails of diverse known lengths (0, 25, 50, 100, 150 and 200 nt) as well as with polyU, polyC, and polyG tails of diverse lengths. These synthetic sequences will also be sequenced using direct RNAseq to be able to directly compare the performance of each of the two methods.

Minor Comments:

- Depiction of poly(A) tails in Figure 1c, S1c is suboptimal. In S1c tail lengths and frequency are difficult to see.

We apologize if the depiction of polyA tails were suboptimal in Figure 1C and S1C, but the goal of these figure panels was not to quantify the tail lengths (this is done in Figure 2) but rather, to illustrate that the tails of diverse lengths (including deadenylated mRNAs) are captured using Nano3P-seq.

To address the reviewer's comment, we will now include supplementary panels that will illustrate the corresponding tail length distributions of the reads underlying Figure 1C and S1C.

- For figure 1C the authors say "In addition, our results confirmed that polyA tail length information was retained in individual reads. Specifically, the majority of reads corresponding to mRNAs had polyA tails". 1C suggests that it is barely a majority: many reads have no green PolyA. This example plants more doubts than it helps the reader.

We will now clarify this in the text.

- Figure 1D lacks detail. Is correlation given before or after log-transformation?

The correlation is reported after log-transformation. This will now be clarified in the figure legend.

- Fig 2B: What are the correlations between the time points

We will now include panels depicting the correlations between time points into Supplementary Figure S3.

- Figure 3a: The material used here is not obvious. Is it the same as in Figures S1a-c?

Yes, it is the same material as shown in Figures S1A-C. We will clarify this in the Figure 3a Figure legend.

- References 27 and 28 seem identical

We thank the reviewer for pointing this out. This was an error due to the automatic bibliography system that considered the two references different. This will now be corrected in our resubmission.

- Figure 4a only shows one example of differences conserved across time points, but the main text says "often conserved across the different time points analyzed (Fig 4a)"

We will now refer also to a new supplementary table, which will show the median tail lengths for each isoform and for all time points. In addition, we will build a supplementary figure panel showing the difference in median tail lengths per isoform.

- Gene and isoform percentages (5.2% and 11.7%) in the text for Figure 4 require confidence intervals and exact numbers.

We will now include statistics with the inclusion of a third replicate.

- Figure 5D is barely mentioned in the text

We will further discuss this Figure panel in our revised version of the manuscript.

Reviewer #2:

Remarks to the Author:

SUMMARY

This paper presents a strategy for analyzing RNA expression using nanopore sequencing of first-strand cDNA copies primed by strand switching. A strength is detection and length estimates of 3' homopolymers, and a relatively straightforward library preparation scheme. The paper is casually written and not ready for careful consideration in its current form. Some essential data are missing, and some of the data are not presented in a transparent manner. This is surprising given the substantial experience of some of the co-authors. In the text that follows we highlight key technical and logical issues, and then we note specific issues line-by-line.

We thank the reviewer for his/her feedback and time in reviewing our work. We agree with the reviewer that Nano3P-seq has the strengths of detecting and estimating length of 3' homopolymers as well as a relatively straightforward library preparation scheme. We apologize if the reviewer found some parts of the manuscript to be casually written, to this end we have now carefully edited the manuscript text as well as have had a professional scientific editorial company review and edit the text, to enhance its clarity and writing style. With regards to the missing data and presentation of the results, we have now added all clarifications and edits following the reviewer's comments, We thank the reviewer for his/her time and efforts in helping us improve the quality of our manuscript.

GENERAL COMMENTS

Title and Abstract

1. *It would be informative to include 'cDNA' in the title and abstract. It was unclear to us until well into the introduction that the new technique was based on first strand cDNA rather than native RNA.

We agree with the reviewer's comment, and will now add the word "cDNA" in the abstract, to ensure this is clear to all readers.

2. *Here and elsewhere the authors use the term 'translatability'. Possibly this is now used in the molecular biology community, but a more traditional term is 'translational efficiency'. In either case, for a general audience it would be useful to precisely define what you mean.

The word translatability, which appeared twice in the manuscript, will now be changed to translational efficiency.

3. *In the last sentence of the abstract, the authors assert that Nano3P-seq can accurately estimate transcript levels. It is unclear if this is at the gene level or at the isoform level. In the last sentence of the abstract, the authors assert that Nano3P-seq can accurately estimate tail lengths of full-length transcripts. As far as we can tell, Nano3P-seq does not establish unequivocally that reads are full-length.

We will now clarify in the abstract that Nano3P-seq can accurately estimate at per-isoform level.

With regards to the second comment of the reviewer, we will remove the words "full length" from the sentence, as we agree it could be misleading. The sentence will now read: "Nano3P-seq can accurately estimate tail lengths of transcripts".

Results

4. *The current draft does not adequately address performance of the Nano3P-seq strategy. Most notably, in the main text there are no data for throughput, and as best we can tell, there are no data for base call accuracy or read length distribution anywhere in the manuscript. These data are important for a fair comparison between direct RNA sequencing and Nano3P-seq (see Discussion section comments below). It is known that cDNA basecall accuracy is typically better than direct RNA basecall accuracy. So why not point this out in the text along with some data?

We thank the reviewer for his/her suggestions. We will now add information regarding the base-calling accuracy and read length distribution for each of the Nano3P-seq sequencing runs included in this work. This information will be included in a new supplementary figure. Moreover, following the reviewer's suggestion, we will discuss these results both in the manuscript text as well as in the Discussion section, and compare them to those observed when using direct RNA sequencing.

5. *The read count log plots are not up to NMeth standards and will be difficult for readers to assess. These include panels Fig 1.D, Fig 2.B, and Fig S1.H. Let's examine Figure 1.D as an example:

To build the figures that the reviewer mentions above, we used normalized read counts (i.e. counts that had been scaled/normalized across samples and replicates to have equal coverage across biological replicates). However, we agree with the reviewer that this scaling might make it difficult for readers to assess certain aspects of the performance of Nano3P-seq. Therefore, we will now rebuild all figures to $\log(\text{read counts})$ instead of $\log(\text{normalized read counts})$. We thank the reviewer for pointing us to this issue.

i) The X axis is labelled 'Log (Expected Counts)'. What is meant by expected counts in this context and how were these values derived? This is not defined in the panel, text, or Methods as far as we could see. Why not use the X-axis units as in the original Sequins manuscript, i.e. \log_{10} Attomoles/ μL ? It is noteworthy that several of the authors on this paper were co-authors on the carefully-executed Sequin (2016) paper.

We will now replace $\log(\text{Expected counts})$ for $\log_{10}(\text{attomoles}/\mu\text{L})$.

ii) It is unclear to us if the data in this panel are for genes or isoforms. In the Sequins(2016) paper, the authors presented Counts vs Attomoles/ μL for genes, isoforms, and exons. This trio of plots would be a useful addition to the Nano3P-seq paper.

We will now include this trio of plots in the Nano3P-seq paper as a new supplementary figure.

iii) To facilitate the reader's examination of these data, please plot untransformed data on \log_{10} - \log_{10} plots with tick marks. Once again, the original Sequins manuscript can serve as a guide.

Please see above.

iv) Please include the slope value for the fitted line. A naive examination of the original Fig 1D might suggest a 1-to-1 correspondence between expected and observed values which appears not to be the case.

This information will be included in the figure legends.

v) In the Sequins (2016) paper, the relationship between counts and concentration was flat below 1 attomole/ μL . The paper under review does not use concentration on the X axis so it is difficult to make a comparison. We are curious if any of the Sequin RNA isoforms were not observed in these control experiments.

Please see above. We will modify the axis of these plots to enhance their clarity.

vi) It is customary to use axis values beyond the upper and lower limits of plotted data. Please fix this.

This will be fixed, and we will ensure that in our resubmitted version the axis values go beyond upper and lower limits of plotted data in all panels of all main and supplementary figures.

vii) In Figure 2B the scales on the plots do not bound the data. Fix this. Also, what was the logic for using natural log? Was this the best way to convey the information to a reader – is that the explanation? Also, the highest density in the distribution is not bound by the axes. Please fix this.

This will be fixed. We will also change the natural log for log10.

6. *Line 209: section title. The authors claim that tail length can be accurately estimated using Nano3P-seq. Please explain to the reader what sort of precision they should anticipate and whether or not that precision is tail length dependent. Also, in Figure S3B, there appears to be a reasonable correlation between the two replicates in the range of 30-60 nt, but that they both overestimate tail lengths at 30 and 60 nucleotides. This is not a big issue, but please be forthright about what the technique delivers.

The precision of tail length estimation will now be assessed with new additional synthetic oligonucleotides containing a broad range of known polyA tail lengths (0,25,50,100,150,200).

7. *Figure 3G: Are the values for transformed or untransformed data?

These were transformed values. We will clarify this in the text and/or change them to untransformed data..

8. *The paper claims isoform documentation. However, Nano3P-seq does not unambiguously establish that the biological 5 prime end of transcripts have been sequenced. Please defend this position. Related (as pointed out above) there is no documentation of the read length distribution for this method.

The reviewer is correct with regards to the fact that nanopore sequencing (including Nano-3Pseq) sequences the RNA or DNA molecules from their 3' ends, and consequently, there will be a 3' sequencing bias, i.e. with decreased coverage of the 5' ends of the molecules, as can be seen in several figure panels (e.g. Figure 1C, Figure 4C, S1E-G). The assignment of read to isoform was done via a third-party software, specifically, we used IsoQuant (<https://github.com/ablab/IsoQuant>) using the read-to-isoform algorithm parameters that are recommended by the developers for nanopore data. The assignment of the isoforms based on IsoQuant is as follows:

- If the read intron chain matches a single known isoform, it is reported as unique. Even if the 5' is degraded, the read may still contain enough information to unambiguously decide which isoform it is;
- If it's degraded such that multiple isoforms match, read is reported as ambiguous;

- If the intron chain matches to none of the isoforms from the annotation it is reported as inconsistent (potentially novel isoform or misalignment).

On the other hand, we will now include read length distributions for the method in the form of additional supplementary figures (see also response to point #4).

Finally, we should note that the fact that we do not establish the biological 5' end does not significantly affect the ability of Nano3P-seq to quantify RNA abundances at the isoform level, as we will now show with our updated plots showing sequin expected VS observed abundance both at per-gene and per-isoform level.

DETAILED COMMENTS

9. Line 73. Try making the sentence more clear by bringing the subject and verb together.

This will be fixed in the resubmitted version.

10. Line 96. Others have extended the 3 prime ends of RNA strands using poly(I) followed by dRNA sequencing. Please see Vo et al RNA 2021. 27: 1497-1511) and Drexler et al Molecular Cell Volume 77: 985-998 (2020).

We will mention this work in this section and cite it.

11. Lines 101-104. Please reconcile these assertions with the preceding comment.

This will be fixed in the resubmitted version.

12. Line 121. Please revisit this statement in light of Vo et al. above.

This point will be revisited as suggested.

13. Line 153. Interpreting data in figure legends (as was done here) is usually discouraged. We are not sure of NMeth's policy on this.

We will edit the figure legend according to the NMeth's policy.

14. Line 163. Please see discussion of Figure 1D in the general comments section.

We will now take the discussion in the general comments section into account and rewrite this part.

15. Line 177. 'Drastic' is an odd word choice here. 'Drastic' measures are not just extreme, they are likely to have harmful side-effects. Better to use a neutral word.

We will choose another word instead of "drastic".

16. Line 191. Polyadenylation of Mt 16s rRNA is also observed in human samples (see citation 12 in this draft paper).

We will now include this information in the sentence with the related citation.

17. Line 220. Here the fit is r^2 , but in Figure S3B it is r . Which is it? This lack of care in data presentation is concerning.

We apologise for this mistake. The fit is r throughout the paper.

18. Line 258 (E). Awkward sentence.

We will now rewrite the sentence.

19. Line 289. Here the advantage of nanopore read lengths is highlighted, but nowhere do the authors document the read length distribution of their technique. This needs to be included.

We will include the read length distribution of our technique.

20. Line 310. The authors make a general statement '.....it demonstrates that Nano3P-seq can provide transcriptome-wide measurements of the polyadenylation status of diverse biological samples'. This is based on a few examples. While this may one day prove to be the case, claiming transcriptome-wide measurements is premature based on this paper.

We respectfully disagree with the reviewer on this comment. In our manuscript, we show that Nano3P-seq is applicable both in in vitro samples (curlcakes, sequins) as well as in vivo, specifically showing that it is applicable transcriptome-wide in 3 different species (zebrafish, yeast, mouse), across different developmental stages (2,4,6hpf) in zebrafish, and across biological replicates. Therefore, we believe that we have demonstrated that Nano3P-seq can provide transcriptome-wide measurements of polyadenylation status across biological samples.

21. Line 401. Not an accurate statement. See Vo paper and Drexler paper cited above.

We will now change this statement according to these papers.

Reviewer #3:

Remarks to the Author:

This manuscript reports the development of a new method, Nano3P-seq, which uses Oxford Nanopore sequencing to provide information on RNA expression level, poly(A) tail composition, and poly(A) tail length. This does not rely on PCR and can also capture non-poly(A) transcripts. The authors apply their new method to some relevant samples and confirm findings generally known about poly(A) tails. This application part is well done and a strength of the manuscript. More generally, there are many aspects of biology where this method could be used. Unfortunately, this manuscript falls short in two related areas – (1) how accurate is the method for poly(A) composition and length and (2) how does it compare directly to other methods for measuring poly(A) tails.

We thank the reviewer for his/her comments, and agree that the method could be used for many different biological aspects and questions. We also agree with the fact that additional benchmarking would be beneficial for the paper, and therefore, we have now planned to include the following additional experiments:

- In addition to using the sequins to assess the quantitative ability of Nano3P-seq to predict polyA tail lengths, we will also sequence a set of synthetic sequences using Nano3P-seq, which contain polyA tails of diverse known lengths (0, 25, 50, 100, 150 and 200 nt). This will address the first area (1) that the reviewer mentions above.

- In addition to sequencing these synthetic samples with Nano3P-seq, they will also be sequenced with direct RNA seq. This will address the second area (2) that the reviewer mentions above.

Major concerns.

1) The manuscript needs to be more rigorous in showing that Nano3P-seq is accurate in measuring poly(A) tail composition and length.

Figure S1 shows that poly(A) tail length can be measured but given that the initial tail length was not carefully characterized, it is not possible to assess accuracy in the sequence data. In Figures 3B, S3C, and S3D, the accuracy of the tail length is measured for the Sequins with known lengths of 30 and 60 bases. That is a good start, but it's important to test a wider range of lengths. Moreover, the authors mention that the measured length with tailfindR was off by 15 bases and they adjusted all measurements accordingly (lines 646-652). It was not adequately explained how they know that this is due to the expectation of a double-stranded cDNA by tailfindR. Why would this be true for both 30 and 60 base tails?

For poly(A) tail composition, the manuscript does not present any data on the error rate for these sequencing runs, the base quality in the poly(A) tail, nor the accuracy of the bases with spike-ins of

known composition. Even without PCR, it's known that reverse transcription and nanopore sequencing produce errors. Error rates should be shown both for base substitutions and insertion/deletions. The authors focus on terminal U residues (Figures 5C and 5D), but also show other bases are present in the last 10 residues (Figure S7A). They should also explore bases along the entire length of the poly(A) tail with known controls to see whether non-A bases in other positions can be accurately measured and studied.

With regards to the first point, to further assess the accuracy of the tail length measurements by Nano3P-seq, we will sequence a new set of synthetic sequences (in addition to the sequins, already included in the manuscript) which will contain polyA tails of diverse known lengths (0, 25, 50, 100, 150 and 200 nt). These sequences will be used to determine the accuracy of the Nano3P-seq method for tail length estimation.

To answer the second point of the reviewer, we will now include quantification of the error rates observed in the tail regions as well as in the whole reads (using the sequins) in Nano3P-seq datasets. Moreover, we will also sequence a new set of synthetic sequences that will contain tails of diverse known lengths and compositions, including polyA polyU, polyC and polyG tails of diverse lengths. We believe that these additional controls will address the reviewer's concerns.

2) One of the most important aspects of a manuscript introducing a new method is to demonstrate how it compares to existing technologies. In the Introduction, the authors mention that PAL-seq and TAIL-seq are limited because they use Illumina short reads and rely on PCR. There, they also mention the dRNA-seq nanopore method is not able to sequence non-poly(A) RNA and their tails. Note that dRNA-seq does not rely on PCR. In the Discussion, the authors mention PacBio-based methods, FLAM-seq and PALso-seq, which provide all the information that Nano3P provides for poly(A)+ transcripts, but with a PCR amplification step. The reads are more expensive, though this may not be a major factor in such experiments and details are not provided. While the PacBio sequencers are expensive, the authors should mention that there are many facilities that will sequence samples submitted by any scientist. If Nano3P-seq provides a meaningful advance for the field, there should be a direct comparison with one of the best Illumina and one of the best PacBio methods with aliquots from the exact same sample. The authors do compare to dRNA-seq in Figures 2F, 3B, S3C, S3D. Additionally, they compare PAL-seq to Nano3P-seq in Figure S4A and S4B, but these are published data derived from a different sample for PAL-seq. There are clearly differences too, so that it is not clear whether they are meaningful, nor which method is more accurate. If there are PCR biases that are not present in this new method, the authors need to show that. And the best way to do this is with a set of spike-in controls, including the Sequins, but also ones with different known poly(A) tail lengths and compositions (as in the PALso-seq paper, ref. 44, but not necessarily limited to what is done in that paper).

We fully agree with the reviewer that introducing a new method requires demonstration of how it compares to existing technologies. We would like to note that in our opinion, however, we had already included such information in the draft. Specifically, our current draft included:

- i) comparison of Nano3P-seq to direct RNAseq, in synthetic sequins (Figure 3B, S3C and S3D)
- ii) comparison of Nano3P-seq to direct RNAseq, in vivo biological zebrafish samples (Figure S8A-B).
- iii) comparison of Nano3P-seq to PAL-seq, in in vivo zebrafish samples at t=2, 4, 6hpf (Figure S4A).

Therefore, we believe that this concern was already addressed in the manuscript, as we had compared Nano3P-seq performance with two different orthogonal methods, both in vivo samples as well as in vitro samples.

To further address the reviewer's concern, we will: i) work on the text and figures to further clarify the performance and predictions of Nano3P-seq relative to the other two methods (direct RNA nanopore sequencing and PAL-seq), and ii) build direct RNA nanopore sequencing libraries of the new synthetic spike-ins that are designed with polyA and polyU tail ends, as well as with different tail lengths, to compare side by side Nano3P-seq and direct RNA nanopore sequencing on an additional dataset.

Minor concerns.

1) The authors should mention how the input RNA amount for this protocol compares to other protocols because this can be a key experimental design feature. In the Methods section, they start with 100 ng of total RNA, but what is the lower limit for this method? This could be an issue given that this method does not use PCR.

Nano3P-seq libraries use 100ng of input RNA material per sample in a similar fashion to what standard direct cDNA nanopore libraries use. We have not systematically tested decreasing input amounts, as we reasoned that the input amounts required for efficient library preparation should be similar to those of direct cDNA library preparations, where there is no amplification of the cDNA, in a similar fashion to Nano3P-seq libraries. By contrast, direct RNA sequencing libraries use 500ng of input material. Thus, Nano3P-seq provides a similar throughput of sequenced reads as direct RNA sequencing (~1-2 million reads per flowcell) but requires 5X less input RNA material, compared to direct RNA sequencing. We will now mention this in the Discussion.

2) While the authors show nicely that the RNA expression of the Sequins is accurate, what about looking at biases such as 5' vs. 3' ends of the transcript, GC content, and transcript length?

We thank the reviewer for his/her suggestion. We will now include these statistics in a new Supplementary Figure.

3) Line 180-181: What are miscRNAs and scaRNAs?

The MiscRNA category includes diverse RNA types. According to ENSEMBL, the definition of miscRNA is: “Miscellaneous RNA. A non-coding RNA that cannot be classified”. ScaRNAs correspond to “small cajal-body specific RNAs”.

4) Lines 185-191: Although the authors found poly(A) tails on 16S mitochondrial rRNA in both zebrafish and mouse, the conservation of this finding does not fully validate it as correct. The text could be modified to clarify this point.

We will further validate this observation with another method, e.g. using qPCR. In addition, we will include additional replicates supporting these observations.

5) Box plot features should be explained in Figures 3B,D,G,H and elsewhere.

We thank the reviewer for pointing us to this issue. This will now be clarified in the figure legends.

6) Lines 296-297: What is the justification for a threshold of 10 reads per isoform?

We chose 10 reads per isoform as we reasoned that the number is large enough to ensure that the sampling will be representative of the whole population, while maximizing the number of genes or isoforms that will be kept for downstream analyses. We should note that we initially performed this analysis with 30 reads as threshold, but that decreased the number of isoforms kept in the analyses, and the conclusions did not differ from those obtained using 30 reads coverage. Many bioinformatic softwares and analyses in nanopore sequencing data require a minimum of 5-50 reads coverage to support a given site, isoform, modification, etc. In this case, we require a minimum of 10 reads coverage to include the isoform in downstream analyses.

7) Line 343: Perhaps change to “modification” (singular) as there is only one type assessed here.

This will be changed.

8) Line 407: TGIRT should be spelled out and the explanation for why it is better should be clarified.

This will now be clarified in the text and the acronym will be spelled out.

9) Lines 426-439: Perhaps this should be in the Result section?

We will consider moving it to Result section.

10) Line 454: If the authors state their method is “low cost,” they should present information on the cost of reagents for all steps in this method.

We disagree with the reviewer, and consider it beyond the scope of that in order to state in the Discussion section that nanopore is low-cost, we need to present information of the cost of reagents for all steps in the methods. We will however cite other references that already showed and/or compared PacBio, Illumina and Nanopore sequencing.

11) Experimental details are missing in the Materials and Methods section.

All the additional experimental details mentioned below will be amended in the Methods section.

- a. Line 481: Sodium Acetate concentration?
- b. Line 485: Turbo DNase details (concentration, time, temperature) missing
- c. Line 489: How were the mice cared for (feeding, day/night schedule, etc.)? Were the mice male or female?
- d. Line 493: Details about “Protease Inhibitor” missing.
- e. Line 494: Details about “dounce” (# of times, size) missing.
- f. Line 503: Details about “RNase Inhibitor” missing.
- g. Line 521: Details about which “oligo (dT) magnetic beads” missing.
- h. Line 532: What is “bead resuspension”? Is this from the Ribodepletion kit or the Dynabeads kit?
- i. Line 533: What is “bead resuspension buffer”? Same as above.
- j. Line 536: What is “bead wash buffer”? Same as above.
- k. Line 537: What is “depletion buffer”? Same as above.
- l. Lines 570, 572, 573: What is the source of the buffers, Adapter Mix, ABB Buffer, Elution Buffer, Sequencing Buffer, and Loading Buffer? All from Oxford Nanopore?

Yes, all the buffers are from Oxford Nanopore, as part of the library preparation kit that is described and used as part of the Nano3P-seq library.

m. Line 574: Please provide details on the length of the runs and whether any library reloading was done.

Library reloading was not done. For this reason we did not include this information in the table. The duration of the runs will be included as an additional column in Table S5.

n. Line 663: What about the care of the mice?

We will add a statement for the mice samples used in the Animal Ethics Statement.

o. Figure S1: Perhaps include an explanation for the term “curlcake” and refer the reader to the Materials and Methods for details on CC1 and CC2.

We have now referred to the Methods in Figure S1 legend as suggested by the reviewer.

Decision Letter, first revision:

Dear Eva,

Thank you for your letter asking us to reconsider our decision on your Article, "Nano3P-seq: transcriptome-wide analysis of gene expression and tail dynamics using end-capture nanopore sequencing". After careful consideration we have decided that we are willing to consider a revised version of your manuscript that will include additional experiments and analyses proposed in your appeal letter.

Please note that at this stage we cannot make any specific promises, even about sending the revised paper back to the reviewers, until we have read it in its entirety.

- * include a point-by-point response to our referees and to any editorial suggestions
- * please underline/highlight any additions to the text or areas with other significant changes to facilitate review of the revised manuscript
- * address the points listed described below to conform to our open science requirements
- * ensure it complies with our general format requirements as set out in our guide to authors at www.nature.com/naturemethods
- * resubmit all the necessary files electronically by using the link below to access your home page

[Redacted]

We hope to receive your revised paper within eight weeks. If you cannot send it within this time, please let us know. In this event, we will still be happy to reconsider your paper at a later date so long as nothing similar has been accepted for publication at Nature Methods or published elsewhere.

OPEN SCIENCE REQUIREMENTS

REPORTING SUMMARY AND EDITORIAL POLICY CHECKLISTS

When revising your manuscript, please submit reporting summary and editorial policy checklists.

IMAGE INTEGRITY

When submitting the revised version of your manuscript, please pay close attention to our [href="https://www.nature.com/nature-research/editorial-policies/image-integrity">Digital Image Integrity Guidelines. and to the following points below:](https://www.nature.com/nature-research/editorial-policies/image-integrity)

DATA AVAILABILITY

Please include a "Data availability" subsection in the Online Methods. This section should inform readers about the availability of the data used to support the conclusions of your study, including accession codes to public repositories, references to source data that may be published alongside the paper, unique identifiers such as URLs to data repository entries, or data set DOIs, and any other statement about data availability. At a minimum, you should include the following statement: "The data that support the findings of this study are available from the corresponding author upon request", describing which data is available upon request and mentioning any restrictions on availability. If DOIs are provided, please include these in the Reference list (authors, title, publisher (repository name), identifier, year). For more guidance on how to write this section please see:

<http://www.nature.com/authors/policies/data/data-availability-statements-data-citations.pdf>

CODE AVAILABILITY

Please include a "Code Availability" subsection in the Online Methods which details how your custom code is made available. Only in rare cases (where code is not central to the main conclusions of the paper) is the statement "available upon request" allowed (and reasons should be specified).

For more information on our code sharing policy and requirements, please see:
<https://www.nature.com/nature-research/editorial-policies/reporting-standards#availability-of-computer-code>

SUPPLEMENTARY PROTOCOL

To help facilitate reproducibility and uptake of your method, we ask you to prepare a step-by-step Supplementary Protocol for the method described in this paper. We [encourage authors to share their step-by-step experimental protocols](https://www.nature.com/nature-research/editorial-policies/reporting-standards#protocols) on a protocol sharing platform of their choice and report the protocol DOI in the reference list. Nature Research's Protocol Exchange is a free-to-use and open resource for protocols; protocols deposited in Protocol Exchange are citable and can be linked from the published article. More details can found at www.nature.com/protocolexchange/about.

ORCID

Best regards,
Lei

Lei Tang, Ph.D.
Senior Editor
Nature Methods

Reviewers' Comments:

Reviewer #1:

Remarks to the Author:

Begik et al present the Nano3P-Seq method. Nano3P-Seq uses adapters and a template-switch reaction. It makes cDNAs with gene body and entire poly(A) tail. Importantly, it can equally analyze PolyA(-) RNAs. The authors apply Nano3P-Seq to the maternal-to-zygotic transition (MZT) and compare mouse, zebrafish and yeast transcriptomes. They find increased non-coding RNAs as the MZT progresses. They also see longer poly(A) tails during the MZT and show differences between different decay modes. Furthermore, poly(A) lengths also distinguish isoforms in one gene. Mismatch frequencies correlate very highly between pre-rRNA and mature rRNA in yeast and mouse. M1acp3, however is more frequent in mature rRNA. Finally, authors show that RNAs with polyuridylated tail ends have different tail lengths.

Nano3p-Seq has conceptual advances over published methods (e.g., FLAM-Seq). However, extensive experimental work is needed to prove that all findings are correct. Moreover, parts of the work lack clarity.

We thank the reviewer for his/her feedback and time in reviewing our work. In this revised version, following the reviewer's suggestion we now include additional experimental work to further validate and benchmark the Nano3P-seq method. Therefore, in our revised version of our manuscript, we have now:

1. **Increased the number of replicates of certain experiments:**
 - a. We have sequenced a third replicate of zebrafish time-course experiments (for each time point) and have sequenced it using Nano3P-seq, allowing us to provide p-values to assess the significance in the increased representation of certain RNA biotypes during the embryo development.
 - b. We now include an additional replicate for yeast total RNA samples sequenced using Nano3P-seq, to confirm the absence of m1acp3Y in pre-rRNA, relative to mature rRNA, in independent biological replicates.
2. **We have designed and sequenced a set of synthetic sequences with a wide range of known polyA tail lengths to further assess the accuracy of Nano3P-seq in the estimation of polyA tail lengths.** In our previous version, we were using the "Sequins" to assess the ability of Nano3P-seq to accurately predict polyA tail lengths (which contain either 30nt or 60nt long tails). In our revised version, we have now **also sequenced a set of novel synthetic sequences with a broader range of known polyA tail lengths (0, 15, 30, 60, 90 and 120 nt).**
3. **We have designed and sequenced using Nano3P-seq a novel set of synthetic sequences to assess the ability of Nano3P-seq to accurately identify and quantify tail composition variations.** Specifically, we have now included synthetic sequences which include diverse tail end compositions (e.g. AAAAU,AAAAC,AAAAG,AAAAA), with diverse lengths (e.g. U, UUU, UUUUU) as well as sequences with internal Gs.
4. Two additional orthogonal methods (polyA tail length kit and direct RNA sequencing) have been used to **validate the presence of polyA tails in mitochondrial LSU rRNA.**

We hope that this additional experimental work will now address all the concerns/comments that the reviewer might have had in the previous submitted version.

Major comments:

1. "Indeed, per-read analysis of mitochondrial rRNA reads revealed that a significant proportion of 16S mitochondrial rRNA contained a polyA tail": Is Nano3P-Seq accurately quantifying both versions of 16S mitochondrial rRNA? Experimental validation is needed here.

Following the reviewer's suggestion, we have now included a new figure panel depicting the number of reads mapping to mouse mitochondrial SSU rRNA, with and without polyA tail, respectively (**Figure S2i**).

With regards to the second point, we now provide experimental validation of these results, using two different orthogonal methods: i) qPCR experiments (**Figure S2j**, left panel), and ii) validation of the polyA tail presence using a commercial "PolyA tail assay kit", and sequencing the corresponding amplified bands using Sanger sequencing (**Figure S2j**, right panel). All methods confirm the presence of polyA tail in the 16s mitochondrial rRNA. qPCR supports similar proportions of polyA tailed RNA, but with higher standard deviation than Nano3P-seq.

For the reviewer's convenience, the figures mentioned above are also embedded below.

2. Figure 1B is central but hard to interpret. Y-axis ("Counts") and legend ("Nano3P-seq captures a wide range of RNA biotypes in the mouse brain") lack clarity. Are these gene counts or read counts? What is the exact definition? They seem low for mRNAs (<2000?), which raises doubts. Similar issues affect Fig 2D.

We apologize if this was not sufficiently clear in our previous submission. These plots were log (normalized counts), not absolute read counts. This normalization was done to ensure that all replicates had the same sequencing coverage/depth. However, we agree that this could be confusing for the reviewers and future readers, so we have now modified all plots that were using log(normalized counts) for log(read counts). For the reviewer's convenience, below we embed the new updated versions of **Figure 1B** and **Figure 2D**.

Figure 1B:

Figure 2D:

3. In Figure 2, the authors use ribo depletion [“we isolated total RNAs from zebrafish embryos at 2, 4 and 6 hours post-fertilization (hpf) in biological duplicates, ribo-depleted the samples, and sequenced them using the Nano3P-seq protocol”]. Then they find similar rRNA and mRNA abundance (Figure 2D). Did ribo depletion not work? Is a different sample used here? Please clarify.

From our experiments, we found that ribodepletion oligonucleotides were not as effective in ribodepleting maternally rRNAs compared to zygotic rRNAs. Total RNA profiles before and after ribodepletion can be found in Figure S2A in our original submission (also embedded below for the reviewer’s convenience), where it can be seen that 18s maternal rRNA ribodepletion was more effective than 28s maternal rRNA ribodepletion.

Figure S2A:

We should note, however, that incomplete ribodepletion will not affect Nano3Pseq performance, but it will lead to a higher proportion of rRNAs in the sequenced zebrafish embryo samples. In order to avoid future confusion by the readers and clarify this point, and also to avoid discrepancies in rRNA

abundances across replicates due to differences in ribodepletion efficiency, we have considered that it is best to not include rRNAs in **Figure 2D**, since the abundance of rRNAs in ribodepleted samples varies across replicates depending on the efficiency of ribodepletion, and can be a confounder in the results (we should note that this is in fact the case, i.e. we have observed that ribodepletion efficiency varies across replicates if the ribodepletion experiment is not performed at the same time and/or with the same set of buffers/oligos).

4. There is interesting biology in Figure 2D: "the abundance of non-coding RNA populations, including miscRNAs, scaRNAs and snoRNAs, increased as the MZT progressed", but p values are not indicated. Please give p-values and statistical details.

In our previous submission, we had not provided p-values as we had only included two biological replicates for each developmental time point in our previous analyses. To address the reviewer's question, we have now included a **third replicate** for each time point of the zebrafish time-course experiments (t= 2, 4 and 6hpf) sequenced using Nano3P-seq. This has now allowed us to obtain p-values to assess the significance of the increased representation of certain RNA biotypes during the embryo development. Our analysis show that miRNAs, lincRNAs and snoRNAs are significantly up-regulated during MZT, as shown in new Figure 2D (embedded above, comment #2).

5. Figure 3B, for the 30nt-tail sequences, dRNAseq seems to have medians closer to the expected line than Nano3P-Seq (e.g., for R1_14, R1_103 & R1_73). For some others Nano3P-Seq seems to be better. Please present details in text and improve figure for easy visual assessment.

Following the reviewer's suggestion, to address this point we have now made a new Supplementary Table (**Table S4**) that includes the median polyA tail length prediction for each sequin, to ensure that all details are available to the reviewer and future readers, in addition to showing the results in **Figure 3B**. We should note that **Figure 3B** has now been moved to Supplementary Material (**Figure S3C**), as we have now updated the Main Figure panel to also include cDNA standards which cover a broader range of polyA tail lengths than the sequins (new **Figure 3C**).

6. The manuscript focuses on zebrafish, but analysis of pre- and mature rRNA only uses mouse and yeast. What is the reason? Please, compare across >2 species with identical experiments.

We now clarified in the text that zebrafish embryos do not have *de novo* transcription of zygotic rRNAs during early embryo stages (Locati et al., 2017: PMID:28500251). This is the reason why we only included yeast and mouse samples in the pre-rRNA/mature rRNA comparative analyses, but not zebrafish samples (i.e. we did not find any read mapping to pre-rRNA at any of the time points studied (t=2,4,6hpf). To clarify this in the manuscript, we have now incorporated the text below into the Results section of the manuscript:

"This analysis could not be performed in the zebrafish Nano3P-seq datasets due to the absence of de novo transcription of zygotic rRNAs in early embryo stages (...)"

7. The authors determine m1acp3 levels in rRNA and pre-rRNA, which is "in agreement with previous observations".

We have now included more references in the text, related to the late appearance of m1acp3Y modifications in the maturation of rRNA in different species. Please see below the references we added in the manuscript:

Reference 1 : Brand, R. C., Klootwijk, J., Planta, R. J. & Maden, B. E. Biosynthesis of a hypermodified nucleotide in *Saccharomyces carlsbergensis* 17S and HeLa-cell 18S ribosomal ribonucleic acid. *Biochem. J* 169, 71–77 (1978).

Reference 2 : Liang, X.-H., Liu, Q. & Fournier, M. J. Loss of rRNA modifications in the decoding center of the ribosome impairs translation and strongly delays pre-rRNA processing. *RNA* 15, 1716–1728 (2009).

a) Please define variability of mismatch frequency across ≥ 3 replicates.

We have now included mismatch frequency across replicates in yeast, as well as in mouse, in order to show the variability of mismatch frequency across replicates. Figures are embedded below for the reviewer's convenience, and show that there is little variability in the results in both species.

Figure 4d (mouse):

Figure S4e (yeast):

b) Please validate the m1acp3 estimate with a different method.

To provide an orthogonal validation of this result, we have now included a direct RNA sequencing run of *in vitro* polyadenylated total yeast RNA, showing that the mismatch signature between pre-rRNA and mature rRNA only differs on the m1acp3Y site, in agreement with our Nano3P-seq results (Figure S4d).

8. U, UU, UUU and UUUU 3'ends are extremely rare. There is little proof that these are not sequencing errors or other artifacts.

a) Please use spike-in sequences with A, U, UU, UUU and UUUU ends. Process these in separate runs and define recall and precision.

Following the reviewer's suggestion, in addition to using the 'sequins' to assess the quantitative ability of Nano3Pseq to predict polyA tail lengths (which contain either 30nt or 60nt long tails), we have now also sequenced with Nano3P-seq an additional set of synthetic cDNA sequences that contain polyA tails of diverse known lengths (0, 15, 30, 60, 90 and 120 nt) (Figure 3b, embedded below) as well as with polyU (U, UUU or UUUUU), polyC, and polyG tails of diverse lengths (Figure 5a, embedded below). Each of these standards was independently barcoded and sequenced using Nano3P-seq, as the reviewer suggested.

Figure 3b:

Figure 5a:

The accuracy of the predictions is shown in Figure 3c for tail lengths, and Figure S5a,b for tail heterogeneity, both embedded below.

Figure 3c:

Figure S5a,b:

b) Please compare the frequency of U runs (length=1,2,3,4) to G and C runs.

Following the reviewer's suggestion, we have now reanalyzed the tail composition of Nano3Pseq datasets to include in the analysis all possible variations in terms of tail heterogeneity (i.e. not focused only on terminal Us). The tails were binned depending on whether they contained internal non-A bases (Int-C, Int-G, Int-U) or terminal non-A bases (Term-U, Term-G, Term-C). We observe that the most abundant type of non-A bases in tails are in fact Gs (Figure 5d). Each of the tail types was then compared in terms of polyA tail lengths, finding that once the maternal genome has been cleared, terminal Us are found in polyA tail lengths that are typically shorter than the median tail lengths seen at that given time point (Figure 5e)

c) Please prove that imperfect adapter sequences do not contribute nonA nucleotides

To exclude the possibility that imperfect adapter sequences could contribute to terminal non-A nucleotides that were observed in the data, we also included in the analysis of non-A base composition the synthetic sequin RNA standards, which are expected to contain exclusively A bases. Our results show that the observed frequency of non-A bases in *in vivo* tails is much higher than the one observed in synthetic sequins (Figure 5d, embedded above), thus discarding the hypothesis that non-A bases may belong to imperfect trimming of adapter sequences.

Minor Comments:

- Depiction of poly(A) tails in Figure 1c, S1c is suboptimal. In S1c tail lengths and frequency are difficult to see.

We apologize if the depiction of polyA tails are suboptimal in Figure 1C and S1C, but the goal of this figure panel was not to quantify the tail lengths (this is done in Figure 2) but rather to illustrate that the tails of diverse lengths (including deadenylated mRNAs) are captured using Nano3P-seq.

- For figure 1C the authors say "In addition, our results confirmed that polyA tail length information was retained in individual reads. Specifically, the majority of reads corresponding to mRNAs had polyA tails". 1C suggests that it is barely a majority: many reads have no green PolyA. This example plants more doubts than it helps the reader.

We are not sure we understand the reviewer's concern on this point. As shown in the picture below (Figure 1C), most of the reads (95 out of 110 reads) contain a polyA tail (green region). We believe that the reviewer might have missed the shorter tail lengths in this figure panel, which we embed below for the reviewer's convenience.

- Figure 1D lacks detail. Is correlation given before or after log-transformation?

The correlation is reported after log-transformation. This has now been clarified in the figure legend.

- Fig 2B: What are the correlations between the time points

We have now included a correlation matrix depicting the correlations between the time points shown in Figure 2B as well as across replicates in Figure S2d. This panel is included below for the reviewer's convenience.

- Figure 3a: The material used here is not obvious. Is it the same as in Figures S1a-c?

Yes, it is the same material as shown in Figures S1A-C. We have now clarified this in the Figure 3a legend.

- References 27 and 28 seem identical

We thank the reviewer for pointing this out to us. This was an error due to the automatic bibliography system that for some strange reason, considered the two references different. This has now been corrected in our revised manuscript.

- Figure 4a only shows one example of differences conserved across time points, but the main text says "often conserved across the different time points analyzed (Fig 4a)"

We now refer also to a new supplementary table in the text (**Table S5**), which includes the median tail lengths for each isoform and for all time points. We now also provide a new table in which we show which transcripts significantly change their polyA tail length during the MZT (**Table S6**). In addition, we now provide additional examples in the main Figure (**Figure 4c**)

- Gene and isoform percentages in the text for Figure 4 require confidence intervals and exact numbers.

Confidence values to these percentages have now been added in the text.

- Figure 5D is barely mentioned in the text

We have now significantly extended the analysis of tail composition (Figure 5), both in terms of description of results in the main text as well as in the form of additional Figure panels and Supplementary Material.

Reviewer #2:

Remarks to the Author:

SUMMARY

This paper presents a strategy for analyzing RNA expression using nanopore sequencing of first-strand cDNA copies primed by strand switching. A strength is detection and length estimates of 3' homopolymers, and a relatively straightforward library preparation scheme. The paper is casually written and not ready for careful consideration in its current form. Some essential data are missing, and some of the data are not presented in a transparent manner. This is surprising given the substantial experience of some of the co-authors. In the text that follows we highlight key technical and logical issues, and then we note specific issues line-by-line.

We thank the reviewer for his/her feedback and time in reviewing our work. We agree with the reviewer that Nano3P-seq has the strengths of detecting and estimating length of 3' homopolymers as well as a relatively straightforward library preparation scheme. We apologize if the reviewer found some parts of the manuscript to be casually written, to this end we have now carefully edited the manuscript text as well as have had a professional scientific editorial company review and edit the text, to enhance its clarity and writing style. With regards to the missing data and presentation of the results, we have now added all clarifications and edits following the reviewer's comments. We thank the reviewer for his/her time and efforts in helping us improve the quality of our manuscript.

GENERAL COMMENTS

Title and Abstract

1. *It would be informative to include 'cDNA' in the title and abstract. It was unclear to us until well into the introduction that the new technique was based on first strand cDNA rather than native RNA.

We agree with the reviewer's comment, and have now added the word "cDNA " in the abstract, to ensure this is clear to all readers.

2. *Here and elsewhere the authors use the term 'translatability'. Possibly this is now used in the molecular biology community, but a more traditional term is 'translational efficiency'. In either case, for a general audience it would be useful to precisely define what you mean.

The term 'translatability' has now been changed for 'translational efficiency'.

3. *In the last sentence of the abstract, the authors assert that Nano3P-seq can accurately estimate transcript levels. It is unclear if this is at the gene level or at the isoform level. In the last sentence of the abstract, the authors assert that Nano3P-seq can accurately estimate tail lengths of full-length transcripts. As far as we can tell, Nano3P-seq does not establish unequivocally that reads are full-length.

We now clarify in the abstract that Nano3P-seq can accurately estimate tail length at per-isoform level, which now reads: *"Overall, Nano3P-seq is a simple and robust method for accurately estimating transcript levels, tail lengths, and tail composition heterogeneity in individual reads, with minimal library preparation biases, both in the coding and non-coding transcriptome."*

With regards to the second comment of the reviewer, we have now removed the word "full length" from the sentence, as we agree it could be misleading.

Results

4. *The current draft does not adequately address performance of the Nano3P-seq strategy. Most notably, in the main text there are no data for throughput, and as best we can tell, there are no data for base call accuracy or read length distribution anywhere in the manuscript. These data are important for a fair comparison between direct RNA sequencing and Nano3P-seq (see Discussion section comments below). It is known that cDNA basecall accuracy is typically better than direct RNA basecall accuracy. So why not point this out in the text along with some data?

We thank the reviewer for his/her suggestions. We now added information regarding the throughput, base-calling accuracy and read length distribution for both Nano3P-seq and dRNA-seq in the Discussion section. These results are also available in **Figure S7** and **Table S11** (which we also include below for the reviewer's convenience). Overall, we find that Nano3P-seq produces more sequencing throughput compared to dRNAseq for matching runs (same input type from same species), longer reads (**Figure S7f**) and similar base-calling accuracies (**Figure S7g**).

Throughput (Table S11) - only zebrafish runs sequenced in MinION flowcells are included below:

We should note that direct RNA sequencing runs used here show fewer reads (250-750K) than what is typically expected based on ONT's website (1-2M reads), but we believe that this is related to the specific input samples used in these experiments, and that same actually applies here to Nano3P-seq (we have done other runs in other organisms such as E.coli afterwards, obtaining almost 5M reads).

Read length distribution (Figure S7f) :

Basecall accuracy (Figure S7g) :

5. *The read count log plots are not up to NMeth standards and will be difficult for readers to assess. These include panels Fig 1.D, Fig 2.B, and Fig S1.H. Let's examine Figure 1.D as an example:

To build the figures that the reviewer mentions above, we had previously used normalized read counts (i.e. counts that had been scaled/normalized across samples and replicates to have equal coverage across biological replicates). However, we agree with the reviewer that this scaling might make it difficult for readers to assess certain aspects of the performance of Nano3P-seq such as throughput. Therefore, we have now rebuilt these figures using log (read counts) and log (RPM), instead of log(normalized read counts). We thank the reviewer for pointing us to this issue. We embed below the new versions of these figure panels for the reviewer's convenience:

Updated Figure 1D:

Updated Figure 2B:

Updated Figure S1H (New S1I) :

5. *The read count log plots are not up to NMeth standards and will be difficult for readers to assess. These include panels Fig 1.D, Fig 2.B, and Fig S1.H. Let's examine Figure 1.D as an example:

To build the figures that the reviewer mentions above, we had previously used normalized read counts (i.e. counts that had been scaled/normalized across samples and replicates to have equal coverage across biological replicates). However, we agree with the reviewer that this scaling might make it difficult for readers to assess certain aspects of the performance of Nano3P-seq such as throughput. Therefore, we have now rebuilt these figures using log (read counts) and log (RPM), instead of log(normalized read counts). We thank the reviewer for pointing us to this issue. We embed below the new versions of these figure panels for the reviewer's convenience:

Updated Figure 1D:

Updated Figure 2B:

Updated Figure S1H (New S11) :

i) The X axis is labelled 'Log (Expected Counts)'. What is meant by expected counts in this context and how were these values derived? This is not defined in the panel, text, or Methods as far as we could see. Why not use the X-axis units as in the original Sequins manuscript, i.e. Log(10) Attomoles/uL? It is noteworthy that several of the authors on this paper were co-authors on the carefully-executed Sequin (2016) paper.

We have now replaced log(Expected counts) for log₁₀(attomoles/uL). Please see updated **Figure 1d** embedded in the previous response.

ii) It is unclear to us if the data in this panel are for genes or isoforms. In the Sequins(2016) paper, the authors presented Counts vs Attomoles/uL for genes, isoforms, and exons. This trio of plots would be a useful addition to the Nano3P-seq paper.

We have now included plots for both genes (**Figure 1d**) and isoforms (**Figure S1h**). Please see both figure panels embedded below.

Figure 1d (per-gene):

Figure S1h (per-transcript):

iii) To facilitate the reader's examination of these data, please plot untransformed data on Log(10)-Log(10) plots with tick marks. Once again, the original Sequins manuscript can serve as a guide.

The revised figure panels now show log₁₀-log₁₀ axis as suggested. Please see Figure 1d and S1h embedded above.

iv) Please include the slope value for the fitted line. A naive examination of the original Fig 1D might suggest a 1-to-1 correspondence between expected and observed values which appears not to be the case.

This information has now been included in the figure legend.

v) In the Sequins (2016) paper, the relationship between counts and concentration was flat below 1 attomole/uL. The paper under review does not use concentration on the X axis so it is difficult to make a comparison. We are curious if any of the Sequin RNA isoforms were not observed in these control experiments.

Please see above. We have now modified the axis of these plots to enhance their clarity.

With regards to the results, there are some RNA sequin isoforms that are not detected by Nano3P-seq (see Figure S1h embedded above). We should note that sequins were sequenced as spike-ins, not as a full dataset in a flow cell on its own.

vi) It is customary to use axis values beyond the upper and lower limits of plotted data. Please fix this.

In our resubmitted version the axis values go beyond upper and lower limits of plotted data in all panels of all main and supplementary figures.

vii) In Figure 2B the scales on the plots do not bound the data. Fix this. Also, what was the logic for using natural log? Was this the best way to convey the information to a reader – is that the explanation? Also, the highest density in the distribution is not bound by the axes. Please fix this.

This has now been fixed. We also changed the natural log for log₁₀. We embed below the new versions of these plots for the reviewer's convenience.

6. *Line 209: section title. The authors claim that tail length can be accurately estimated using Nano3P-seq. Please explain to the reader what sort of precision they should anticipate and whether or not that precision is tail length dependent. Also, in Figure S3B, there appears to be a reasonable correlation between the two replicates in the range of 30-60 nt, but that they both overestimate tail lengths at 30 and 60 nucleotides. This is not a big issue, but please be forthright about what the technique delivers.

To tackle this concern, the precision of tail length estimation is now assessed with new additional synthetic oligonucleotides containing a broad range of known polyA tail lengths (0,15,30,60,90,120) (Figure 3b). We find that our method accurately estimates polyA tail lengths in a broad range of tail lengths (Figure 3c, also embedded above).

Figure 3b:

Figure 3c:

We should note that the reason for choosing cDNA standards instead of RNA ones is that there was no company that would synthesize such long polyA tails in RNA form. However, since Nano3P-seq is a cDNA-based method, these standards can be used to illustrate the accuracy of our method in predicting polyA tail lengths in cDNA products.

7. *Figure 3G: Are the values for transformed or untransformed data?

These were transformed (normalized) values. We now use log₁₀(RPM) values in this figure panel. This figure is now renumbered to Figure 3H (please see below):

8. The paper claims isoform documentation. However, Nano3P-seq does not unambiguously establish that the biological 5 prime end of transcripts have been sequenced. Please defend this position. Related (as pointed out above) there is no documentation of the read length distribution for this method.

The reviewer is correct with regards to the fact that in nanopore sequencing datasets (including Nano-3Pseq) there will be a 3' sequencing bias, i.e. with decreased coverage of the 5' ends of the molecules, as can be seen in several figure panels (e.g. **Figure 1c**, **Figure 4c**, **Figure S1e-g**). However, in case a given read cannot be unambiguously assigned to an isoform, the read is discarded from the per-isoform analysis. Specifically, the assignment of read-to-isoform was done using IsoQuant (<https://github.com/ablab/IsoQuant>), a third-party software as described in the *Methods* section, with recommended parameters for nanopore data. The assignment of the isoforms done by IsoQuant is as follows:

- If the read intron chain matches a single known isoform, it is reported as unique. Even if the 5' is degraded, the read may still contain enough information to unambiguously decide which isoform it is;
- If it's degraded such that multiple isoforms match, read is reported as ambiguous;
- If the intron chain matches to none of the isoforms from the annotation it is reported as inconsistent (potentially novel isoform or misalignment).

With regards to the second comment, we now include read length distributions for the method in the form of additional supplementary figures (see also response to point #4), in comparison to the read length distributions of the matching dRNA-seq sample (4hpf) (**Figure S7f**, embedded below)

DETAILED COMMENTS

9. Line 73. Try making the sentence more clear by bringing the subject and verb together.

We have now slightly rephrased the sentence, and hope it will now be clearer.

10. Line 96. Others have extended the 3 prime ends of RNA strands using poly(I) followed by dRNA sequencing. Please see Vo et al RNA 2021. 27: 1497-1511) and Drexler et al Molecular Cell Volume 77: 985-998 (2020).

We have now mentioned and cited these works in the Introduction. We thank the reviewer for his/her suggestion.

11. Lines 101-104. Please reconcile these assertions with the preceding comment.

This has now been fixed in the resubmitted version.

12. Line 121. Please revisit this statement in light of Vo et al. above.

The statement and paragraph has now been rephased in the context of the Vo et al. and Drexler et al. works.

13. Line 153. Interpreting data in figure legends (as was done here) is usually discouraged. We are not sure of NMeth's policy on this.

We now edited the figure legend according to the NMeth's policy.

14. Line 163. Please see discussion of Figure 1D in the general comments section.

We are not sure what the reviewer is referring to with this comment. We have now edited Figure 1d following the reviewer's suggestions in his/her previous comments.

15. Line 177. 'Drastic' is an odd word choice here. 'Drastic' measures are not just extreme, they are likely to have harmful side-effects. Better to use a neutral word.

We agree with the reviewer, and have now used a more neutral term.

16. Line 191. Polyadenylation of Mt 16s rRNA is also observed in human samples (see citation 12 in this draft paper).

We have now edited this paragraph in order to mention other studies:

"In agreement with this observation, we found that polyA-tailed 16S mitochondrial rRNAs were present not only in zebrafish (Figure S2g), but also in mouse (Figure S2h,i), suggesting that this feature is conserved across species and not a sequencing artifact, in agreement with previous reports^{22,23}. " (reference #23 is in human samples)

17. Line 220. Here the fit is r^2 , but in Figure S3B it is r . Which is it? This lack of care in data presentation is concerning.

We apologize for this typo, this has now been corrected in the plot. The fit is now R throughout the paper in order to keep the consistency across figure panels.

18. Line 258 (E). Awkward sentence.

We have now edited this sentence.

19. Line 289. Here the advantage of nanopore read lengths is highlighted, but nowhere do the authors document the read length distribution of their technique. This needs to be included.

We now include the read length distribution of our technique compared to dRNA-seq (**Figure S7f**). Please see the figure embedded below. Using the same input data, the median read length of Nano3P-seq was 1307nt, whereas the median read length of dRNAseq was 907nt.

20. Line 310. The authors make a general statement '.....it demonstrates that Nano3P-seq can provide transcriptome-wide measurements of the polyadenylation status of diverse biological samples'. This is based on a few examples. While this may one day prove to be the case, claiming transcriptome-wide measurements is premature based on this paper.

We respectfully disagree with the reviewer on this comment. In our manuscript, we show that Nano3P-seq is applicable both in *in vitro* samples (curlicakes, sequins) as well as *in vivo*. Specifically, we show its applicability transcriptome-wide in 3 different species (zebrafish, yeast, mouse), across different developmental stages (2,4,6hpf) in zebrafish, and across biological replicates. In addition, we now also include additional synthetic constructs (cDNA) which span a broader range of polyA tail lengths. Therefore, we believe that we have demonstrated that Nano3P-seq can provide accurate and reproducible transcriptome-wide measurements of polyadenylation status across biological samples.

21. Line 401. Not an accurate statement. See Vo paper and Drexler paper cited above.

We have now changed this statement according to these papers as pasted below :

"Nanopore dRNA-seq has been proposed as an alternative long-read sequencing technology for studying polyA tail lengths^{12,13}; however, the standard dRNA-seq approach (Figure 1a) cannot capture deadenylated RNAs, molecules with non-canonical tailings (e.g., polyuridine), or molecules with polyA tails shorter than 10 nt, thus biasing the view of the transcriptome toward polyadenylated molecules. Customized dRNA-seq methods involving in vitro poly(G/I)-tailing have been developed to overcome some of these limitations, but a lack of bioinformatic tools to distinguish polyI and polyA signals limits their applicability to study polyA tail length differences across transcripts in these datasets^{14,15}. In addition, dRNA-seq requires 500 ng of RNA as input, whereas Nano3P-seq requires as little as 50 ng, thus decreasing the required input material by 10-fold. "

Reviewer #3:

Remarks to the Author:

This manuscript reports the development of a new method, Nano3P-seq, which uses Oxford Nanopore sequencing to provide information on RNA expression level, poly(A) tail composition, and poly(A) tail length. This does not rely on PCR and can also capture non-poly(A) transcripts. The authors apply their new method to some relevant samples and confirm findings generally known about poly(A) tails. This application part is well done and a strength of the manuscript. More generally, there are many aspects of biology where this method could be used. Unfortunately, this manuscript falls short in two related areas – (1) how accurate is the method for poly(A) composition and length and (2) how does it compare directly to other methods for measuring poly(A) tails.

We thank the reviewer for his/her comments, and agree that the method could be used for many different biological aspects and questions. We also agree with the fact that additional benchmarking would be beneficial for the paper. To this end, in addition to using the sequins to assess the quantitative ability of Nano3P-seq to predict polyA tail lengths, we now include:

- (i) **a set of synthetic sequences with a wide range of known polyA tail lengths** to further assess the **accuracy of Nano3P-seq in the estimation of polyA tail lengths**. In our previous version, we were using the 'Sequins' (which contain either 30nt or 60nt long tails) to assess the quantitative ability of Nano3P-seq to estimate polyA tail lengths. In our revised version, we have now also sequenced a set of novel synthetic sequences with a broader range of known polyA tail lengths (0, 15, 30, 60, 90 and 120 nt).
- (ii) **a set of synthetic sequences** to assess the ability of Nano3P-seq to **accurately identify and quantify tail composition variations**. Specifically, we have now included synthetic sequences which include diverse tail end compositions (e.g. AAAAU,AAAAC,AAAAG,AAAAA), with diverse lengths (e.g. U, UUU, UUUUU) as well as sequences with internal Gs.

Therefore, we believe that the new resubmitted version addresses the two points mentioned above:

- With respect to (1), i.e. how accurate is the method for polyA composition and length, these two points are now addressed with these two new datasets mentioned above (in addition to the 'sequins', which were also included in the previous version). The details can be found in response to comment #1 below.
- With respect to (2), i.e., how our method compares to other methods measuring polyA tail lengths, in our previous submission we had already compared Nano3P-seq to PAL-seq results (**Figure 3i,j**, also embedded below for the reviewer's convenience). We now also compare Nano3P-seq to direct RNA nanopore sequencing (dRNA-seq) (**Figure S7d-g**, also embedded below). Our results show that Nano3P-seq polyA tail estimations correlate well with both PAL-seq (Pearson R=0.71-0.85) and dRNA-seq (Pearson's R= 0.94).

Figure S3: Nano3P-seq vs PAL-seq

Figure S7: Nano3P-seq vs dRNA-seq

Major concerns.

1) The manuscript needs to be more rigorous in showing that Nano3P-seq is accurate in measuring poly(A) tail composition and length.

Figure S1 shows that poly(A) tail length can be measured but given that the initial tail length was not carefully characterized, it is not possible to assess accuracy in the sequence data. In Figures 3B, S3C, and S3D, the accuracy of the tail length is measured for the Sequins with known lengths of 30 and 60 bases. That is a good start, but it's important to test a wider range of lengths. Moreover, the authors mention that the measured length with *tailfindR* was off by 15 bases and they adjusted all measurements accordingly (lines 646-652). It was not adequately explained how they know that this is due to the expectation of a double-stranded cDNA by *tailfindR*. Why would this be true for both 30 and 60 base tails?

We agree with the reviewer that poly(A) tail length should be measured in a broader range than 30 and 60bp (i.e. the sequins). To this end, we have now sequenced a new set of synthetic sequences (in addition to the sequins, already included in the manuscript) which contain poly(A) tails of diverse known lengths (0, 15, 30, 60, 90 and 120 nt) (Figure 3b). We find that our method accurately estimates poly(A) tail lengths in a broad range of tail lengths (Figure 3c, also embedded below).

Figure 3b:

Figure 3c:

Finally, with regards to the 15nt adjustment done in the previous version of the manuscript, we had hypothesized that the need of this adjustment could be due to the single strand nature of our Nano3P-seq libraries, compared to the expected double stranded cDNA that *tailfindR* expects. However, we now believe that this was not the reason for this offset. During the preparation of the resubmission of this work, we have collaborated with the authors of *tailfindR* tool that we used for tail length estimation, and we realized that the 15nt overestimation problem was in fact due to using a different adapter than the direct cDNA method (the adapter of Nano3Pseq is in fact not the same as in direct cDNA sequencing,

causing the offset). When this was changed in the code, this 'correction' was no further needed. We have now we created a Nano3P-seq branch in the Github repository of the tailfindR tool (<https://github.com/adnaniazi/tailfindr/tree/nano3p-seq>), which is written specifically for Nano3P-seq data analysis, which does not require any adjustments of polyA tail length predictions by the future users after the polyA tail prediction has been made by the tool. This branch is also provided as a supplementary file (File S2) in the resubmitted version of the manuscript.

For poly(A) tail composition, the manuscript does not present any data on the error rate for these sequencing runs, the base quality in the poly(A) tail, nor the accuracy of the bases with spike-ins of known composition. Even without PCR, it's known that reverse transcription and nanopore sequencing produce errors. Error rates should be shown both for base substitutions and insertion/deletions. The authors focus on terminal U residues (Figures 5C and 5D), but also show other bases are present in the last 10 residues (Figure S7A). They should also explore bases along the entire length of the poly(A) tail with known controls to see whether non-A bases in other positions can be accurately measured and studied.

To address this point, we now include the following information:

1. We now provide quantitative measures of non-A base compositions in the tail regions of zebrafish mRNAs (Figure 5D, S6A and Table S8,S10). Moreover, we perform the same identical analysis in the 'sequins', which provide useful controls to assess the 'background error' coming from the reverse transcription and nanopore sequencing. We find that errors in basecalling accuracy in polyA tails are mainly in the form of insertions or deletions in the homopolymeric regions (e.g. 8As are predicted instead of 9As). However, what we are actually analyzing are non-A bases, and in these bases, the base-calling error is actually low (2.4% of the sequin tails contained non-A bases in some form). By contrast, we find that 35%, 31%, and 16% of the zebrafish reads (at 2, 4, and 6 hpf, respectively) contained non-A bases in their tails (Figure 5D, embedded below). We believe that these results outrule sequencing or base-calling artifacts as a major source of the base composition variation observed in *in vivo* zebrafish samples using Nano3P-seq, which we understand was the concern of the reviewer.

Figure 5d:

2. In addition, we have now sequenced a new set of synthetic sequences that will contain tails of diverse known lengths and compositions, including polyA polyU, polyC and polyG tails of diverse lengths (Figure 5a-c and S5a-b), to assess the accuracy of our approach in detecting known variations in the reads.

Figure 5a-c:

Figure S5a,b:

2) One of the most important aspects of a manuscript introducing a new method is to demonstrate how it compares to existing technologies. In the Introduction, the authors mention that PAL-seq and TAIL-seq are limited because they use Illumina short reads and rely on PCR. There, they also mention the dRNA-seq nanopore method is not able to sequence non-poly(A) RNA and their tails. Note that dRNA-seq does not rely on PCR. In the Discussion, the authors mention PacBio-based methods, FLAM-seq and PALso-seq, which provide all the information that Nano3P provides for poly(A)+ transcripts, but with a PCR amplification step. The reads are more expensive, though this may not be a major factor in such experiments and details are not provided. While the PacBio sequencers are expensive, the authors should mention that there are many facilities that will sequence samples submitted by any scientist. If Nano3P-seq provides a meaningful advance for the field, there should be a direct comparison with one of the best Illumina and one of the best PacBio methods with aliquots from the exact same sample. The authors do compare to dRNA-seq in Figures 2F, 3B, S3C, S3D. Additionally, they compare PAL-

seq to Nano3P-seq in Figure S4A and S4B, but these are published data derived from a different sample for PAL-seq. There are clearly differences too, so that it is not clear whether they are meaningful, nor which method is more accurate. If there are PCR biases that are not present in this new method, the authors need to show that. And the best way to do this is with a set of spike-in controls, including the Sequins, but also ones with different known poly(A) tail lengths and compositions (as in the PAL-seq paper, ref. 44, but not necessarily limited to what is done in that paper).

We fully agree with the reviewer that introducing a new method requires demonstration of how it compares to existing technologies. We would like to note that in our opinion, however, we had already included comparison to two different orthogonal methods (dRNA-seq and PAL-seq):

- i) comparison of polyA tail length estimates of Nano3P-seq to dRNA-seq, in synthetic sequins (Figure S3C-E, embedded below).
- ii) comparison of Nano3P-seq to dRNA-seq, in vivo biological zebrafish samples (Figure S7D-E embedded below).
- iii) comparison of Nano3P-seq to PAL-seq, in *in vivo* zebrafish samples at t=2, 4, 6hpf (Figure S3I-J, embedded below).

Figure S3C-E:

Figure S7D-E:

Figure S3I-J:

With regards to PacBio-based methods, we find that Nano3P-seq overcomes some of the limitations that have been stated by the own authors developing the method(s). Citing the FLAM-seq paper: *“Thanks to TAIL-seq, a number of terminal modifications of poly(A) tails were reported to play a role in mRNA decay. However, TAIL-seq cannot be used to detect tail modifications within the vast majority of tail nucleotides as Illumina sequencing quality strongly deteriorates in homopolymeric stretches. With FLAM-seq, instead, we can sequence through the entire tails, yet we cannot unambiguously identify 3' terminal modifications.”*

In Nano3P-seq, we find that can identify 3' terminal modifications, as shown in Figure 5b and S5a (embedded below), where the ground truth is known (synthetic sequences).

Minor concerns.

1) The authors should mention how the input RNA amount for this protocol compares to other protocols because this can be a key experimental design feature. In the Methods section, they start with 100 ng of total RNA, but what is the lower limit for this method? This could be an issue given that this method does not use PCR.

Nano3P-seq libraries use 50-100ng of input RNA material per sample. We have not systematically tested decreasing input amounts, as we reasoned that the input amounts required for efficient library preparation should be similar to those of direct cDNA library preparations, where there is no amplification of the cDNA, in a similar fashion to Nano3P-seq libraries. By contrast, direct RNA sequencing libraries use 500ng of input material. Thus, Nano3P-seq provides a similar (or slightly improved) throughput of sequenced reads as direct RNA sequencing (~0.5-2 million reads per flowcell, see **Table S11**) but requires 5-10X less input RNA material, compared to direct RNA sequencing. We now discuss this point in the Discussion section.

2) While the authors show nicely that the RNA expression of the Sequins is accurate, what about looking at biases such as 5' vs. 3' ends of the transcript, GC content, and transcript length?

We have now addressed some of these aspects in the supplementary material, including Nano3P-seq transcript (read) length in **Figure S7f**, and base-calling accuracy in **Figure S7g**, relative to dRNA-seq.

3) Line 180-181: What are miscRNAs and scaRNAs?

The MiscRNA category includes diverse RNA types. According to ENSEMBL, the definition of miscRNA is: "Miscellaneous RNA. A non-coding RNA that cannot be classified". ScaRNAs correspond to "small cajal-body specific RNAs". We have now clarified this in the legend of **Figure 1**.

4) Lines 185-191: Although the authors found poly(A) tails on 16S mitochondrial rRNA in both zebrafish and mouse, the conservation of this finding does not fully validate it as correct. The text could be modified to clarify this point.

We further validated this observation with other methods, i.e. qPCR and Sanger sequencing (**Figure S2i,j** embedded below). In addition, we now included additional replicates supporting these observations.

Figure S2i,j:

5) Box plot features should be explained in Figures 3B,D,G,H and elsewhere.

We thank the reviewer for pointing us to this issue. This has now been clarified in the figure legends.

6) Lines 296-297: What is the justification for a threshold of 10 reads per isoform?

We chose 10 reads per isoform as we reasoned that the number is large enough to ensure that the sampling will be representative of the whole population, while maximizing the number of genes or isoforms that will be kept for downstream analyses. We should note that we initially performed this analysis with 30 reads as a threshold, but that decreased the number of isoforms kept in the analyses, and the conclusions did not differ from those obtained using 30 reads coverage. Many bioinformatic softwares and analyses in nanopore sequencing data require a minimum of 5-50 reads coverage to support a given site, isoform, modification, etc. In this case, we require a minimum of 10 reads coverage to include the isoform in downstream analyses.

7) Line 343: Perhaps change to "modification" (singular) as there is only one type assessed here.

This has now been changed in the revised version.

8) Line 407: TGIRT should be spelled out and the explanation for why it is better should be clarified.

Following the reviewer's suggestion, we have now added the spelling of TGIRT. The paragraph now reads as follows:

"Moreover, the use of TGIRT (Thermostable group II intron reverse transcriptase) in the Nano3P-seq protocol not only maximizes the production of full-length cDNAs, but also ensures the inclusion of RNA molecules that are highly structured and/or modified, which would often not be captured -or their representation would be significantly biased- using standard viral reverse transcriptases."

9) Lines 426-439: Perhaps this should be in the Result section?

We thank the reviewer for his/her suggestion; however, we consider that it is best to keep this in the Discussion, because the biases caused by polyA selection or ribodepletion are not specifically a result of our Nano3P-seq analysis, but rather is a side observation. When we tried to relocate this paragraph into the Results section, we were unable to find an appropriate spot that would not disrupt the narrative, as it is not directly related to any of the result sections. Therefore, we think that it is better to leave these sentences in the Discussion, rather than as a separate unrelated section in the Results section.

10) Line 454: If the authors state their method is "low cost," they should present information on the cost of reagents for all steps in this method.

We kindly disagree with the reviewer on this point, as we think that it is not needed to show the costs of reagents for all steps of the method to claim that it is low-cost, because most of the steps are identical to direct cDNA nanopore sequencing, and previous literature has already performed such comparison of costs of the different sequencing methods. Therefore, to address the reviewer's concern, we now cite previous works that already showed and/or compared PacBio, Illumina and Nanopore sequencing. Please see the updated paragraph below:

"These features set Nano3P-seq as a potent, low-cost (Cui et al. 2020) method that can provide mechanistic insights into the regulation of RNA molecules and improve our understanding of mRNA tailing processes and post-transcriptional control."

11) Experimental details are missing in the Materials and Methods section.

All the additional experimental details mentioned below by the reviewer have now been amended in the Methods section.

- a. Line 481: Sodium Acetate concentration?
- b. Line 485: Turbo DNase details (concentration, time, temperature) missing
- c. Line 489: How were the mice cared for (feeding, day/night schedule, etc.)? Were the mice male or female?
- d. Line 493: Details about "Protease Inhibitor" missing.
- e. Line 494: Details about "dounce" (# of times, size) missing.
- f. Line 503: Details about "RNase Inhibitor" missing.
- g. Line 521: Details about which "oligo (dT) magnetic beads" missing.
- h. Line 532: What is "bead resuspension"? Is this from the Ribodepletion kit or the Dynabeads kit?
- i. Line 533: What is "bead resuspension buffer"? Same as above.
- j. Line 536: What is "bead wash buffer"? Same as above.
- k. Line 537: What is "depletion buffer"? Same as above.
- l. Lines 570, 572, 573: What is the source of the buffers, Adapter Mix, ABB Buffer, Elution Buffer, Sequencing Buffer, and Loading Buffer? All from Oxford Nanopore?

Yes, all the buffers are from Oxford Nanopore, as part of the library preparation kit that is described and used as part of the Nano3P-seq library.

In addition, we provide a supplementary file (**File S1**) that includes a step-by-step detailed protocol to build Nano3P-seq libraries.

m. Line 574: Please provide details on the length of the runs and whether any library reloading was done.

Library reloading was not done. The duration of the runs has now been included as an additional column in **Table S11**.

n. Line 663: What about the care of the mice?

We have now added a statement for the mice samples used in the Animal Ethics Statement. We thank the reviewer for highlighting this, which was missing in our previous submission.

o. Figure S1: Perhaps include an explanation for the term "curlcake" and refer the reader to the Materials and Methods for details on CC1 and CC2.

We have now referred to the Methods in **Figure S1** legend as suggested by the reviewer.

Decision Letter, second revision:

Dear Eva,

Thank you for your letter detailing how you would respond to the reviewer#3's concerns regarding your Article, "Nano3P-seq: transcriptome-wide analysis of gene expression and tail dynamics using end-capture nanopore cDNA sequencing". We have decided to invite you to revise your manuscript as you have outlined, before we reach a final decision on publication. We encourage you to perform Nano3P-seq on HeLa cells, which would provide a direct comparison with FLAM-seq.

* include a point-by-point response to the reviewers and to any editorial suggestions

* please underline/highlight any additions to the text or areas with other significant changes to facilitate review of the revised manuscript

- * address the points listed described below to conform to our open science requirements
- * ensure it complies with our general format requirements as set out in our guide to authors at www.nature.com/naturemethods
- * resubmit all the necessary files electronically by using the link below to access your home page

[Redacted] This URL links to your confidential home page and associated information about manuscripts you may have submitted, or that you are reviewing for us. If you wish to forward this email to co-authors, please delete the link to your homepage.

We hope to receive your revised paper within 4 weeks. If you cannot send it within this time, please let us know. In this event, we will still be happy to reconsider your paper at a later date so long as nothing similar has been accepted for publication at Nature Methods or published elsewhere.

OPEN SCIENCE REQUIREMENTS

REPORTING SUMMARY AND EDITORIAL POLICY CHECKLISTS

IMAGE INTEGRITY

DATA AVAILABILITY

Please include a “Data availability” subsection in the Online Methods. This section should inform readers about the availability of the data used to support the conclusions of your study, including accession codes to public repositories, references to source data that may be published alongside the paper, unique identifiers such as URLs to data repository entries, or data set DOIs, and any other statement about data availability. At a minimum, you should include the following statement: “The data that support the findings of this study are available from the corresponding author upon request”, describing which data is available upon request and mentioning any restrictions on availability. If DOIs are provided, please include these in the Reference list (authors, title, publisher (repository name), identifier, year). For more guidance on how to write this section please see: <http://www.nature.com/authors/policies/data/data-availability-statements-data-citations.pdf>

CODE AVAILABILITY

Please include a “Code Availability” subsection in the Online Methods which details how your custom code is made available. Only in rare cases (where code is not central to the main conclusions of the paper) is the statement “available upon request” allowed (and reasons should be specified).

SUPPLEMENTARY PROTOCOL

To help facilitate reproducibility and uptake of your method, we ask you to prepare a step-by-step Supplementary Protocol for the method described in this paper. We [encourage authors to share their step-by-step experimental protocols](https://www.nature.com/nature-research/editorial-policies/reporting-standards#protocols) on a protocol sharing platform of their choice and report the protocol DOI in the reference list. Nature Research's Protocol Exchange is a free-to-use and open resource for protocols; protocols deposited in Protocol Exchange are citable and can be linked from the published article. More details can found at www.nature.com/protocolexchange/about.

ORCID

Best regards,
Lei

Lei Tang, Ph.D.
Senior Editor
Nature Methods

Reviewers' Comments:

Reviewer #1:

Remarks to the Author:

Begik et al have revised their manuscript with extensive validation. These experiments overall support the biological conclusions of the manuscript. Perfect measurements are hard to prove for sequencing, but validations support the quantification of polyadenylated and non-polyadenylated 16S mitochondrial rRNA, determination of m1acp3Y in rRNAs and tail-length estimations. This could create interest among polyA-tail and ribosomal RNA researchers.

The authors also clarified experimental observations regarding different performance of ribodepletion and transcriptional states of rRNAs in zebrafish embryos among others. This makes the manuscript more accessible.

Conversely, the revised figures 1b, 2d and 2f (previously hard to interpret) show that only rRNA and mRNA (in figure 1b also snRNA) receive sufficient read numbers. In the current state mentioning categories, such as snoRNAs, scaRNAs and lncRNAs in the abstract raises expectations that are not met in the text. Authors should restrict the abstract to categories that give strong results.

Reviewer #3:

Remarks to the Author:

This revised manuscript is much improved and addresses many of the concerns raised in the initial review. In particular, the inclusion of known sequence controls spanning a greater range of poly(A) tails and with non-A bases is an excellent addition. Unfortunately, this manuscript still falls short in showing how it compares directly to other methods for measuring poly(A) tails. Even if this new method has some capabilities that previous methods do not, it is important to compare in areas where other methods previously have established benchmarks.

Major concern.

As requested in the initial review, a direct comparison of methods with the same sample is the best way to show how a new method performs relative to existing methods. The authors did not do this and thus fall short of expectations for a new method. Comparing only to another nanopore method and published data from a different sample is not sufficient. The details of such a comparison were listed in the previous review.

Minor concerns.

1) Page 5: For the text ("By contrast, much fewer non-coding RNA populations were globally captured ... (Figure 2e)."), there seems to be a difference between what is written and shown in the figure. The

figure shows that the % of reads for non-coding RNA is lower for dRNA-seq, but it does not address whether the total number of non-coding RNAs detected changes. It would be good to clarify this.

2) Page 5: What is “per-read analysis”?

3) Page 5: The Sanger sequencing (Figure S2j) seems to show only four A bases. Is this correct? Is the poly(A) tail really only four bases or was some lost in processing?

4) Figure 3: Are the rules for all the box plots (panels c, e, h, i) the same? It’s not clear as written.

5) Page 9: For the “analyzed transcripts varied significantly in polyA tail length” would it be possible to list a p-value used for the significance?

6) Page 10: Please add information describing how the authors “assigned reads mapping to SSU rRNAs as either “precursor” or “processed”....” and what the term “mismatch frequency” means.

7) Page 12: The description of the results in Figure S5b and Figure 5c is technically correct, but does not reflect the actual accuracy at each base or position. For example, for the IntG control, there are A bases detected where G bases are present. In an actual experiment, it is likely that there would be a mixture of bases present at a given position, so that it will not be a “yes or no” answer and the fraction correctly detected will matter more. This aspect of the experiment should be reported more clearly.

8) Figure 5c: Would it be possible to provide the actual numbers used to draw those plots?

9) Figure 5d: Is the expected % reads = 100 for all the Sequin controls? And is this also the case for Figure S6a and Tables 8 and 10? This should be explicitly stated some place in the manuscript.

10) Page 15: The authors state that only 50 ng of total RNA is needed for a Nano3P-seq library. It would be good to explicitly show a comparison of the results for 50 vs. 100 ng as input – or at least tell the reader where the 50 ng data can be found in the manuscript.

11) Page 16: Should be “rule out” not “outrule.”

12) Page 16: Citing the other paper for “low cost” is still not sufficient as it’s not clear to which methods this comparison is being made. And it’s not clear if all the costs in the Nano3P-seq method are included in the costs cited by the reference. Better to take out this contention if the authors do not want to include their actual costs.

13) Page 18: What is the concentration of SUPERase In RNase Inhibitor?

14) Page 22: What quantity of RNA was used as input in the dRNA-seq experiments?

15) Page 23: It would be good to state explicitly how the analysis was done for the newly added cDNA controls with different poly(A) tail lengths and non-A bases in the Analysis of Nano3P-seq datasets section (or elsewhere in the Methods section).

16) Figure S1c: Should be “above” and “below” not “left” and “right”

17) Figure S6: mislabeled as Figure S5.

18) Table S11: Additional explanation in text about the problems with some of these samples – 90A and 120A. It was not clear whether these would be problematic in future experiments.

19) Library Preparation Protocol for Nano3P-seq

a. “Materials and consumables required” – are these all of the reagents for entire protocol? Some items are mentioned later, but not listed here.

b. There should be part numbers and sources for each item here.

c. What type of purification is needed for each oligo?

d. RnaseIN Promega – more information needed.

e. AMX Adapter Ligation – source and part numbers?

20) Response to Reviewers: Authors discuss errors in base calling accuracy in polyA tails being mostly insertions and deletions in homopolymeric regions. Where is this shown in the manuscript? It's worth mentioning this specifically in the manuscript.

21) Response to Reviewers: Authors did not look at 5' vs. 3', GC, or length biases for the Sequins and actual RNA samples in their data. These are fairly standard analyses.

Author Rebuttal, second revision:

Reviewers' Comments:

Reviewer #1:

Remarks to the Author:

Begik et al have revised their manuscript with extensive validation. These experiments overall support the biological conclusions of the manuscript. Perfect measurements are hard to prove for sequencing, but validations support the quantification of polyadenylated and non-polyadenylated 16S mitochondrial rRNA, determination of m1acp3Y in rRNAs and tail-length estimations. This could create interest among polyA-tail and ribosomal RNA researchers. The authors also clarified experimental observations regarding different performance of ribodepletion and transcriptional states of rRNAs in zebrafish embryos among others. This makes the manuscript more accessible. Conversely, the revised figures 1b, 2d and 2f (previously hard to interpret) show that only rRNA and mRNA (in figure 1b also snRNA) receive sufficient read numbers. In the current state mentioning categories, such as snoRNAs, scaRNAs and lncRNAs in the abstract raises expectations that are not met in the text. Authors should restrict the abstract to categories that give strong results.

We thank the reviewer for his/her comments.

Following the reviewer's suggestion, in the revised version of our manuscript, we have mentioned only mRNA, rRNA and snRNA in the abstract, which are the 3 most abundant RNA biotypes that we observe in our mouse sample shown in Figure 1b. However, we would like to note that Figure 1b was built from a mouse sample that was sequenced in the same flowcell together with 'sequins' (used in Figures 1d and S1). Therefore, a significant proportion of the flowcell was used to estimate the sequins abundances and polyA tail length estimations, which took most of the coverage of the flowcell (82% of the reads in replicate1, 50% in replicate 2). For this reason, in our opinion it is not that Nano3P-seq cannot capture these additional RNA biotypes (e.g. lncRNAs, snoRNAs or scaRNAs), as we show that it does capture them, but that the sample does not need to include such a high proportion of internal standards 'sequins'. Here, we included a higher proportion as we were also interested in benchmarking the

performance of Nano3P-seq in the 'sequins' themselves (e.g. Figure 1d). In addition, a given sample could also be sequenced with deeper coverage (e.g. PromethION run instead of MinION run) or enriched in whatever RNAs are of interest to the future researcher. Similarly, improved ribodepletion would also increase the relative abundance of less frequent RNA biotypes when using Nano3P-seq, without requiring more sequencing depth.

Reviewer #3:

Remarks to the Author:

This revised manuscript is much improved and addresses many of the concerns raised in the initial review. In particular, the inclusion of known sequence controls spanning a greater range of poly(A) tails and with non-A bases is an excellent addition. Unfortunately, this manuscript still falls short in showing how it compares directly to other methods for measuring poly(A) tails. Even if this new method has some capabilities that previous methods do not, it is important to compare in areas where other methods previously have established benchmarks.

Major concern.

As requested in the initial review, a direct comparison of methods with the same sample is the best way to show how a new method performs relative to existing methods. The authors did not do this and thus fall short of expectations for a new method. Comparing only to another nanopore method and published data from a different sample is not sufficient. The details of such a comparison were listed in the previous review.

We thank the reviewer for his/her comments and suggestions. However, we respectfully disagree with the reviewer with regards to the insufficient comparison of our method with other methods for measuring poly(A) tails. As the reviewer mentions above, we had compared our method (Nano3P-seq) to two different orthogonal methods used to estimate polyA tail lengths, namely:

1. Direct RNA nanopore sequencing, using a "matched sample" as the reviewer suggests (i.e. this data is produced as part of this work); and
2. PAL-seq data from a previously published study (Subtelny et al. 2014).

To address the reviewer's concern, we have now included the following additional Nano3P-seq experiments and results in the revised version of the manuscript:

1. We have sequenced a new Nano3P-seq run on human HeLa cells, which allows us to compare the Nano3P-seq method to additional previously published orthogonal methods. Specifically, we can now compare our new Nano3P-seq data on HeLa cells to FLAM-seq, TAIL-seq and PAL-seq. Previously we could not compare our method to these additional orthogonal methods because they had not used the same species/sample types (e.g. FLAM-seq was performed only on HeLa and C.elegans, so we could not

previously compare it to Nano3P-seq). These results are now included in the manuscript in the form of Figure S8a and Table S13.

We find that Nano3P-seq correlates best with FLAM-seq and PAL-seq ($R=0.47$ in both cases), and worse in the case of TAIL-seq ($R=0.19$, likely due to the short-read nature of this orthogonal technique - predictions of polyA tail lengths using TAIL-seq in HeLa samples were never beyond 100nt). On the other hand, we observed that the comparison between the 3 orthogonal methods among each other is lower (0.1-0.31) (Figure S8a, see also Table S13).

2. We have now also included comparisons of Nano3Pseq to previously published yeast PAL-seq and PAT-seq datasets, which were not included in our previous submission (Figure S8b).

Finally, we would like to note that previous works reporting novel methods to determine polyA tail lengths had also used publicly available data to orthogonally validate their methods (i.e. they did not generate data with those orthogonal methods in the same “matched” samples, see table below). Establishing these orthogonal methods (e.g. TAIL-seq, FLAM-seq) in our laboratory to be able to run side by side both methods on the same “matched” sample would take months of work (these techniques are not currently established in our lab). Moreover, we already included a “matched” sample comparison by comparing dRNA-seq and Nano3P-seq, which are completely different techniques (one sequences cDNA, the other sequences native RNA) even if both are using the same sequencing platform (i.e. nanopore sequencing).

Method	Reference	Comparison to other methods
PAL-seq	Subtelny et al, 2014 Nature	* None, only used synthetic oligo standards with known tail lengths
TAIL-seq	Chang et al, 2014 Mol Cell	* PAT assay for some genes (not transcriptome-wide)
FLAM-seq	Legnini et al, 2019 Nat Methods	* Comparison to TAIL-seq (PUBLIC DATA) * Comparison to PAL-seq (PUBLIC DATA)
PAIso-seq	Liu et al, 2019 Nat Comm	* Comparison to TAIL-seq (PUBLIC DATA)
PAT-seq	Harrison et al, 2015 RNA	* Comparison to PAL-seq (PUBLIC DATA)

dRNA-seq	Workman et al, 2019 Nat Methods	* None, only used synthetic oligo standards with known tail lengths
----------	---	---

We hope that the inclusion of these additional experiments and comparisons in the revised version of our manuscript will address the reviewer's concerns.

Minor concerns.

1)Page 5: For the text (“By contrast, much fewer non-coding RNA populations were globally captured ... (Figure 2e).”), there seems to be a difference between what is written and shown in the figure. The figure shows that the % of reads for non-coding RNA is lower for dRNA-seq, but it does not address whether the total number of non-coding RNAs detected changes. It would be good to clarify this.

We thank the reviewer for his/her comment, this sentence is now rephrased in order to better reflect what is shown in the figure: “By contrast, much fewer non-coding RNA populations were globally captured (relative to coding RNA populations) when dRNA-seq was applied to the same samples (Figure 2e).”

2)Page 5: What is “per-read analysis”?

Per-read analysis refers to the analysis of the tail lengths populations, regardless of the gene that they are mapping to. Therefore, when building density plots “per-read”, all reads have equal weight (e.g. Figure 3d, left panel).

By contrast, per-gene analysis of tail lengths uses the median pA tail lengths per-gene as input, to then generate the density plot distribution (e.g. Figure 3d, right panel).

3)Page 5: The Sanger sequencing (Figure S2j) seems to show only four A bases. Is this correct? Is the poly(A) tail really only four bases or was some lost in processing?

Although the Sanger sequencing chromatogram (embedded below for the reviewer's convenience) shows 4 bases based on the “consensus base-calling”, the chromatogram shows that there are peaks corresponding to A bases at subsequent positions, although they are smaller than G peaks, with decreasing frequency (and for this reason, the final “call” is a G). We interpret this chromatogram as the result of sequencing an heterogeneous population of polyA tail lengths sequenced using Sanger sequencing, where G/I tailing is incorporated in populations of slightly different pA tail lengths, For this reason, the signal of poly(A) is still present in subsequent

bases (beyond the 4 “called” bases) and its strength decreases with the increase of the G. Therefore, the poly(A) tail lengths of this population is short, but not just 4 bases.

4) Figure 3: Are the rules for all the box plots (panels c, e, h, i) the same? It’s not clear as written.

The rules for all boxplots shown in Figure 3 were mentioned at the end of Figure 3 legend, i.e. :“Boxplot limits are defined by lower (bottom) and upper (top) quartiles. The bar indicates the median, and whiskers indicate +/- 1.5X interquartile range.” In the case of panel e, the individual data points are also shown (dotplot), in addition to the boxplot, as mentioned in the figure legend: “Each dot represents a read”. In the case of panels h and i, violin plots are shown in addition to the boxplots, providing additional information with regards to the distribution of the data. Please let us know if any panel may need further clarification.

5) Page 9: For the “analyzed transcripts varied significantly in polyA tail length” would it be possible to list a p-value used for the significance?

We have now added the p-value used for significance in the main text ($p < 0.05$).

6) Page 10: Please add information describing how the authors “assigned reads mapping to SSU rRNAs as either “precursor” or “processed”....” and what the term “mismatch frequency” means.

We now edited the parts that needed more information:

“To this end, reads mapping to SSU rRNAs were assigned to either “precursor” or “processed” isoforms (based on location of the 3’end of the read) (...).”

And:

“We observed that the mismatch frequency (misincorporations from the reverse transcriptase) at the m1acp3Ψ- modified site was very high in mature rRNAs but not present in pre-rRNAs, suggesting that this modification is only present in mature rRNA populations.”

7) Page 12: The description of the results in Figure S5b and Figure 5c is technically correct, but does not reflect the actual accuracy at each base or position. For example, for the IntG control, there are A bases detected where G bases are present. In an actual experiment, it is likely that there would be a mixture of bases present at a given position, so that it will not be a “yes or no” answer and the fraction correctly detected will matter more. This aspect of the experiment should be reported more clearly.

The cDNA standards were sequenced in a same flowcell, in a multiplexed manner. Therefore, the background errors that are seen in the polyA tails of the cDNA standards in Figure 5c and S5b actually do not reflect the background base-calling errors (which we estimate to be 2.47% based the sequins run, see Table S11), but rather, they reflect demultiplexing errors from Guppy algorithm that is provided by ONT (e.g. some cDNA standards are incorrectly demultiplexed, and therefore some polyA-only standards are misassigned to the IntG barcode, causing the observed background signal in Figure 5c). This happens because in the case of cDNA standards, we employed cDNA standards that had the same identical sequence with different tail compositions, and therefore we rely on demultiplexing to assign a given read to a specific cDNA standard, whereas in normal cases, we would be able to assign a read to a specific read to a given cDNA standard (or transcript, if we were

using a biological sample) based on mapping, in addition to demultiplexing. For this reason, we only use the cDNA standards to demonstrate that Nano3P-seq captures a broad diversity of tail compositions (Figure 5c and S5b), but not to assess the basecalling error rate of the tails. To do the latter (assess basecalling error within the tail regions), we use the sequins run, which we include in Figures 5d and S6a. We would be happy to include an additional run of cDNA standards with different tail compositions that have been sequenced in independent flowcells, if the reviewer considers this point essential. However, we would like to remark the fact that in normal biological samples, this should not be an issue, as the assignment of reads to transcripts is done based on the mapping of the read.

8) Figure 5c: Would it be possible to provide the actual numbers used to draw those plots?

We have now provided the data with the underlying numbers used to build Figure 5c as Table S10.

9) Figure 5d: Is the expected % reads = 100 for all the Sequin controls? And is this also the case for Figure S6a and Tables 8 and 10? This should be explicitly stated in some place in the manuscript.

Yes, the expected % reads with polyA tail with A composition for sequin controls is 100%, as the polyA tail sequence of the 'sequins' is encoded as DNA sequence in the plasmid. We now clarified this in the manuscript (i.e. that 'sequins' are expected to have an homogeneous polyA tail).

10) Page 15: The authors state that only 50 ng of total RNA is needed for a Nano3P-seq library. It would be good to explicitly show a comparison of the results for 50 vs. 100 ng as input – or at least tell the reader where the 50 ng data can be found in the manuscript.

We only used 100 ng input for the initial optimisation experiments that were initially done, which have not been included in the manuscript. All the sequencing experiments included in this work have used 50 ng of RNA as input. Therefore, to avoid confusion, we have now simplified the methods section, and just mention using 50 ng input, which is what we have used in all the libraries that were finally included in this work.

11) Page 16: Should be “rule out” not “outrule.”

We thank the reviewer for this correction. This is now corrected in the manuscript.

12) Page 16: Citing the other paper for “low cost” is still not sufficient as it's not clear to which methods this comparison is being made. And it's not clear if all the costs in the Nano3P-seq method are included in the costs cited by the reference. Better to take out this contention if the authors do not want to include their actual costs.

We have now removed the word “low cost” from this sentence.

13) Page 18: What is the concentration of SUPERase In RNase Inhibitor?

We have now added the volume of SUPERase IN RNase Inhibitor used in the buffer.

14) Page 22: What quantity of RNA was used as input in the dRNA-seq experiments?

We used 450ng of polyA-enriched RNA as input for the dRNA-seq libraries. This information has now been added to the Methods section.

15) Page 23: It would be good to state explicitly how the analysis was done for the newly added cDNA controls with different poly(A) tail lengths and non-A bases in the Analysis of Nano3P-seq datasets section (or elsewhere in the Methods section).

We have now added detailed information on how we mapped and analysed the cDNA standards within the “Analysis of Nano3P-seq datasets” section.

16) Figure S1c: Should be “above” and “below” not “left” and “right”

We thank the reviewer for pointing out this typo. This has now been fixed in the figure legend.

17) Figure S6: mislabeled as Figure S5.

We thank the reviewer for pointing out this typo. We now fixed it in the supplementary figures document.

18) Table S11: Additional explanation in text about the problems with some of these samples – 90A and 120A. It was not clear whether these would be problematic in future experiments.

We have now removed this comment from Table S11 (now Table S12). Instead, we have now added a detailed explanation in the Methods section (within the section “Analysis of Nano3P-seq datasets”).

19) Library Preparation Protocol for Nano3P-seq

- a. “Materials and consumables required” – are these all of the reagents for entire protocol? Some items are mentioned later, but not listed here.

This is now fixed in the protocol (File S1).

- b. There should be part numbers and sources for each item here.

We have now added any missing information for all ‘materials and consumables’ in the protocol. We thank the reviewer for pointing us to this.

- c. What type of purification is needed for each oligo?

We have now added the purification type used for each oligo in the oligonucleotide table included as part of File S1.

- d. RnaseIN Promega – more information needed.

We now added the volume and the catalogue number of the RNase Inhibitor in this text and other parts of the manuscript where RNase Inhibitor was used.

- e. AMX Adapter Ligation – source and part numbers?

This is included as part of the Direct cDNA-Sequencing ONT protocol (SQK-DCS109). This is now clarified in the text.

20) Response to Reviewers: Authors discuss errors in base calling accuracy in polyA tails being mostly insertions and deletions in homopolymeric regions. Where is this shown in the manuscript? It's worth mentioning this specifically in the manuscript.

The fact that nanopore sequencing struggles to base-call homopolymeric regions (and then this mostly appears as “indels”, e.g. 5As are base-called as 4As and so on), is a well-documented phenomenon in the literature, and a known issue acknowledged by ONT. It is such a well-established phenomenon, that there have been several softwares developed by the community to precisely “correct” for those sequencing errors that specifically tend to accumulate in homopolymeric regions (e.g. Homopolish (Huang, Liu, and Shih 2021)).

For this reason, available softwares that have been developed to estimate polyA tail lengths from nanopore sequencing data (e.g. Nanopolish or TailfindR) do not estimate polyA tail lengths from the base-called sequence, but rather by comparing the relative duration of the current signal corresponding to the polyA tail region, relative to the duration of the read that contained it (e.g. Figure S3A). We have now briefly mentioned this in the Methods section, to make clear to the readers why we do not use basecalling to assess polyA tail lengths.

21) Response to Reviewers: Authors did not look at 5' vs. 3', GC, or length biases for the Sequins and actual RNA samples in their data. These are fairly standard analyses.

With regards to GC bias, NGS-sequenced data are known to exhibit GC-bias, mostly due to PCR amplification (Benjamini and Speed 2012; Chen et al. 2013), causing either GC-rich or AT-rich DNA sequences to have a lower depth of sequencing. However, several articles (Goldstein et al. 2019; Sevim et al. 2019; Delahaye and Nicolas 2021) have concluded that Nanopore sequencers do not suffer from such bias, as its library preparation does not require a PCR amplification step. We have now cited these works in the manuscript.

With regards to 5' and 3' biases, the nature of Nano3P-seq and direct RNAseq (where reads are only sequenced from their 3' end) will cause that coverage is higher at 3' than 5' ends. We now include some examples illustrating how the coverage looks in some sequins and zebrafish mRNAs, comparing side by side the coverage tracks of Nano3P-seq and direct RNAseq. We have now included this information as Figure S7h-i, also embedded below for the reviewer's convenience.

REFERENCES

- Benjamini, Yuval, and Terence P. Speed. 2012. "Summarizing and Correcting the GC Content Bias in High- Throughput Sequencing." *Nucleic Acids Research* 40 (10): e72.
- Chen, Yen-Chun, Tsunglin Liu, Chun-Hui Yu, Tzen-Yuh Chiang, and Chi-Chuan Hwang. 2013. "Effects of GC Bias in next-Generation-Sequencing Data on de Novo Genome Assembly." *PloS One* 8 (4): e62856.
- Delahaye, Clara, and Jacques Nicolas. 2021. "Sequencing DNA with Nanopores: Troubles and Biases." *PloS One* 16 (10): e0257521.
- Goldstein, Sarah, Lidia Beka, Joerg Graf, and Jonathan L. Klassen. 2019. "Evaluation of Strategies for the Assembly of Diverse Bacterial Genomes Using MinION Long-Read Sequencing." *BMC Genomics*. <https://doi.org/10.1186/s12864-018-5381-7>.
- Huang, Yao-Ting, Po-Yu Liu, and Pei-Wen Shih. 2021. "Homopolish: A Method for the Removal of Systematic Errors in Nanopore Sequencing by Homologous Polishing." *Genome Biology* 22 (1): 95.
- Sevim, Volkan, Juna Lee, Robert Egan, Alicia Clum, Hope Hundley, Janey Lee, R. Craig Everroad, et al. 2019. "Shotgun Metagenome Data of a Defined Mock Community Using Oxford Nanopore, PacBio and Illumina Technologies." *Scientific Data* 6 (1): 285.

Decision Letter, third revision:

Dear Eva,

Thank you for submitting your revised manuscript "Nano3P-seq: transcriptome-wide analysis of gene expression and tail dynamics using end-capture nanopore cDNA sequencing" (N METH-A47361C). It has now been seen by the original referee and their comments are below. The reviewers find that the paper has improved in revision, and therefore we'll be happy in principle to publish it in Nature Methods, pending minor revisions to satisfy the referees' final requests and to comply with our editorial and formatting guidelines.

TRANSPARENT PEER REVIEW

Nature Methods offers a transparent peer review option for new original research manuscripts submitted from 17th February 2021. We encourage increased transparency in peer review by publishing the reviewer comments, author rebuttal letters and editorial decision letters if the authors agree. Such peer review material is made available as a supplementary peer review file. Please state in the cover letter 'I wish to participate in transparent peer review' if you want to opt in, or 'I do not wish to participate in transparent peer review' if you don't. Failure to state your preference will result in delays in accepting your manuscript for publication.

Thank you again for your interest in Nature Methods Please do not hesitate to contact me if you have any questions.

Best regards,
Lei

Lei Tang, Ph.D.
Senior Editor
Nature Methods

ORCID

Reviewer #3 (Remarks to the Author):

This revised manuscript is much improved and addresses all of the concerns raised previously. The authors make a convincing case that comparison with published results for other methods is sufficient for this study. A few minor concerns remain.

Minor concerns.

- 1) Page 5: An explanation of “per-read analysis” should be provided in the text.
- 2) Page 12: “accurately estimated the non-A base content in the tails (Figure S6b)” – should this be Figure S5b?
- 3) Page 12: It would be helpful to explain the issues with base calling in Figure 5c in a Supplementary Note or the figure legend.
- 4) Page 16: For the comparisons with other methods, the Pearson correlations are low, as noted. Perhaps it would be worth also looking at Spearman correlations in case the relative tail lengths are more similar? Also, given that TAIL-seq is limited to a length of 100 bases, perhaps also report the correlation only for those poly(A) tails measured as under 100 bases by Nano3P-seq
- 5) Page 20: Where were the HeLa cells obtained? Were they tested for mycoplasma?
- 6) Figure S2j: Would it be possible to add to the figure legend the explanation provided about the Sanger sequencing chromatogram and the poly(A) tail?

Author Rebuttal, third revision:

ANSWERS TO REVIEWERS

Reviewer #3:

This revised manuscript is much improved and addresses all of the concerns raised previously. The authors make a convincing case that comparison with published results for other methods is sufficient for this study. A few minor concerns remain.

We would like to thank the reviewer for his/her comments and time in reviewing our work again. Please find below point-by-point responses to the minor concerns raised.

Minor concerns.

1) Page 5: An explanation of “per-read analysis” should be provided in the text.

We have now removed the term “per-read” to avoid possible confusion to the reader.

2) Page 12: “accurately estimated the non-A base content in the tails (Figure S6b)” – should this be Figure S5b?

Thank you for pointing this out, this should indeed be Figure S5b. We have now corrected this in the revised version of the manuscript.

3) Page 12: It would be helpful to explain the issues with base calling in Figure 5c in a Supplementary Note or the figure legend.

We have now added the following paragraph as Supplementary Note 1:

“Tail composition analyses in cDNA standards (Figure 5c and S5a) revealed slight inaccuracies in the predictions, mainly in cDNA standards containing 5U, 5C or 5G at their 3’ ends. We believe that this phenomenon is caused by the homopolymeric nature of these sequences. Indeed, previous reports have shown that increased base-calling errors (in the form of insertions and/or deletions) in nanopore sequencing datasets frequently occur at homopolymeric positions consisting of 5 or more identical consecutive bases. These inaccuracies are unlikely to be seen in biological systems, since Nano3P-seq (and previous literature) in in vivo datasets shows that the majority of homopolymeric non-A terminal ends are shorter than 5 bases.”

4) Page 16: For the comparisons with other methods, the Pearson correlations are low, as noted. Perhaps it would be worth also looking at Spearman correlations in case the relative tail lengths are more similar? Also, given that TAIL-seq is limited to a length of 100 bases, perhaps also report the correlation only for those poly(A) tails measured as under 100 bases by Nano3P-seq

Following the reviewer’s suggestion, we have now included the spearman correlation in Table S13. We found that correlation values did not improve when using Spearman correlation.

5) Page 20: Where were the HeLa cells obtained? Were they tested for mycoplasma?

HeLa cell pellets were obtained from a lab from our institute (mentioned in acknowledgements), who purchased the cell line from ATCC. Cells were tested for Mycoplasma contamination. We have now added this information to the Materials and Methods section.

6) Figure S2j: Would it be possible to add to the figure legend the explanation provided about the Sanger sequencing chromatogram and the poly(A) tail?

Following the reviewer's suggestion, we have now edited the figure legend, and have added the following text to clarify this point:

"Although the chromatogram only shows 4 A nucleosides based on the "consensus prediction", there are additional A bases that can also be seen in subsequent positions, which have smaller peaks than the G peaks, with decreasing frequency. This phenomenon is the result of sequencing an heterogeneous population of polyA tail lengths, where G/I tails are incorporated into RNA populations with slight differences in their polyA tail lengths."

Final Decision Letter:

Dear Dr Novoa,

I am pleased to inform you that your Article, "Nano3P-seq: transcriptome-wide analysis of gene expression and tail dynamics using end-capture nanopore cDNA sequencing", has now been accepted for publication in Nature Methods. Your paper is tentatively scheduled for publication in our February 2023 print issue, and will be published online prior to that. The received and accepted dates will be 15th Oct 2021 and 3rd Nov 2022. This note is intended to let you know what to expect from us over the next month or so, and to let you know where to address any further questions.

Once your paper is typeset, you will receive an email with a link to choose the appropriate publishing options for your paper and our Author Services team will be in touch regarding any additional information that may be required.

Please note that *Nature Methods* is a Transformative Journal (TJ). Authors may publish their research with us through the traditional subscription access route or make their paper immediately open access through payment of an article-processing charge (APC). Authors will not be required to make a final decision about access to their article until it has been accepted. [Find out more about Transformative Journals](https://www.springernature.com/gp/open-research/transformative-journals)

Your paper will now be copyedited to ensure that it conforms to Nature Methods style. Once proofs are generated, they will be sent to you electronically and you will be asked to send a corrected version within 24 hours. It is extremely important that you let us know now whether you will be difficult to contact over the next month. If this is the case, we ask that you send us the contact information (email, phone and fax) of someone who will be able to check the proofs and deal with any last-minute problems.

If, when you receive your proof, you cannot meet the deadline, please inform us at rjsproduction@springernature.com immediately.

Once your manuscript is typeset and you have completed the appropriate grant of rights, you will receive a link to your electronic proof via email with a request to make any corrections within 48 hours. If, when you receive your proof, you cannot meet this deadline, please inform us at rjsproduction@springernature.com immediately.

Once your paper has been scheduled for online publication, the Nature press office will be in touch to confirm the details.

Once your paper has been scheduled for online publication, the Nature press office will be in touch to confirm the details.

Content is published online weekly on Mondays and Thursdays, and the embargo is set at 16:00 London time (GMT)/11:00 am US Eastern time (EST) on the day of publication. If you need to know the exact publication date or when the news embargo will be lifted, please contact our press office after you have submitted your proof corrections. Now is the time to inform your Public Relations or Press Office about your paper, as they might be interested in promoting its publication. This will allow them time to prepare an accurate and satisfactory press release. Include your manuscript tracking number NMETH-A47361D and the name of the journal, which they will need when they contact our office.

About one week before your paper is published online, we shall be distributing a press release to news organizations worldwide, which may include details of your work. We are happy for your institution or funding agency to prepare its own press release, but it must mention the embargo date and Nature Methods. Our Press Office will contact you closer to the time of publication, but if you or your Press Office have any inquiries in the meantime, please contact press@nature.com.

Nature Portfolio journals [encourage authors to share their step-by-step experimental protocols](https://www.nature.com/nature-research/editorial-policies/reporting-standards#protocols) on a protocol sharing platform of their choice. Nature Portfolio 's Protocol Exchange is a free-to-use and open resource for protocols; protocols deposited in Protocol Exchange are citable and can be linked from the published article. More details can found at www.nature.com/protocolexchange/about.

Please note that you and any of your coauthors will be able to order reprints and single copies of the issue containing your article through Nature Portfolio 's reprint website, which is located at <http://www.nature.com/reprints/author-reprints.html>. If there are any questions about reprints please send an email to author-reprints@nature.com and someone will assist you.

Best regards,
Lei